**EMBO** *reports*

# Functional profiling and visualization of the sphingolipid metabolic network in vivo

Fei-Yang Tzou [1], Cheng-Li Hong [1], Kai-Hung Chen[1], John P Vaughen[2], Wan-Syuan Lin[1], Chia-Heng Hsu[1], Irma Magaly Rivas-Serna[3], Kai-Yi Hsu[4,5], Shuk-Man Ho[6,7], Michael Raphael Panganiban [8], Hsin-Ti Hsieh[8], Yi-Jhan Li [1], Yi Hsiao [1], Hsin-Chun Yeh[1], Cheng-Yu Yu[9], Hong-Wen Tang [10,11], Ya-Hui Chou [8], Chia-Lin Wu [6,7], Chung-Chuan Lo[4,5], Vera C Mazurak[3], M Thomas Clandinin[3], Shu-Yi Huang [12] & Chih-Chiang Chan [1,13]✉

## Abstract

Sphingolipids govern diverse cellular processes; their dysregulation underlies numerous diseases. Despite extensive characterizations, understanding the orchestration of the sphingolipid network within living organisms remains challenging. We established a versatile genetic platform of CRISPR-engineered reporters of 52 sphingolipid regulators, recapitulating endogenous gene activity and protein distribution. This platform further allows conditional protein degradation for functional characterization. In addition, we developed the biosensor OlyA^w to detect ceramide phosphoethanolamine and visualize membrane raft dynamics in vivo. Using this platform, we established comprehensive profiles of the sphingolipid metabolic network in the brain at the transcriptional and translational levels. The highly heterogeneous patterns indicate extensive coordination between distinct cell types and regions, suggesting the brain functions as a coherent unit to execute specific steps of sphingolipid metabolism. As a proof-of-concept application, we showed cell type-specific requirements of sphingomyelinases, including *CG6962/dSMPD4* and *CG3376/aSMase*, degrading distinct subcellular pools of ceramide phosphoethanolamine to maintain brain function. These findings establish a foundation for future studies on brain sphingolipid metabolism and showcase the utilization of this genetic platform in elucidating in vivo mechanisms of sphingolipid metabolism.

**Keywords** Brain; Sphingolipids; Systemic Profiling; Spatial Heterogeneity; Cell-type Specificity
**Subject Categories** Biotechnology & Synthetic Biology; Metabolism; Methods & Resources

## Introduction

Sphingolipids (SPLs) are a diverse group of lipids that serve as membrane structural components and signaling molecules. Deregulation of SPL metabolizing genes and the consequent alteration in SPL species is associated with numerous human diseases, including neurological disorders, skin diseases, and cancer (Dunn et al, 2019; Pan et al, 2023). Given their critical role in physiology and human diseases, SPL metabolic mechanisms have been extensively studied at biochemical and cellular levels (Hannun and Obeid, 2018). However, how SPL homeostasis is coordinated and maintained across tissues or at the organismal level is unclear.

With current methods, it is challenging to comprehensively examine the SPL metabolic network in vivo. Cell culture-based studies have identified regulatory mechanisms of SPL metabolism (Kumagai et al, 2007; Siow and Wattenberg, 2012; Köberlin et al, 2015; Senkal et al, 2017; Su et al, 2019; Fan et al, 2021) but lack the context of multicellular systems. Although knockout animals provide insights into the pathophysiological consequences following the chronic or acute loss of individual SPL enzymes and the associated SPL species (Ledesma et al, 2011; Jennemann and Gröne, 2013; Wegner et al, 2016; Hammerschmidt et al, 2019; Kuo et al, 2022), they may not reveal the full complexity of the entire SPL network. For a broad overview of SPL metabolic networks, transcriptomics, proteomics, and lipidomics have nominated candidate enzymes and lipids as present in different tissues and cell types (Jiang et al, 2020; Karlsson et al, 2021; Li et al, 2022;

[1]Graduate Institute of Physiology, College of Medicine, National Taiwan University, Taipei 100, Taiwan. [2]Department of Anatomy, University of California San Francisco, CA 94114, USA. [3]Department of Agricultural, Food, and Nutritional Science, University of Alberta, Edmonton, AB T6G 2R3, Canada. [4]Institute of Systems Neuroscience, National Tsing Hua University, Hsinchu 300044, Taiwan. [5]Brain Research Center, National Tsing Hua University, Hsinchu 300044, Taiwan. [6]Department of Biochemistry and Graduate Institute of Biomedical Sciences, College of Medicine, Chang Gung University, Taoyuan 33302, Taiwan. [7]Department of Neurology, New Taipei Municipal TuCheng Hospital, Chang Gung Memorial Hospital, New Taipei City 236017, Taiwan. [8]Institute of Cellular and Organismic Biology, Academia Sinica, Taipei 11529, Taiwan. [9]Interdisciplinary Program of Life Sciences and Medicine, National Tsing-Hua University, Hsinchu 300044, Taiwan. [10]Program in Cancer and Stem Cell Biology, Duke-NUS Medical School, 8 College Road, Singapore 169857, Singapore. [11]Division of Cellular & Molecular Research, Humphrey Oei Institute of Cancer Research, National Cancer Centre Singapore, Singapore 169610, Singapore. [12]Department of Medical Research, National Taiwan University Hospital, Taipei 100, Taiwan. [13]Research Center for Developmental Biology and Regenerative Medicine, National Taiwan University, Taipei 106, Taiwan. ✉E-mail: chancc1@ntu.edu.tw

Fitzner et al, 2020). However, exchanges of both proteins and lipids can occur between cells and tissues, and many SPL and lipid biosynthetic enzymes are regulated allosterically, necessitating functional analysis of the regulatory networks in a broader context. Thus, a streamlined approach is needed for the systemic profiling and functional assessment of SPL metabolism in vivo.

*Drosophila* provides a powerful platform for gene profiling and functional analysis of SPL metabolism in vivo. Core SPL metabolic pathways are evolutionarily conserved in flies (Acharya and Acharya, 2005), including those for sphingosine and ceramide, bioactive SPL derivatives (S1P, C1P), and complex sphingolipids, such as the analog of mammalian sphingomyelin, ceramide phosphoethanolamine (CerPE). Notably, the relatively low degree of genetic redundancy in *Drosophila* simplifies functional analysis; for example, flies have a single ceramide synthase as opposed to six in mammals (Levy and Futerman, 2010; Bauer et al, 2009). Previous studies leveraged *Drosophila* genetics to elucidate the physiological functions of SPL metabolism in vivo. Initial characterization of SPL enzymes indicated essential roles in body growth (Bauer et al, 2009), reproductive functions (Herr et al, 2004; Phan et al, 2007; Kunduri et al, 2022), tissue development (Herr et al, 2003; Hebbar et al, 2020; Tsarouhas et al, 2023; Hull et al, 2024), and survival (Rao et al, 2007). Moreover, tissue-specific analyses highlight complex SPL networks in the brain, with glial glucocerebrosidase regulating neurite remodeling (Vaughen et al, 2022) and neuronal survival (Wang et al, 2022b), and the CerPE synthetase gene, *Cpes*, generating glial membrane processes that enwrap axons (Ghosh et al, 2013) and cell bodies (Kunduri et al, 2018) to modulate neural activity and circadian behavior (Chen et al, 2022). While these studies identified key functions of specific SPL regulators, the interactions between these regulators and the overall SPL regulatory network remain poorly understood. Systemic profiling of SPL regulators in *Drosophila* will elucidate in vivo mechanisms of SPL homeostasis by revealing the spatiotemporal gene expression patterns, protein distribution, and functional requirements across the entire metabolic network.

Another technical hurdle for SPL research in living organisms is the lack of reliable and direct tools for visualizing SPL species in specific cells and compartments. While lipidomics by mass spectrometry is required to resolve the molecular features necessary for identifying individual lipid species, standard mass spectrometry profiling typically pools lipid species across entire organisms or tissues. Imaging mass spectrometry techniques, though promising, remain limited in the spatiotemporal resolution (Wang et al, 2022a; Miller et al, 2023). To visualize SPL dynamics in cells or subcellular compartments, fluorescently tagged SPLs and clickable SPL probes enable in situ visualization of SPLs (Chen and Devaraj, 2021; Götz et al, 2020) but are limited to cell culture-based studies. In contrast, genetically encoded lipid biosensors expressing fluorescently tagged lipid-binding motifs can illuminate endogenous lipid localization in living tissues (Hardie et al, 2015). Existing genetically encoded SPL probes are limited but have been deployed in cell culture (Ono et al, 2022; Niekamp et al, 2022) and used ex vivo to label subcellular CerPE in the brain (Bhat et al, 2015) and testis (Kunduri et al, 2022). Further developing genetically encoded SPL probes for in vivo studies would help decipher bulk lipidomics data, SPL enzyme mutant phenotypes, and the dynamic regulation of the SPL network in real-time.

Here, we established a genetic platform in *Drosophila* for the systemic profiling and manipulation of SPL regulators at both transcriptional and translational levels. The platform consists of four major components. First, we constructed a collection of CRISPR knock-in reporter animals for in situ profiling of transcriptional activity and protein distribution of 52 genes involved in SPL metabolism. Second, these reporter animals allow for genetic manipulations, such as shRNA-mediated knockdown or overexpression, inside cell types actively producing the SPL transcript. Third, we achieved selective alteration of SPL protein levels in specific cells with a genetically encoded degradation machinery. Fourth, we engineered a genetically encoded fluorescent probe for monitoring the dynamics of CerPE and SPL-rich membrane rafts in vivo, which we characterized across SPL perturbations using our toolkit. Using this platform, we performed systemic profiling of the SPL metabolic network in the brain and discovered that the expression and protein distributions of SPL regulators varied dramatically across brain regions and cell types, including the distinct segregation of lysosomal and non-lysosomal catabolic enzymes in the glia versus neurons, as well as the neuronal enrichment of glycosphingolipid biosynthetic enzymes. Together, our results suggest that SPL metabolism is likely coordinated across different cell types and regions in the brain.

As a proof-of-principle application, we demonstrated the cellular requirements for CerPE regulation in vivo by characterizing the *CG6962* gene (hereafter *dSMPD4*), a putative ortholog of the human neutral sphingomyelinase gene *SMPD4* that is associated with microcephaly (Magini et al, 2019). We localized dSMPD4 with high specificity to nuclear membranes in the mushroom body and gustatory neurons. Strikingly, removing *dSMPD4* caused nuclear blebbing, the accumulation of specific C18 CerPE lipids, and learning deficits. Furthermore, we showed that while acidic SMase (aSMase) is localized to glial lysosomes, it modulates CerPE degradation in both neurons and glia and regulates ref(2)P/p62 proteostasis and locomotor functions, suggesting that CerPE homeostasis in the brain is controlled by SMases in both cell-autonomous and nonautonomous manners. Altogether, this comprehensive genetic platform enables multifaceted expression and functional studies of SPL metabolism in a living multicellular organism.

## Results

### HG knock-in animals of SPL metabolism genes reveal endogenous gene expression and protein localization

To investigate the SPL metabolic network (Fig. 1A) within endogenous regulatory contexts and under physiological conditions in vivo, we introduced a 3XHA-T2A-GAL4 (hereafter referred to as HG; Fig. 1B) reporter cassette by CRISPR/Cas9-mediated knock-in into the immediate upstream of the stop codon of each of the 52 SPL regulatory genes, including 46 enzymes, an enzyme inhibitor, an enzyme activator, and four lipid transfer proteins (Fig. 1A; Table EV1). The inclusion of the self-cleaving T2A peptides induces ribosome skipping (Diao and White, 2012), leading to translations of a 3XHA-tagged endogenous protein and a transcription activator GAL4 from the same mRNA transcript (Fig. 1B). We isogenized the HG lines to the $w^{1118}$ background

## A

### Sphingolipid metabolic network

**Human gene**
***Fly orthologs with 3XHA-T2A-GAL4 knockin***

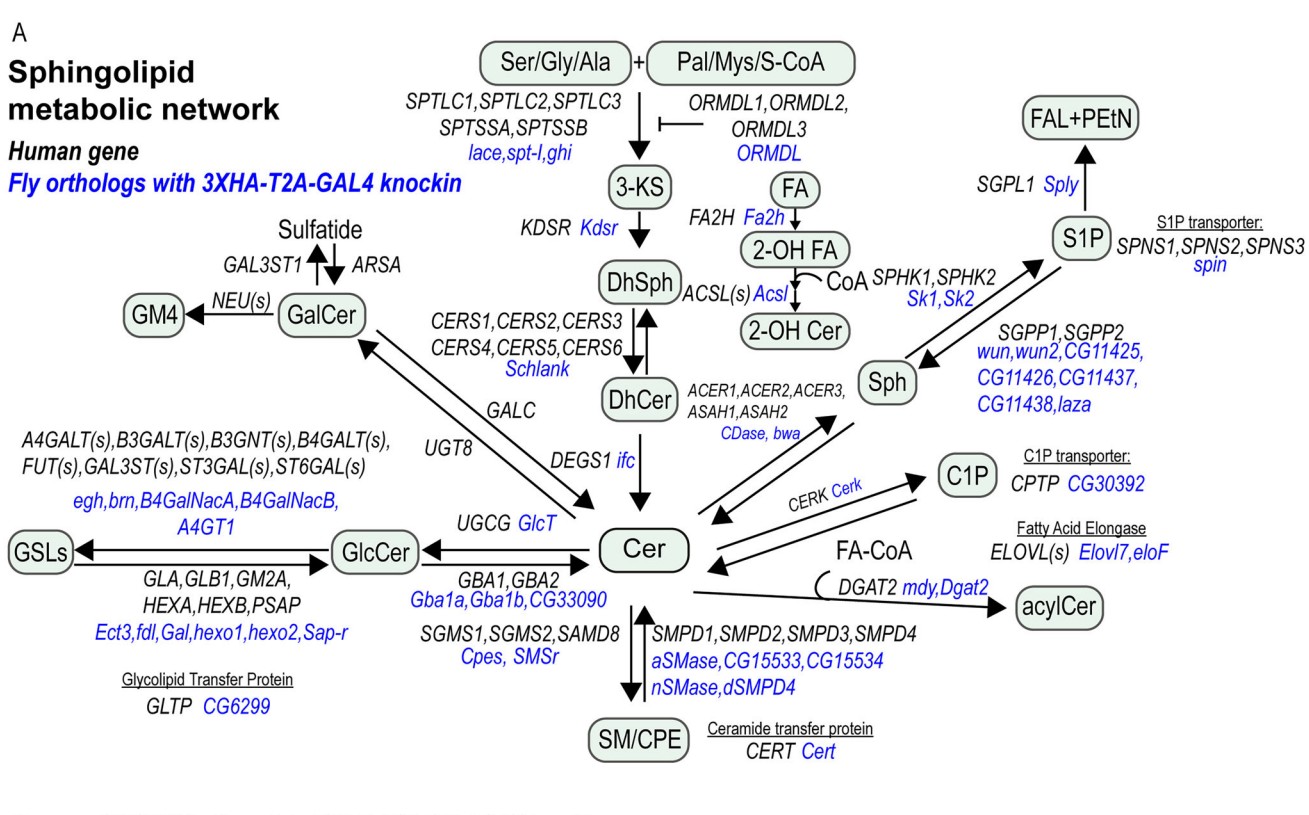

## B

CRISPR/Cas9-mediated 3XHA-T2A-GAL4 (HG) knockin

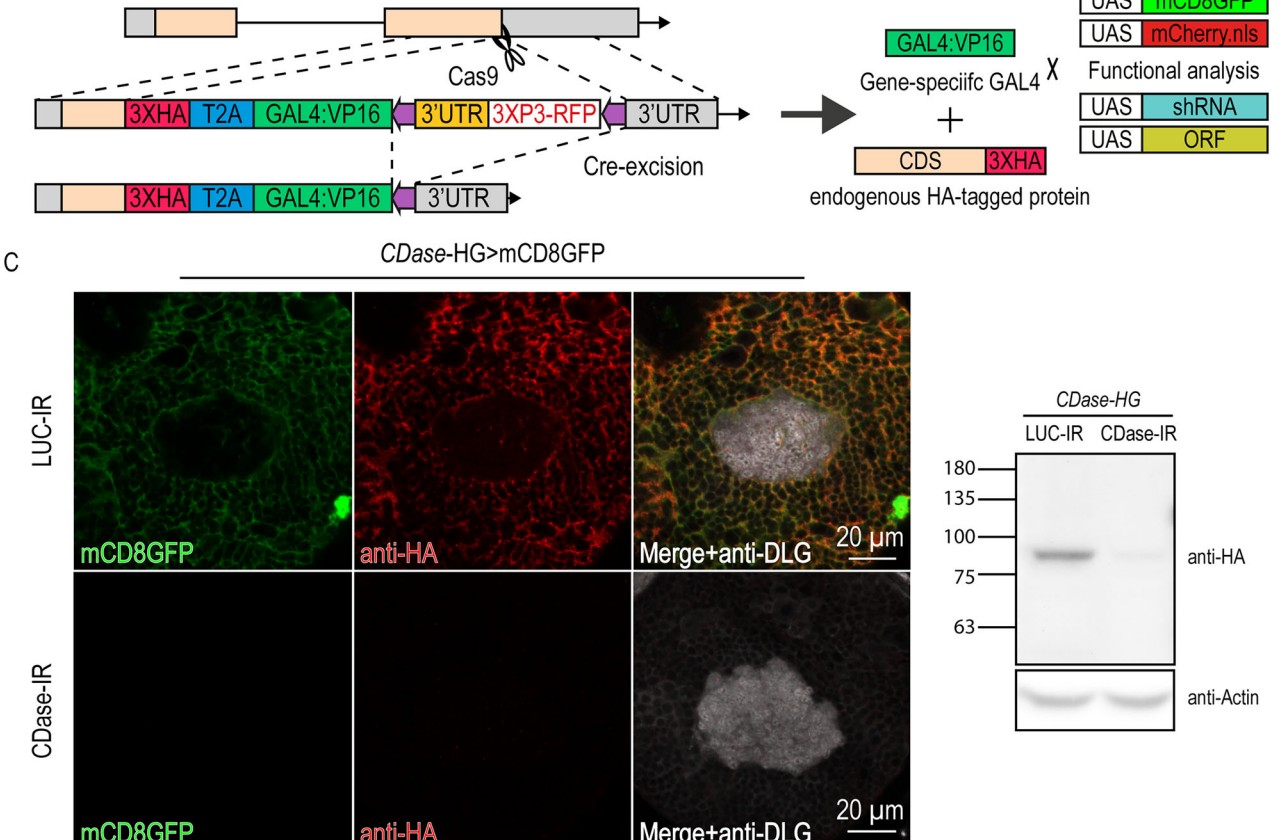

## C

*CDase*-HG>mCD8GFP

**Figure 1.** CRISPR/Cas9-mediated 3XHA-T2A-GAL4 (HG) knock-in produces 3XHA-tagged protein and the GAL4 from the endogenous transcript.

(A) The conserved metabolic pathways in flies and humans. Human genes are colored in black; fly homologs are colored in blue. The HG collection includes all 52 SPL regulators shown in the figure. (B) Schematic of the reporter cassette insertion. The 3xHA-tagged CDS and GAL4 are translated into two separate proteins from the same transcript due to the T2A sequence in between. *Hsp70Ba* 3'UTR stabilizes the transcript before Cre-mediated excision of the 3XP3RFP selection marker, and it is stabilized by endogenous 3'UTR after excision. The mCD8GFP and mCherry.nls driven by gene-specific GAL4 represent transcriptional profiles. RNAi knockdown or transgenic overexpression driven by gene-specific-GAL4 allows for genetic manipulations with spatiotemporal precision. (C) Validation of GAL4 expression and anti-HA immunostainings using *CDase* RNAi knockdown driven by *CDase*-HG. (Left) The expression pattern (mCD8GFP; green) and the protein distribution (anti-HA immunostaining; magenta) of CDase-3XHA in the posterior view of the calyx from the adult brain of young flies (1-week-old). The RNAi knockdown of *CDase* (*CDase*-HG>*CDase*-IR) driven by *CDase*-HG decreased both mCD8GFP and anti-HA immunoreactivity compared to the control knockdown (*CDase*-HG>*luciferase (LUC)-IR*). (Right) Anti-HA immunoblotting of the total lysate of adult *CDase*-HG flies. The level of CDase-3XHA protein was reduced using *CDase*-RNAi driven by *CDase*-HG. Source data are available online for this figure.

through three generations of backcrossing and then examined whether the HG insertion caused any lethality. Although the coding sequence and regulatory elements were not edited, 5 out of 52 isogenized HG lines, including *schlank, Acsl, Elovl7, spin*, and *Sap-r*, showed lethality at the adult stage (Table EV1). In addition, when these five lines were crossed with Cre stocks to remove the 3XP3RFP selection marker, the genotype of interest could not be recovered. Besides the five lethal alleles, the *CG11426* HG line was infertile after Cre excision. These observations suggest that while most HG insertions did not cause any gross phenotypes, there are instances where HG insertion can disrupt gene function, potentially from adding the 3XHA tag to the c-terminus of the protein.

Given that 3'UTRs can regulate cell type-specific protein expression (Merritt et al, 2008), we evaluated whether the *Hsp70Ba* 3'UTR introduced with the HG cassette affects SPL gene expression (Fig. 1B). We tested 11 HG lines and observed some discrepancies in a few of the HG lines before and after Cre excision (Appendix Fig. S1-S3 and Appendix Table S1). Therefore, subsequent gene profiling used HG lines after Cre excision, where only the endogenous 3'UTRs were present. Of the 52 HG lines, we used pre-Cre alleles of *schlank, Acsl, Elovl7, spin, Sap-r*, and *CG11426* for gene profiling, as their post-Cre alleles could not be obtained.

Next, we tested whether the 3XHA-tagged protein and the GAL4 were translated from the endogenous transcript by knocking down ceramidase (encoded by *CDase*) using *CDase*-RNAi driven by *CDase*-HG (Fig. 1C). Indeed, targeting the *CDase* transcript led to a simultaneous reduction in both mCD8GFP and anti-HA signals (Fig. 1C). The co-regulation supports that the 3XHA-tagged protein was translated from the same transcript as GAL4.

We then investigated the expression and localization of the 3XHA-tagged protein in HG lines. The anti-HA immunoblots of the total lysate of adult flies detected a dominant band at the predicted molecular weight for 41 of the 52 HG flies, whereas 11 lines were below the detection limit (Fig. 2A; Appendix Fig. S4; Table EV1). To examine whether the localization of the 3XHA-tagged protein in HG lines resembles that of the endogenous protein, we compared the anti-HA immunostainings to antibody staining results that have been reported in the literature. We showed the colocalization (Pearson's $R^2 = 0.65$) of anti-HA and anti-Ifc immunofluorescence in the L3 salivary glands of the heterozygote of the *ifc*-HG line (Fig. 2B). Moreover, anti-HA immunostaining of the *schlank*-HG revealed both ER and nuclear localizations, consistent with the known subcellular distribution of Schlank (Bauer et al, 2009; Sociale et al, 2018) (Fig. EV1A). These data indicate that HG knockins can reveal endogenous protein localization.

We next validated GAL4 expression patterns by comparing HG lines with two independent methods of gene expression profiling: CRIMIC-GAL4 (Lee et al, 2018) and single-cell transcriptomics of the adult brain (Davie et al, 2018). CRIMIC is a large collection of GAL4 lines that insert an artificial exon containing splice acceptor, T2A-GAL4, and a polyadenylation signal into the first coding intron of the gene (Lee et al, 2018). The CRIMIC-GAL4s thereby recapitulate gene expression by using the promoter activity while causing loss of function by arresting the transcription of targeted genes, and we obtained 11 of the CRIMIC lines in SPL genes. First, we compared the adult brain expression patterns of our HG lines with 3 genes, *sk1, gba1b*, and *CDase*, from the CRIMIC-GAL4 collection that have been validated in the literature (Wang et al, 2022b; Vaughen et al, 2022; Chung et al, 2023). For a quantitative comparison, we labeled neuronal and glial nuclei by antibodies and tallied the percentage of HG or CRIMIC-GAL4-driven nls-mCherry that colocalized with neurons and glia. Both HG and CRIMIC-GAL4 revealed glia-enriched expressions of *sk1, gba1b*, and *CDase*, despite variations in the numbers of nls-mCherry positive cells (Figs. 2C–E and EV1B–D). We further compared the 8 other CRIMIC-GAL4s with HG for cell-type expression patterns in the brain (Appendix Fig. S1). Five out of the 8 genes showed consistent cell-type expression patterns, except for *lace, mdy*, and *sply* (Appendix Table S1). HG lines of *lace* and *sply* revealed neuron-enriched expression, while the corresponding CRIMIC-GAL4 showed glial enrichment (Appendix Fig. S1F,G). In contrast, *mdy* HG lines showed glial enrichment, while *mdy* CRIMIC-GAL4 indicated higher neuronal expressions (Appendix Fig. S1H). We further compared the cell-type expression patterns of these genes with single-cell transcriptomics of the adult brain (Appendix Fig. S1I–K; Davie et al, 2018; Data ref: Davie et al, 2018). Clustering revealed both glial and neural expression of *sply* and *lace* (broader than CRIMIC and HG lines), and strong glial enrichment of *mdy* (consistent with HG lines). We chose *wun2* for further validation against the patterns of SPL metabolic genes that were well resolved in subtype glia in the transcriptomic dataset (Davie et al, 2018; Data ref: Davie et al, 2018), because *wun2* was identified as a marker gene of an astrocyte-like glia cluster (Fig. 2F). UAS-mCD8GFP expressed from the HG allele of *wun2* colocalized with LexAop-rCD2RFP signals driven by an astrocyte-like glia-specific GMR25H07-LexA (Fig. 2G), consistent with a recent report that *wun2* functions in astrocyte-like glia (Chen et al, 2024). In summary, the HG lines showed consistent expression profiles with the CRIMIC-GAL4s and single-cell transcriptomics, supporting that HG knock-in lines recapitulate the endogenous transcriptional and translational profiles of the targeted genes.

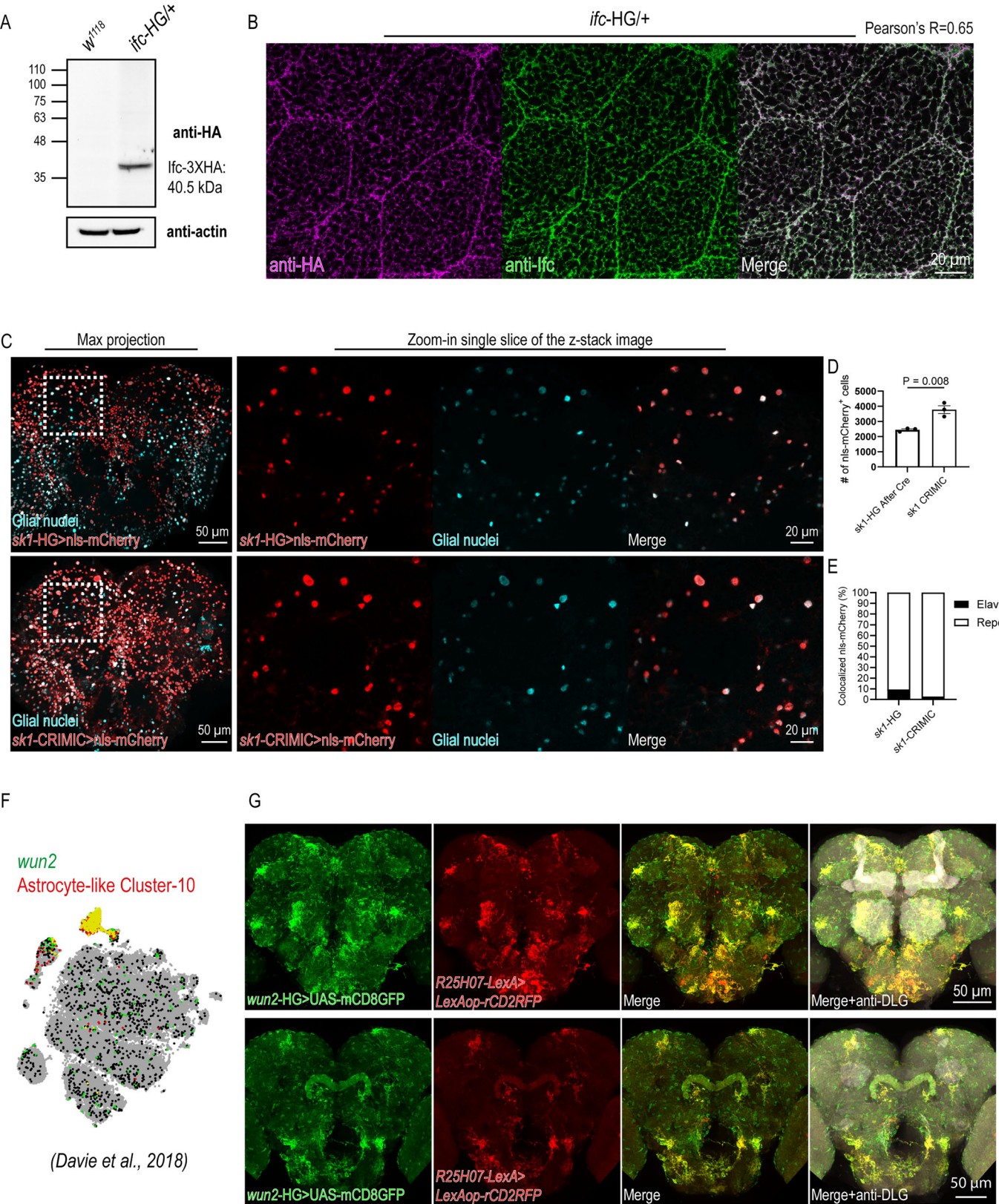

**Figure 2. HG lines recapitulate endogenous protein localization and gene expression.**

(A) Anti-HA immunoblots of $w^{1118}$ and heterozygous *ifc*-HG (*ifc*-HG/+) young adult flies (1 week of age). (B) Anti-HA and anti-Ifc (Jung et al, 2017) co-stainings of heterozygous *ifc*-HG (*ifc*-HG/+) L3 salivary glands. (C) Comparison of gene expression by *sk1*-HG and *sk1* CRIMIC-GAL4 expressions. The GAL4 expression in the adult brain of 1-week-old flies is visualized with UAS-nls-mCherry. The colocalization of mCherry with the glial marker Repo by immunostaining (Cyan) indicates glia-enriched expression of *sk1*. (D) Quantification of nls-mCherry positive cells driven by *sk1*-HG and *sk1*-CRIMIC. Data are represented as mean ± SEM of three biological repeats. The P value (P = 0.008) was calculated using a two-tailed unpaired Student's t test. (E) Quantification of nls-mCherry colocalized to neuronal and glial markers. Bar graphs showed the percentage of nls-mCherry spots colocalizing to neuronal nuclei (anti-Elav; black bars) and glial nuclei (anti-Repo; white bars) in the total number of labeled nls-mCherry (colocalizing with Elav or Repo). (F) Visualization of *wun2* cellular expression using SCENIC t-SNEs of the 57 K dataset from (Davie et al, 2018; Data ref: Davie et al, 2018). (G) Comparison of the expression pattern of *wun2*-HG with that of *GMR25H07*-LexA, which is a known astrocyte-like glia driver in the brain (top, anterior view; bottom, posterior view). The GAL4 expression is visualized with UAS-mCD8GFP, and the LexA expression is visualized with LexAop-rCD2RFP. The colocalization of mCD8GFP (Green) and rCD2RFP (Red) indicates that *wun2* predominantly expressed in the astrocyte-like glia. Source data are available online for this figure.

## The HG knock-in animals revealed heterogeneous gene expression of the SPL metabolic network in the brain

Disruptions of SPL metabolism cause developmental and degenerative neurological disorders (Pan et al, 2023), indicating that SPLs are critically required in the nervous system. Therefore, we focused on the brain for further profiling of each SPL gene using the HG toolkit. To our surprise, we found that the expression patterns of these genes were highly heterogeneous across cell types (Fig. 3; Appendix Figs. S5–21) and different brain regions (Fig. EV2; Appendix Figs. S22–30).

To quantify the expression of SPL regulatory genes in neuron versus glia, we visualized gene expression using nls-mCherry and co-stained with antibodies targeting neuronal (anti-Elav) and glial (anti-Repo) nuclei (Fig. 3A,B; Appendix Figs. S5–21). Of the 52 genes we analyzed (Fig. 1A), 20 genes were enriched in neurons and 18 in glia (Fig. 3C). Strikingly, genes that are involved in glycosphingolipids biosynthesis, including *GlcT, egh, brn, CG14517,* and *CG17223*, were almost exclusively expressed in neurons (>99% neuronal, $P < 0.001$; Fig. 3A,C), whereas lysosomal degradation enzymes were strongly enriched in glia (>90% glial, $P < 0.001$), including *gba1b, aSMase,* and *CDase* (Fig. 3B,C). In contrast, non-lysosomal degradative enzymes, including *sply* (99.08% neuronal, $P < 0.001$), *CG33090* (96.05% neuronal, $P = 0.017$), and *dSMPD4* (93.60% neuronal, $P = 0.034$) were significantly enriched in neurons. Altogether, the unexpected segregation of gene expression patterns in neurons versus glia provides the foundation for studying cell-type-specific mechanisms of SPL metabolism.

To gain insight into this regionalization of gene activities, we aligned 3D confocal images of HG-driven mCD8GFP, which labels cellular processes, to a standardized brain wiring map (Appendix Fig. S31) (Chiang et al, 2011). Notably, *dSMPD4, SMSr, nSMase,* and *ORMDL* were separated from other genes and formed a cluster based on similar expression patterns with higher levels in the mushroom body (Fig. EV2A,B). These enzymes play a role in the feedback control of ceramide synthesis: neutral sphingomyelinases drive ceramide production through positive feedback (Jaffrézou et al, 1998), while SMSr and ORMDL negatively regulate ceramide production (Siow and Wattenberg, 2012; Vacaru et al, 2009). This functional segregation of a subset of SPL regulators may indicate co-regulation of these SPL genes in this brain region. We further categorized 25 brain regions into clusters according to their correlations in the expression profiles of SPL genes. The clustering of distinct neuropils indicated the modular organization of the SPL metabolic network (Fig. EV2C). These observations suggested that

the regulation of SPL gene expression is functionally organized within distinct anatomical regions of the brain.

## Endogenous tagging reveals autonomous SPL protein regionalization and nonautonomous SPL protein uptake in the brain

The 3XHA tags on the endogenous SPL proteins enable the systemic documentation of protein distribution. We co-stained the brain of each HG line with HA, neuronal, and glial markers to visualize protein distributions (Fig. 4A,B; Appendix Figs. S32–47). Most HG lines (40 out of 52) showed significantly higher anti-HA immunoreactivity than the $w^{1118}$ control (Table EV1). To correlate protein distributions with HG transcript expression, we quantified the intensity of the anti-HA signals surrounding neuronal or glial nuclei and analyzed the neuron-to-glia ratio of anti-HA immunoreactivity (Fig. 4C). We found that neuronally enriched genes, including *schlank, dSMPD4, kdsr, CG33090, CG30392, lace, spin, Acsl, egh, cerk, SMSr,* and *fdl*, have significantly higher neuron/glia ratios of anti-HA immunoreactivity than the $w^{1118}$ control (Fig. 4C). In contrast, the neuron/glia ratio of anti-HA immunoreactivity of the glial enriched genes, including *sk1* and *CDase*, was significantly lower than the $w^{1118}$ control, consistent with the cell type-enriched gene expression (Fig. 3C). Surprisingly, despite not expressing transcriptionally in the brain, three SPL catabolic enzymes robustly accumulated in the brain in a cell-nonautonomous manner (*gba1a, CG15533,* and *CG15534*; Fig. EV3). Interestingly, all three enzymes are mainly expressed by the fly midgut (Leader et al, 2018; Kinghorn et al, 2016) and are predicted to contain signal peptides known to direct protein secretion (Teufel et al, 2022) (Appendix Fig. S48). As the midgut generates lipoproteins that can travel to the brain (Brankatschk and Eaton, 2010; Palm et al, 2012), these enzymes may nonautonomously regulate glucosylceramide (Gba1a) or CerPE (CG15533 and CG15534) catabolism of dietary SPLs accumulated in the brain. Taken together, SPL metabolism in the brain appears to be largely maintained cell-autonomously, but nonautonomous machinery may also play a role.

## Controlled protein degradation by the deGradHA system

To complement the gene manipulations at the DNA and mRNA levels, we directly manipulated SPL enzyme protein levels by combining the HG toolkit with deGradHA, which comprises an F-box domain of the *Drosophila* E3 ligase slmb and an anti-HA nanobody for proteasomal degradation of HA-tagged protein (Fig. 5A) (Caussinus et al, 2012; Vigano et al, 2021). Because HG

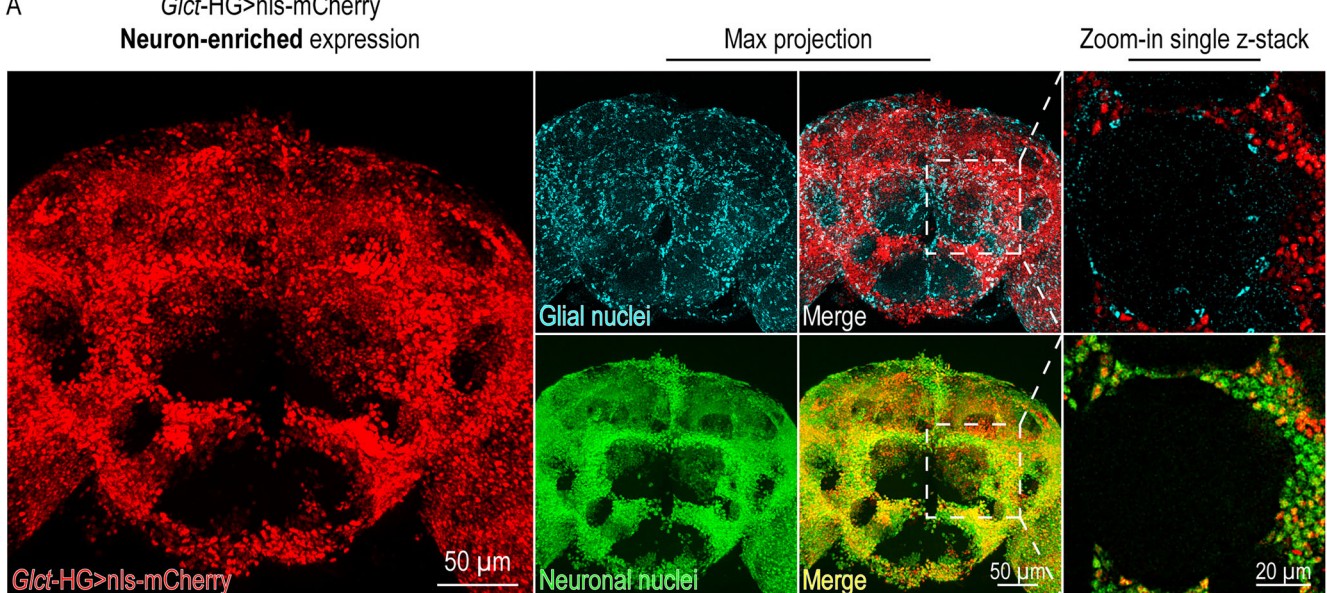

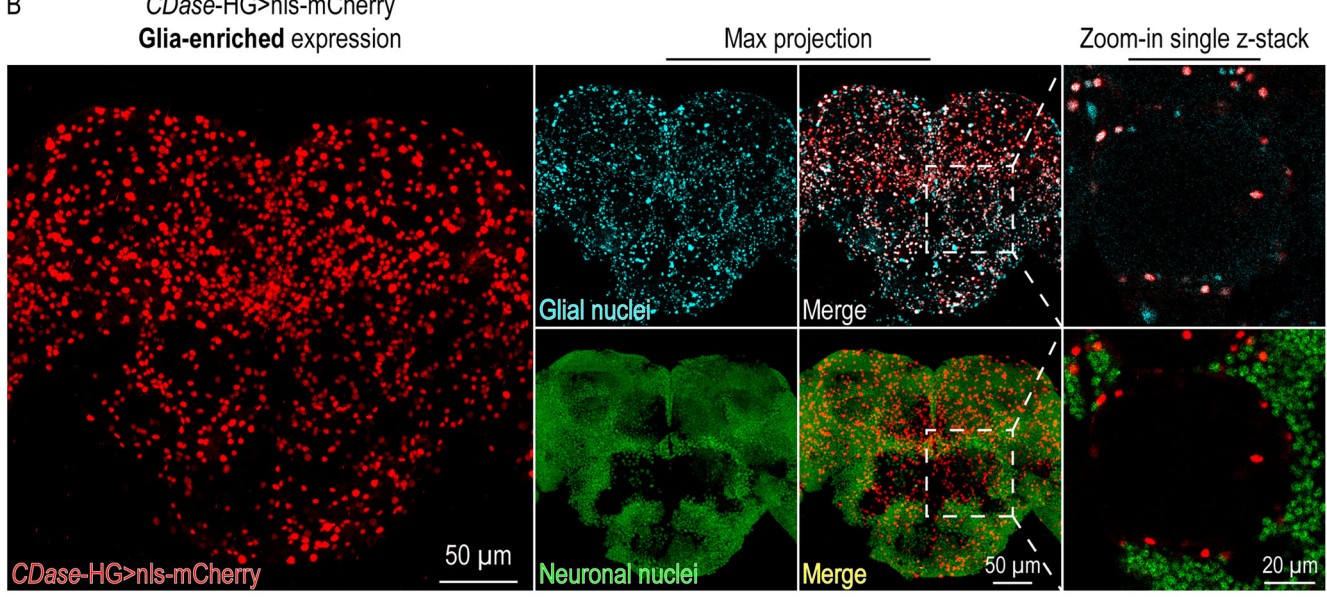

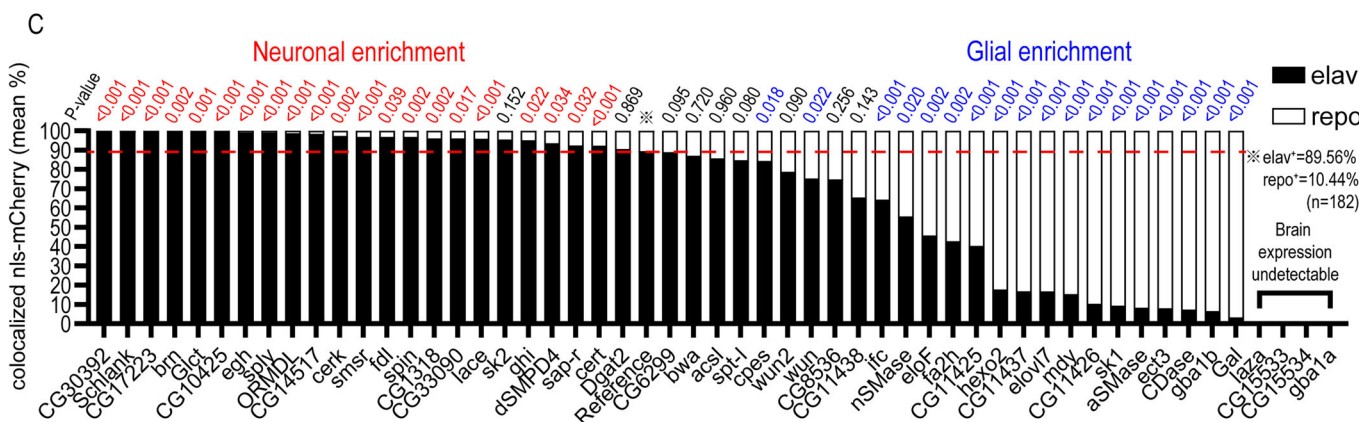

**Figure 3. Transcriptional profiling with HG revealed cell-type-enriched expression patterns of SPL regulators in the brain.**

(A) Neuronal enrichment of *Glct*-HG expression. The cell-type expression pattern of *Glct* genes in the central brain of 1-week-old flies was visualized with UAS-nls-mCherry and co-stainings of neuronal (anti-Elav, green) and glial nuclei (anti-Repo; cyan). (B) Glial enrichment of *CDase*-HG expression. The cell-type expression pattern of *CDase* genes in the central brain of 1-week-old flies was visualized with UAS-nls-mCherry and co-stainings of neuronal (anti-Elav, green) and glial nuclei (anti-Repo; cyan). (C) Quantification and statistical analyses of nls-mCherry colocalized to neuronal and glial markers in whole-mount central brains. Bar graphs showed the percentage of nls-mCherry spots colocalizing to neuronal nuclei (anti-Elav; black bars) and glial nuclei (anti-Repo; white bars) in the total number of labeled nls-mCherry (the sum of Elav- and Repo-colocalizing nls-mCherry spots). The red dashed line indicates the percentage of Elav spots (89.56%, calculated from 182 brains) in the total number of labeled nuclei (the sum of Elav and Repo). Data were presented as mean percentages of at least two independent experiments with $n \geq 3$ biological repeats. The neuronal or glial enrichment of gene expression is defined based on $P$ values calculated using two-tailed unpaired Student's $t$ test (*CG30392*, $P < 0.001$; *Schlank*, $P < 0.001$; *CG17223*, $P < 0.001$; *brn*, $P = 0.002$; *GlcT*, $P = 0.001$; *CG10425*, $P < 0.001$; *egh*, $P < 0.001$; *sply*, $P < 0.001$; *ORMDL*, $P < 0.001$; *CG14517*, $P < 0.001$; *cerk*, $P = 0.002$; *smsr*, $P < 0.001$; *fdl*, $P = 0.039$; *spin*, $P = 0.002$; *CG1318*, $P = 0.002$; *CG33090*, $P = 0.017$; *lace*, $P < 0.001$; *sk2*, $P = 0.152$; *ghi*, $P = 0.022$; *dSMPD4*, $P = 0.034$; *cert*, $P = 0.033$; *Dgat2*, $P = 0.870$; *CG6299*, $P = 0.095$; *bwa*, $P = 0.720$; *acsl*, $P = 0.961$; *spt-I*, $P = 0.080$; *Cpes*, $P = 0.018$; *wun2*, $P = 0.090$; *wun*, $P = 0.022$; *CG8536*, $P = 0.259$; *CG11438*, $P = 0.143$; *ifc*, $P < 0.001$; *nSMase*, $P = 0.020$; *eloF*, $P = 0.002$; *fa2h*, $P = 0.002$; *CG11425*, $P < 0.001$;.*hexo2*, $P < 0.001$; *CG11437*, $P < 0.001$; *elovl*, $P < 0.001$; *mdy*, $P < 0.001$; *CG11426*, $P < 0.001$; *sk1*, $P < 0.001$; *aSMase*, $P < 0.001$; *ect-3*, $P < 0.001$; *CDase*, $P < 0.001$; *gba1b*, $P < 0.001$; *Gal*, $P < 0.001$) comparing the means between the percentage of nls-mCherry colocalizing with Elav in the total number of labeled nls-mCherry (the sum of Elav- and Repo-colocalizing nls-mCherry spots) and the percentage of Elav in the total number of labeled nuclei (the sum of Elav and Repo spots). Representative figures of nls-mCherry images of all SPL regulators are presented in Appendix Figs. S5–21. Source data are available online for this figure.

lines contain gene-specific GAL4 activity, we generated a LexAop-deGradHA transgene and used the LexA/LexAop expression system for cell type-specific protein depletion (Fig. 5B). Given that HG GAL4 activity showed that *dSMPD4* is preferentially expressed in neurons and that *aSMase* exhibited glia-enriched expression (Fig. 3C), we targeted these SMases by neuronal or glial expression of deGradHA. Western blot results showed that neuronal expression of deGradHA (driven by *nSyb*-LexA) drastically reduced the levels of neuronal-enriched dSMPD4-3XHA protein but not the glia-enriched aSMase-3XHA protein in the brain (Figs. 5C and EV4A). In contrast, glial expression of deGradHA (driven by *repo*-LexA) depleted aSMase-3XHA while leaving dSMPD4-3XHA levels unchanged (Figs. 5D and EV4B). These results demonstrated that deGradHA can control protein degradation in a cell-type-specific manner. We then utilized the deGradHA system to test the protein distribution of nSMase-3XHA in neurons versus glia. *nSMase* showed a relatively balanced expression in the numbers of nls-mCherry colocalizing with neurons and glia (55.6% neuron, 44.4% glia; Fig. 3C). We found that the overall levels of nSMase-3XHA protein were significantly reduced by neuronal deGradHA but not glial deGradHA (Fig. EV4C,D), indicating that the protein is enriched in neurons. The Western blot results were corroborated by immunostainings of 3XHA-tagged SMases with neuronal or glial nuclei markers, further confirming the cell type-specific protein depletion (Figs. 5E and EV4E,F).

### The Ostreolysin A transgene associates with membrane rafts and indicates the abundance and dynamics of CerPE

CerPE is a structural analog of SM and the most abundant SPL detected in cell membranes in the fly. We constructed a genetically encoded biosensor to visualize how diverse SMases regulate cellular or subcellular CerPE. The mushroom-derived protein Ostreolysin A strongly and selectively associates with a CerPE and cholesterol mixture (Bhat et al, 2015; Endapally et al, 2019) and thus enables the detection of CerPE and its associated microdomains in vivo. Since SM/CerPE is mainly located on the outer leaflet of the plasma membrane, we incorporated a signaling peptide and the bright green fluorescent tag mWasabi to Ostreolysin A (hereafter referred to as OlyA$^w$) (Fig. 6A). OlyA$^w$ overexpression did not affect neuronal function in 1-week- and 3-week-old flies as shown by

normal depolarization of photoreceptors (Fig. EV5A), suggesting overexpression of OlyA$^w$ did not cause neurotoxicity. This was consistent with a previous study showing that OlyA treatment alone is non-toxic to insects (Panevska et al, 2019).

We then examined the subcellular distribution of OlyA$^w$ using both the brain and L3 salivary glands, which contain larger cells more amenable to subcellular localization. While the CerPE synthase (Cpes) located mainly in the Golgi apparatus (Vacaru et al, 2013), OlyA$^w$ did not colocalize with Golgi markers GM130 (*cis*-Golgi) and Golgin245 (*trans*-Golgi) (Appendix Fig. S49). Instead, the OlyA$^w$ fluorescence is mainly found in cortical ER in the L3 salivary glands (Appendix Fig. S49). In both adult brains and L3 salivary glands, OlyA$^w$ preferably colocalized with the ER membrane marker calnexin over other organelles (Appendix Figs. S49 and 50). Despite a lack of obvious visual overlap with the Golgi markers, quantitative analysis showed a modest correlation ($r = 0.262$) between OlyA$^w$ and anti-GM130 signals, which could be attributed to non-specific antibody staining within the cortical ER. In *Cpes*-knockout salivary glands, the OlyA$^w$ colocalization with calnexin was slightly reduced (Fig. 6B,C). These results indicated that while intracellular localizations of OlyA$^w$ are partially dependent on *Cpes*, the background fluorescence of ER retention limits its utility for reporting the subcellular localization of CerPE lipids.

Next, we validated whether the intensity of OlyA$^w$ fluorescence correlates with cellular levels of CerPE. The fluorescent intensity of OlyA$^w$ on the ER membrane significantly decreased in *Cpes*-knockout salivary glands (Fig. 6D). Moreover, knockdown of *Cpes* also reduced OlyA$^w$ intensity in the salivary gland (Fig. EV5B), while overexpressing *Cpes* significantly increased OlyA$^w$ intensity (Fig. EV5C). In addition, *aSMase* knockdown led to a drastic increase in the OlyA$^w$ signal (Fig. EV5D). Importantly, overexpression of Cpes also significantly increased OlyA$^w$ intensity in the adult brain (Fig. EV5E). These data demonstrate that OlyA$^w$ fluorescence intensity correlates with CerPE levels.

Because OlyA$^w$ contains a secretory peptide, we utilized the unique organization of the *Drosophila* compound eyes to test if the OlyA$^w$ probe is secreted into extracellular spaces and can label neighboring cells. In the cross-sectional view of the compound eye, seven photoreceptors can be seen in each ommatidium, with pigment glia in the interommatidial space (Fig. 6E). When driven by the pigment glia-specific *54C-GAL4*, OlyA$^w$ fluorescence primarily labeled the interommatidial space, with only a few OlyA$^w$

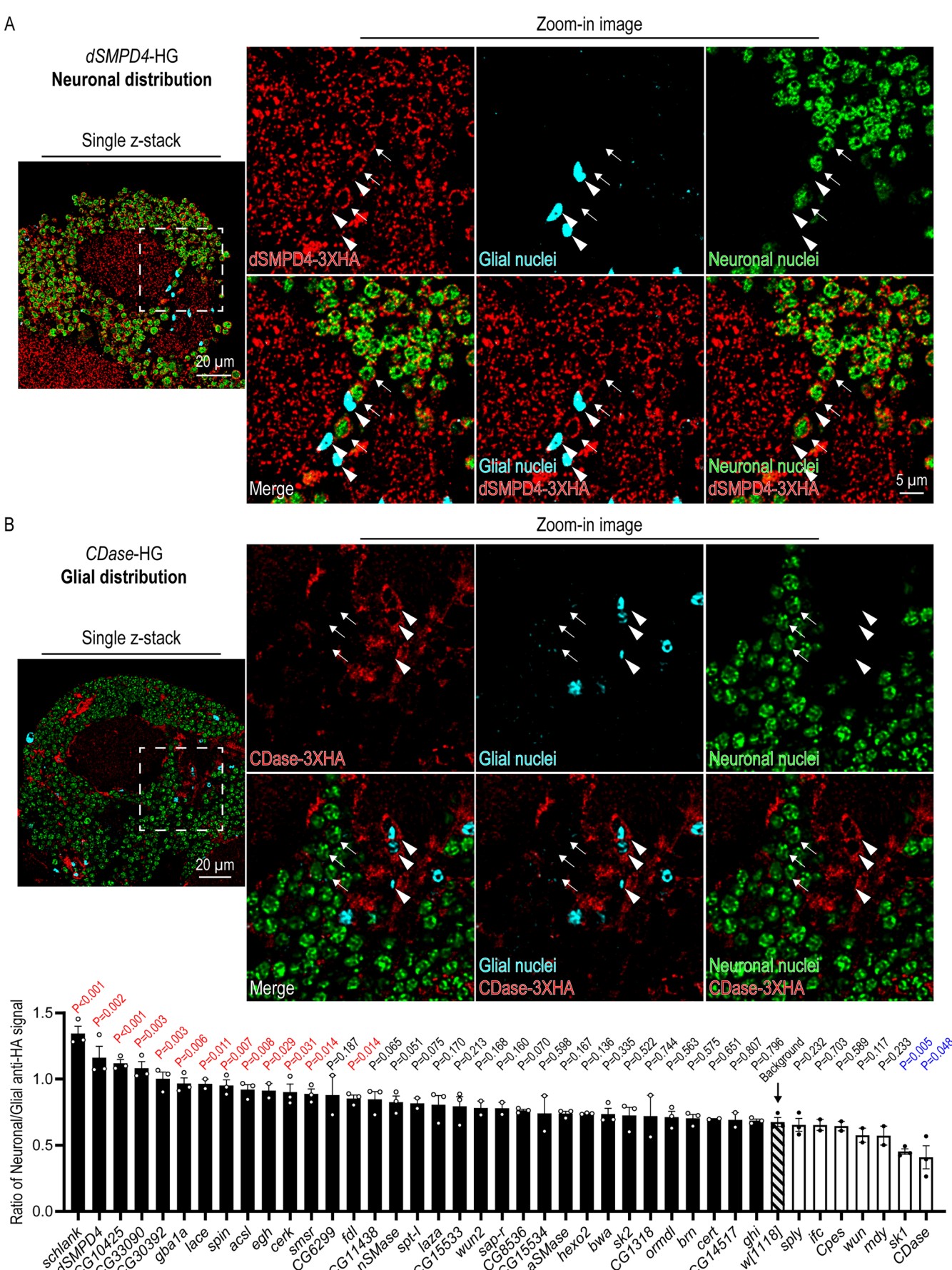

**Figure 4.  Translational profiling with HG revealed the protein distribution of SPL regulators in specific cell types in the brain.**

(A) Neuronal distribution of dSMPD4-3XHA protein. The protein distribution of dSMPD4-3XHA is visualized by anti-HA immunostaining (red) with co-stainings of neuronal (green; anti-Elav) and glial (cyan; anti-Repo) nuclei of young adult brains (1-week-old). (Left) The representative image is taken from the posterior view of the Calyx, and the dashed-line square indicates the region of zoom-in images shown on the right. (Right) Zoom-in images show anti-HA signals surrounding neuronal (arrow) and glial (arrowhead) nuclei. (B) Glial distribution of CDase-3XHA protein. The protein distribution of CDase-3XHA is visualized by anti-HA immunostaining (red) with co-stainings of neuronal (green; anti-Elav) and glial (cyan; anti-Repo) nuclei of young adult brains (1-week-old). (Left) The representative image is taken from the posterior view of the Calyx, and the dashed-line square indicates the region of zoom-in images shown on the right. (Right) Zoom-in images show anti-HA signals surrounding neuronal (arrow) and glial (arrowhead) nuclei. (C) Quantification and statistical analyses of anti-HA signals in neurons and glia. Confocal images were captured from the posterior view of the Calyx using whole-mount brains of young adult (1-week-old) HG lines and the $w^{1118}$ control. The mean intensity of the anti-HA channel surrounding neuronal (anti-Elav) and glial (anti-Repo) nuclei from 51 confocal images (1 μm interval; 50 μm in total) were measured as neuronal and glial anti-HA signals, respectively. The bar graph shows the ratio of neuronal to glial anti-HA signals. Data are represented as mean ± SEM ($n \geq 2$, biological repeats). $P$ values were calculated using two-tailed unpaired Student's $t$ test (Schlank, $P < 0.001$; dSMPD4, $P = 0.002$; CG10425, $P < 0.001$; CG33090, $P = 0.003$; CG30392, $P = 0.003$; gba1a, $P = 0.006$; lace, $P = 0.011$; spin, $P = 0.007$; acsl, $P = 0.008$; egh, $P = 0.029$; cerk, $P = 0.031$; smsr, $P = 0.014$; CG6299, $P = 0.187$; fdl, $P = 0.014$; CG11438, $P = 0.065$; nSMase, $P = 0.051$; spt-I, $P = 0.075$; laza, $P = 0.170$; CG15533, $P = 0.213$; wun2, $P = 0.168$; sap-r, $P = 0.160$; CG8536, $P = 0.070$; CG15534, $P = 0.598$; aSMase, $P = 0.167$; hexo2, $P = 0.136$; bwa, $P = 0.335$; sk2, $P = 0.522$; CG1318, $P = 0.744$; ORMDL, $P = 0.563$; brn, $P = 0.575$; cert, $P = 0.651$; CG14517, $P = 0.807$; ghi, $P = 0.796$; sply, $P = 0.232$; ifc, $P = 0.703$; Cpes, $P = 0.589$; wun, $P = 0.117$; mdy, $P = 0.233$; sk1, $P = 0.005$; CDase, $P = 0.048$) comparing the means of the ratio of neuron-to-glia anti-HA signals between HG lines and in the $w^{1118}$ background. Representative figures for anti-HA immunostaining from each HG line were presented in Appendix Fig. S32–47. Source data are available online for this figure.

puncta in the photoreceptors (Fig. 6F, arrowhead). Conversely, when using the photoreceptor-specific rh1-GAL4, most OlyA^w signals were observed in photoreceptors with some puncta in the interommatidial space (Fig. 6G, arrowhead). In the adult brain, OlyA^w labels the glial nuclei and membrane processes using a glial driver, while it labels neurons in the cortex when using the neuronal driver (Fig. EV5F). These results showed that OlyA^w predominantly labels the expressing cells, even though it may potentially be secreted.

We next examined whether OlyA^w binds to membrane rafts or membrane domains enriched with SM/CerPE and cholesterol. The OlyA^w puncta colocalized with the raft markers flo2-RFP and mCherry-GPI (Fig. 6H–J) but not the non-raft marker mCherry-CAAX (Appendix Fig. S51A). Furthermore, using acceptor photobleaching fluorescence resonance energy transfer (AP-FRET) (Appendix Fig. S51B), we photobleached flo2-RFP and observed that the intensity of OlyA^w was elevated, indicating close proximity to flo2 (Appendix Fig. S51C). In contrast, the OlyA^w intensity did not change upon mCherry bleaching in mCherry-CAAX or non-acceptor control backgrounds (Appendix Fig. S51C). These results validated that OlyA^w puncta labels lipid raft in the brain.

Finally, we tested the possible application of OlyA^w in live imaging using ex vivo adult brains. The puncta of OlyA^w shifted, emerged, or disappeared over time in the brain, which may suggest changes in membrane raft composition, movement, or (dis) assembly (Appendix Fig. S51D). Interestingly, we observed that depolarization induced by high extracellular potassium further enhanced the dynamics of OlyA^w signals (Appendix Fig. S51E,F), suggesting neuronal activity regulates membrane raft dynamics.

These findings support that OlyA^w can be used to indicate cellular CerPE levels in a cell-type-specific manner. Moreover, despite diffuse and potentially non-specific signals within the cell, the punctate OlyA^w signals reveal the dynamics of CerPE-enriched membrane rafts.

## SM/CerPE homeostasis is maintained by neuronal neutral SMase and glial acidic SMase

We next examined CerPE catabolism in neurons and glia by combining OlyA^w and cell-type-specific RNAi knockdowns of different SMases. Neuronal knockdowns of dSMPD4 using two independent RNAi lines led to a significant increase in the intensity of neuronal OlyA^w (Fig. 7A), while glial knockdown caused no significant changes (Fig. 7B). In contrast, RNAi knockdown of aSMase in glia significantly elevated the intensity of neuronally expressed OlyA^w (Fig. 7B), and the intensity of glial-derived OlyA^w was also significantly increased (Fig. EV6A). Indeed, lipidomic analysis confirmed a significant increase in total CerPE level in the brain upon glia-specific knockdown of aSMase (Fig. EV6B). Notably, nSMase knockdown in either neurons or glia was insufficient to alter OlyA^w signals (Fig. 7A,B), likely due to low knockdown efficiency or redundancy between the cell types (Appendix Fig. S52). These findings demonstrated the cell-type-specific function of SMases and revealed a cell nonautonomous catabolic mechanism that depends on glial aSMase.

## dSMPD4 is required for the maintenance of the nuclear membrane and affects brain functions

We next generated CRISPR/Cas9 knockouts of neutral SMases to elucidate their physiological function (Appendix Fig. S53). Lipidomic analysis of adult brains revealed that overall levels of total lipids, phospholipids, and SPL did not change significantly in dSMPD4^KO brains, whereas nSMase^KO brains strongly increased CerPE and decreased ceramides (Fig. 7C). At the level of the individual lipid species, dSMPD4^KO selectively increased brain d14:1/18:0, d14:1/20:0, d14:2/18:0, d14:2/18:1, and d16:1/18:0 CerPEs (Fig. 7D; Appendix Fig. S54A,B). In contrast, nSMase^KO broadly elevated many CerPE species regardless of acyl chain length (Fig. 7E; Appendix Fig. S54A,B). Downstream of altered CerPE metabolism in dSMPD4^KO brains, complex changes in ceramide levels occurred (Appendix Fig. S54C,D). Principal component analysis separated lipidomic profiles of dSMPD4^KO brains from that of controls for ceramide, CerPE, phosphatidylcholine, phosphatidylserine, and phosphatidylinositol, but not glucosylceramide and the major fly phospholipid phosphatidylethanolamine, indicating both specific alterations in the CerPE pathway and broader effects on subsets of the brain phospholipidome upon dSMPD4 deletion (Appendix Fig. S54E,F).

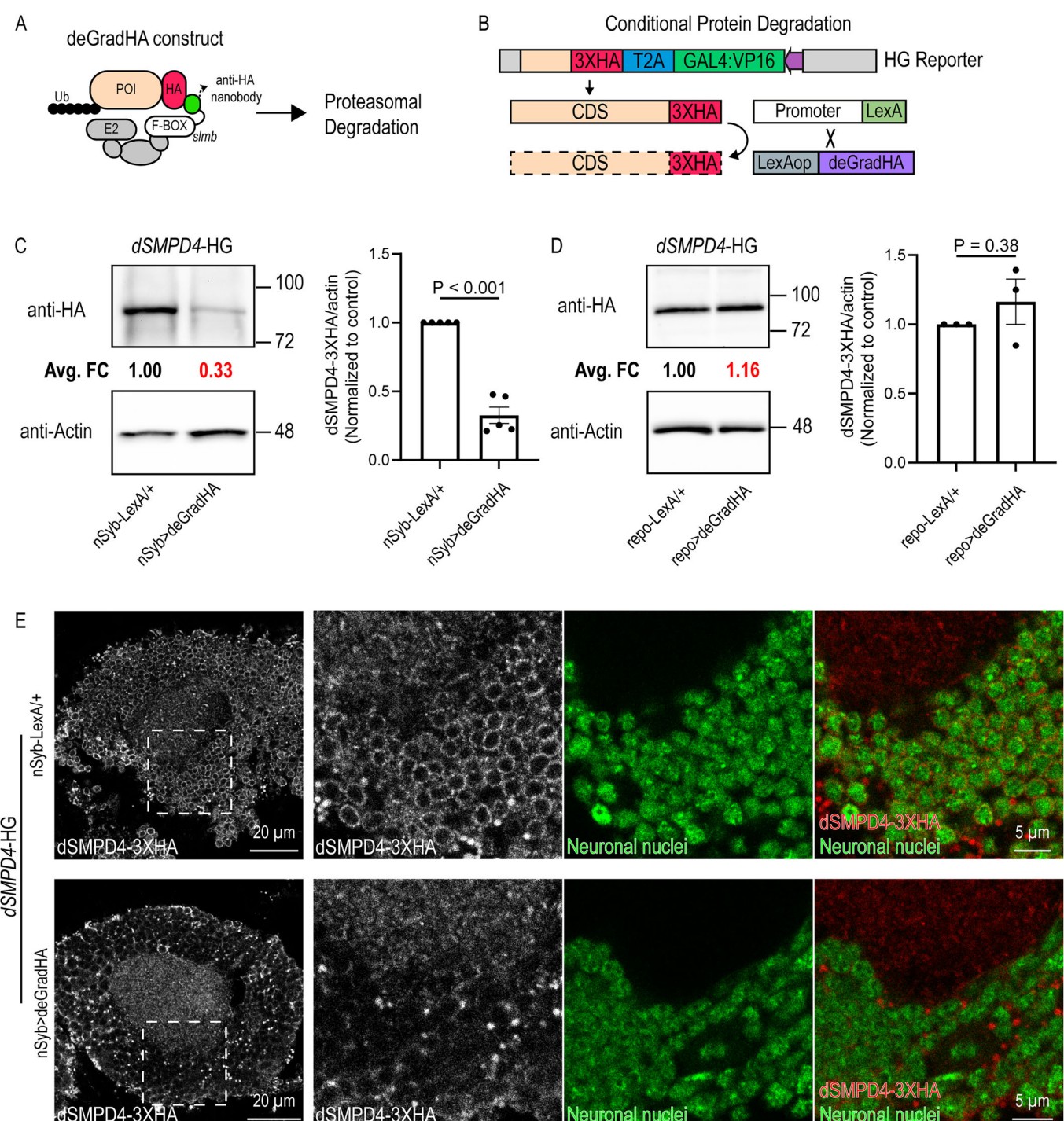

We further characterize the subcellular distribution of SMases in the brain by co-staining 3XHA-tagged proteins with organelle markers in HG lines. We found that nSMase localized to the mitochondria, aSMase was partially associated with lysosomes (Appendix Fig. S55), and dSMPD4 was present on the nuclear membrane (Fig. 7F,G). We noticed that *dSMPD4^{KO}* neurons showed blebbing of the nuclear membrane that was fully rescued by dSMPD4::myc overexpression (Fig. 7H,I). In contrast, knockout of *nSMase* did not cause the nuclear blebbing phenotypes (Fig. 7H,I) despite causing a stronger accumulation of CerPE (Fig. 7C). These

data indicated an organelle-specific function of dSMPD4 in maintaining the integrity of the nuclear membrane. We next used the deGradHA system to distinguish the cell type where dSMPD4 protein functions. Neuronal degradation of dSMPD4-3XHA showed nuclear blebbing, while glial degradation had no such effect (Appendix Fig. S56A), highlighting a neuronal requirement for dSMPD4.

We further examined whether loss of *dSMPD4* and *nSMase* affects brain functions. *dSMPD4* is highly expressed in the mushroom body and gustatory receptor neurons (Fig. EV2A),

**Figure 5. The deGradHA system enables tissue-specific degradation of 3XHA-tagged protein in HG lines.**

(A) Schematic illustration of the deGradHA system. The deGradHA is composed of a single-domain anti-HA nanobody fused to an F-Box motif derived from the *Drosophila slmb* gene. The binding of deGradHA to HA-tagged protein leads to degradation via the ubiquitin-proteasome system. (B) The LexAop-deGradHA enables conditional degradation of the 3XHA tagged in HG lines using tissue-specific LexA drivers. (C) Representative immunoblot of the adult brain extract of *dSMPD4*-HG line with (*nSyb*-LexA/+; *dSMPD4*-HG/*LexAop-deGradHA*) or without (*nSyb*-LexA/+; *dSMPD4*-HG/+) neuronal expression of deGradHA. The average fold change (Avg. FC) of dSMPD4-3XHA immunoreactivity normalized to loading control (anti-Actin) is shown in the figure. (Right) Quantification of the deGradHA protein degradation efficiency. Data are represented as mean ± SEM of 5 independent experiments of biological repeats. The *P* value ($P < 0.001$) was calculated using two-tailed unpaired Student's *t* test. (D) Representative immunoblot of the adult brain extract of *dSMPD4*-HG line with (*Repo*-LexA/+ or y;; *dSMPD4*-HG/*LexAop-deGradHA*) or without (*Repo*-LexA/+ or y;; *dSMPD4*-HG/+) glial expression of deGradHA. (Right) Quantification and statistical analysis of the deGradHA protein degradation efficiency. Data are represented as mean ± SEM of 3 independent experiments of biological repeats. The *P* value ($P = 0.380$) was calculated using two-tailed unpaired Student's *t* test. (E) Neuron-specific degradation of dSMPD4-3XHA protein in the adult brain. The protein distribution of dSMPD4-3XHA is visualized by anti-HA immunostaining with co-stainings of neuronal nuclei (green; anti-Elav) of young adult brains (1-week-old). (Left) The representative image is taken from the posterior view of the Calyx, and the dashed-line square indicates the region of zoom-in images shown on the right. (Right) Zoom-in images show anti-HA signals (gray in the single-channel image; red in the overlay image) surrounding neuronal nuclei (green). Source data are available online for this figure.

which serve as the center for associative learning and taste reception, respectively. *nSMase* exhibited a broader expression, including MB, GRNs, and other neuropils (Fig. EV2A). Olfactory associative learning assays revealed learning deficits in 1-week-old *dSMPD4^KO* flies, whereas *nSMase^KO* flies did not show learning impairment (Fig. 7J). Indeed, a reduced number of the mushroom body neurons can be observed in 1-week-old *dSMPD4^KO* flies (Appendix Fig. S56B). In addition, we observed abnormal feeding behaviors in both *dSMPD4^KO* and *nSMase^KO* flies at 1 week of age (Appendix Fig. S56C). These findings indicated a correlation between the brain expression pattern of *dSMPD4* and its roles in learning and feeding behaviors.

Next, since *aSMase*-knockout flies are not viable in the adult stage (Hull et al, 2024), we performed conditional knockout and tested the cell-type-specific requirement of *aSMase* by expressing aSMase-sgRNA and Cas9 in neurons versus glia. We investigated whether ref(2)P/p62, an autophagy adapter, was affected, as p62 accumulation was reported following SPL accumulation in the contexts of lysosome storage diseases (Vaughen et al, 2022; Kinghorn et al, 2016; Davis et al, 2016). We showed that glial knockout of the *aSMase* led to p62 aggregates, while neuronal knockout had no such effect (Fig. EV6C,D). Moreover, glial but not neuronal knockout of *aSMase* caused locomotor defects (Fig. EV6E). These findings suggested that *aSMase* is required in glia to maintain p62 proteostasis and neuronal functions.

## Discussion

We developed a genetic platform, the HG toolkit, for the systemic investigation of SPL metabolism in a living organism. This resource facilitates the transcriptional and translational profiling of the SPL regulatory network. The platform also enables subsequent functional analyses of candidate enzymes by combining powerful tools, including deGradHA for in situ protein depletion and the OlyA^w biosensor for in vivo SPL visualization.

The HG toolkit has several advantages over existing methods and resources. We engineered GAL4 activity to be controlled by the complete *cis*-regulatory element. Consequently, any subsequent investigation using the GAL4/UAS system will occur under the endogenous control of the target gene. Because of this design, the set of 3xHA-T2A-GAL4 knock-in animals enables spatiotemporal profiling of protein localization without the need to raise antibodies. This imaging-based method allowed most SPL regulators to be directly compared in terms of their expression patterns and protein distributions (Figs. 3 and 4).

Next, the integration of deGradHA with the LexA/LexAop system enhanced the versatility for genetic manipulation (Fig. 5). By combining deGradHA with cell-type-specific promoters, we can selectively degrade the target protein in specific cell types and observe its impact on distant tissues, allowing investigation of nonautonomous functions of secreted proteins. Moreover, deGradHA directly targets the protein of interest, overcoming common limitations of conditional gene knockout/knockdown strategies, such as discrepancies between gene activity versus protein level, and inefficient depletion of proteins with long half-lives. Hence, the deGradHA complements existing methods of genetic manipulation, especially when studying the spatiotemporal functions of proteins.

The genetically encoded OlyA^w is a powerful tool for the quantification of SM/CerPE levels and the real-time visualization of CerPE-enriched microdomains in vivo (Fig. 6). With OlyA^w, changes in CerPE levels of specific cell types within living tissues are easily resolved. The versatility of OlyA^w allows for mechanistic investigations of SPL regulation at cellular and subcellular levels in vivo, complementing lipidomic approaches.

Finally, the fruit fly, with it versatile genetic tools, is an attractive model for studying numerous diseases. Our platform serves as an introductory toolkit that is applicable for physiological observations of SPL regulation in all tissues and time points. It can also be easily adapted for the modeling of human diseases through simple genetic crosses. Hence, this resource provides a valuable platform for the comprehensive understanding of SPL metabolism in a living organism.

## Cell nonautonomous regulation of SPL metabolism

Our systemic profiling revealed heterogeneous expression patterns of SPL regulators in the brain, showing regional enrichment and cell-type-specific expression in different branches of SPL metabolism. It implies that the brain functions as a coherent unit by coordinating various cells and regions to execute specific steps of SPL metabolism, rather than each cell maintaining SPL homeostasis independently. As a proof-of-concept application, we showed that neurons and glia express different SMases to degrade distinct subcellular pools of CerPE. *dSMPD4* is primarily expressed and required in neurons. In contrast, the lysosomal acidic SMase (encoded by *aSMase*) is almost exclusively expressed in glia.

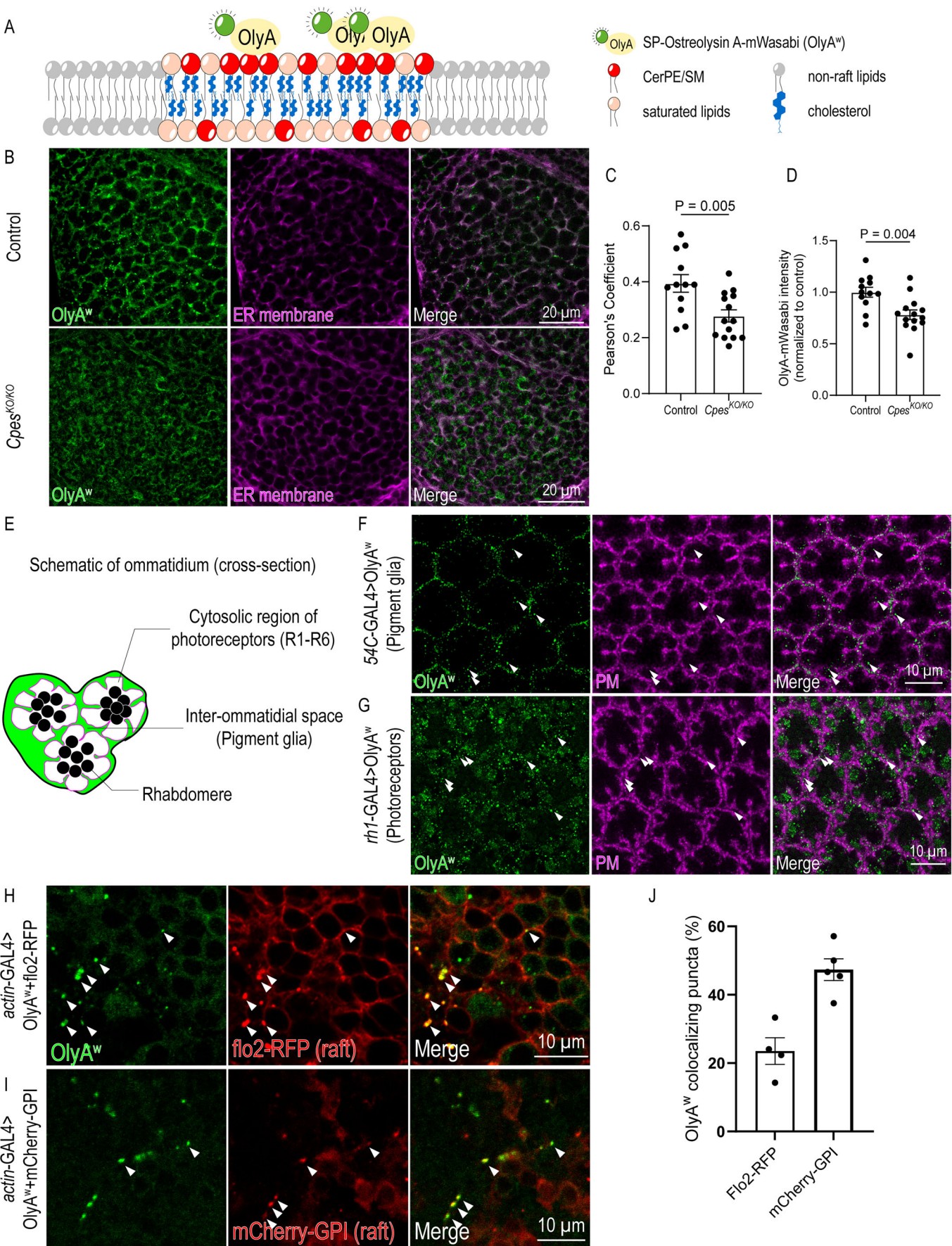

**Figure 6. OlyA^w, a genetically encoded SPL biosensor derived from ostreolysin A, indicates CerPE levels and labels membrane rafts.**

(A) A schematic figure of the OlyA^w biosensor for labeling SPL- and sterol-rich membrane rafts. (B) OlyA^w expression in the L3 salivary glands of control (*tubulin*-GAL4/ *UAS-OlyA^w*) and *Cpes^KO* (*Cpes^KO/Cpes^KO; tubulin*-GAL4/*UAS-OlyA^w*). Cnx99A is a marker of the ER membrane (magenta). (C) Quantification of Pearson's coefficient of OlyA^w and Cnx99A (ER membrane) channels. Data are represented as mean ± SEM of 3 independent experiments. Data points represent biological repeats. The *P* value (*P* = 0.005) was calculated using two-tailed unpaired Student's *t* test. (D) Quantification of mean intensities of the OlyA^w channel colocalizing with the ER membrane. Data are represented as mean ± SEM of 3 independent experiments. Data points represent biological repeats. The *P* value (*P* = 0.004) was calculated using two-tailed unpaired Student's *t* test. (E) A schematic of an ommatidium cross-section. Pigment glia occupy the interommatidial space. (F) OlyA^w expression driven by a pigment glia driver (*54C*-GAL4) in the adult eye. Na^+-K^+-ATPase is the marker of the plasma membrane (PM; magenta). OlyA^w puncta can be found in the cytosol of photoreceptors (arrowhead). (G) OlyA^w expression driven by a photoreceptor driver (*rh1*-GAL4) in the adult eye. Na^+-K^+-ATPase is the marker of the plasma membrane (PM; magenta). OlyA^w puncta can be found in the interommatidial space (arrowhead). (H) Confocal images of the adult brain for showing colocalization OlyA^w with a membrane raft marker, flo2-RFP. OlyA^w was co-expressed with flo2-RFP (*Actin*-GAL4/ +;*UAS-OlyA^w* /*UAS-flo2-RFP*). Arrowheads indicate OlyA^w puncta colocalized with flo2-RFP. (I) Confocal images of the adult brain for showing colocalization OlyA^w with a membrane raft marker, mCherry-GPI. OlyA^w was co-expressed with mCherry-GPI (*Actin*-GAL4/ *UAS-HA-mCherry-GPI;UAS-OlyA^w* /+). Arrowheads indicate OlyA^w puncta colocalized with mCherry-GPI. (J) Quantification of the percentage of flo2-GFP and mCherry-GPI labeled by OlyA^w. Data are represented as mean ± SEM. Data points represent biological repeats. Source data are available online for this figure.

Interestingly, the glial *aSMase* regulates CerPE levels in both neurons and glia. How aSMase in the glial lysosome regulates CerPE homeostasis in neurons is an intriguing question that requires further investigation.

The neural-glial transport of glucosylceramide impacts neuronal proteostasis and structural plasticity (Wang et al, 2022b; Vaughen et al, 2022). It is possible that a similar transport pathway exists for CerPE/SM. While in vivo evidence for CerPE/SM transport between neurons and glia remains elusive, in vitro studies have shown that neurons secrete SM (Chigorno et al, 2006), whereas glial cells uptake exogenous SM via endocytosis and primarily degrade it within lysosomes (Riboni et al, 1994). Notably, SM bound to lipoproteins is present in the cerebrospinal fluid (CSF) (Pitas et al, 1987), and CSF SM levels are reduced in APOE4 carriers (den Hoedt et al, 2023), suggesting a potential role for lipoproteins in brain SM transport. This aligns with the reported role of lipoproteins in SM transport in other tissues, such as the digestive tracts and plasma (Nilsson and Duan, 2006), where apoB/E lipoprotein receptor mediates endocytosis to facilitate SM uptake and lysosomal degradation (Levade et al, 1991). Future studies are warranted to interrogate whether lipoproteins facilitate the transport of CerPE/SM between neurons and glia in the brain.

Alongside the intercellular transport of SPLs, the cell non-autonomous function of enzymes may also play a role in maintaining SPL homeostasis. For instance, although the transcripts for the genes *gba1a*, *CG15533*, and *CG15534* were not detected in the brain, their protein products were readily detectable (Fig. EV3). This suggests the transfer of these enzymes from the midgut, potentially coupling dietary SPL uptake and metabolism in the brain. This hypothesis is consistent with the presence of predicted signaling peptides in these proteins, which is also the case for ceramidase (Appendix Fig. S48), a SPL regulator demonstrated to function nonautonomously across tissues (Acharya et al, 2008). Considering the success of enzyme replacement therapies for lysosomal storage disorders, where exogenous enzymes are taken up by cells (Del Grosso et al, 2022), intercellular enzyme transfer for SPL metabolism under physiological contexts may also be expected. Furthermore, we consistently observed a broader distribution of anti-HA signals compared to fluorescent markers driven by HG, which raises the intriguing possibility of intercellular transport of SPL-regulating enzymes within the brain. Evidently, the cause of this phenomenon is likely gene-dependent, and the discrepancy may also be attributed to other factors. For one, the

differential stability of 3XHA-tagged proteins and fluorescent markers might lead to a temporal mismatch in signal detection. Or, low levels of transcription in certain cells may result in detectable protein expression, as previous studies also observed the discrepancy between the levels of cognate protein and mRNA molecules (Buccitelli and Selbach, 2020). Altogether, while these findings could reveal novel insights into the complex regulation of SPL pathways in the brain, further investigation is needed to elucidate the underlying mechanisms.

## dSMPD4 maintains the integrity of the nuclear membrane in neurons and regulates brain functions

Loss-of-function *SMPD4* variants are linked to microcephaly in humans (Magini et al, 2019; Smits et al, 2023). A recent report shows that *SMPD4* RNAi disrupted nuclear envelope dynamics in HEK293T cells and impaired the proliferation of neural progenitors in the mouse cortex (Smits et al, 2023). Despite these insights, the neuronal role of SMPD4 and its molecular functions remain elusive. Utilizing the HG toolkit, we characterized the function of the fly ortholog *CG6962/dSMPD4* in regulating nuclear membrane integrity, neuronal survival, and brain functions. We observed a striking specificity in SMPD4 localization to nuclear membranes in subsets of neurons involved in associative memory formation, and olfactory learning defects in dSMPD4 mutants. Besides cell biological and neurophysiological studies, our lipidomics analysis revealed an increase in C18-CerPE species in *dSMPD4^KO* brains (Fig. 7D). Multiple mechanisms could account for the preferential increase of C18 fatty-acyl chains in *dSMPD4^KO*: first, dSMPD4 may preferentially process C18 fatty acids; second, dSMPD4-expressing neurons may be enriched for C18 fatty acids; or third, the nuclear membrane environment where dSMPD4 functions may contain high levels of C18-CerPE. As an example of the application of this genetic toolkit, our findings with *dSMPD4^KO* flies offer insights into how SPL metabolism orchestrated by the SMases contributes to brain functions, shedding light on the mechanisms underlying *SMPD4*-associated microcephaly.

## Limitations of the study

While leveraging the powerful genetic tools of *Drosophila* for studying SPL metabolism, it is important to acknowledge the difference in the complexity of the metabolic network between flies

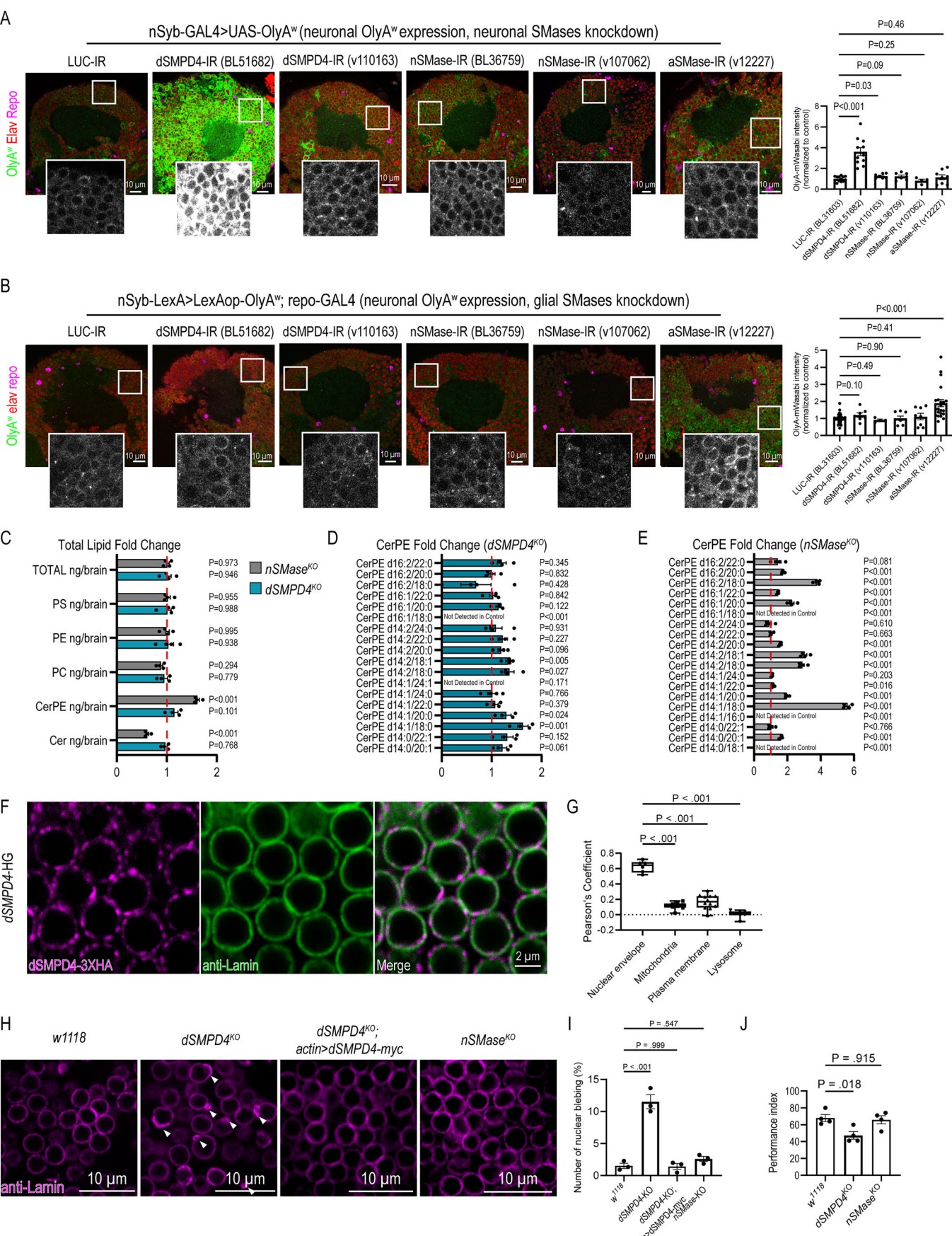

**Figure 7. dSMPD4 and aSMase are required in neurons and glia, respectively, to regulate CerPE catabolism and to maintain brain functions.**

(A) OlyA$^w$ expression and SMases RNAi knockdowns in neurons. UAS-OlyA$^w$ expression and SMases RNAi were driven by nSyb-GAL4 (nSyb-GAL4>OlyA$^w$ + UAS-RNAi). Confocal images were captured from the posterior view of the Calyx in the brain of young adult flies (1 week old). (Right) Quantitative analysis of OlyA$^w$ intensity in the cortical region. P values [dSMPD4-IR (BDSC 51682), $P < 0.001$; dSMPD4-IR (VDRC 110163), $P = 0.03$; nSMase-IR (BDSC 36759), $P = 0.09$; nSMase-IR (VDRC 107062), $P = 0.25$; aSMase-IR (VDRC 12227), $P = 0.46$] were calculated using one-way ANOVA with Dunnett's multiple comparisons. Data are represented as mean ± SEM ($n \geq 4$, biological repeats). Data are representative of at least two independent experiments. (B) OlyA$^w$ expression in neurons and SMases RNAi knockdowns in glia. LexAop-OlyA$^w$ expression is driven by nSyb-LexA, while SMases RNAi were driven by repo-GAL4 (nSyb-LexA>LexAop-OlyA$^w$; repo-GAL4 > UAS-RNAi). Confocal images were captured from the posterior view of the Calyx in the adult brain of young adult flies (1 week old). (Right) Quantifications of OlyA$^w$ intensity in the cortical region. P values [dSMPD4-IR (BDSC 51682), $P = 0.10$; dSMPD4-IR (VDRC 110163), $P = 0.49$; nSMase-IR (BDSC 36759), $P = 0.90$; nSMase-IR (VDRC 107062), $P = 0.41$; aSMase-IR (VDRC 12227), $P < 0.001$] were calculated using one-way ANOVA with Tukey's multiple comparisons. Data are represented as mean ± SEM ($n \geq 4$, biological repeats). Data are representative of at least two independent experiments. (C) Quantification of the fold change in total lipid levels of dSMPD4$^{KO}$ and nSMase$^{KO}$ compared to w$^{1118}$ brains (day 3 females). P values (nSMase$^{KO}$ vs. w$^{1118}$: Total lipids, $P = 0.973$; PS, $P = 0.955$; PE, $P = 0.995$; PC, $P = 0.294$; CerPE, $P < 0.001$; Cer, $P < 0.001$. dSMPD4$^{KO}$ vs. w$^{1118}$: Total lipids, $P = 0.946$; PS, $P = 0.988$; PE, $P = 0.938$; PC, $P = 0.779$; CerPE, $P = 0.101$; Cer, $P = 0.768$.) were calculated using one-way ANOVA with Dunnett's multiple comparisons. Data are represented as mean ± SEM ($n = 4$, biological repeats). (D) Quantification of the fold change in CerPE of dSMPD4-knockout compared to w$^{1118}$ brains (day 3 females). P values (d16:2/22:0, $P = 0.345$; d16:2/20:0, $P = 0.832$; d16:2/18:0, $P = 0.428$; d16:1/22:0, $P = 0.842$; d16:0/22:0, $P = 0.122$; d14:2/24:0, $P = 0.932$; d14:2/22:0, $P = 0.227$; d14:2/20:0, $P = 0.096$; d14:2/18:1, $P = 0.005$; d14:2/18:0, $P = 0.027$; d14:1/24:1, $P = 0.171$; d14:1/24:0, $P = 0.766$; d14:1/22:0, $P = 0.379$; d14:1/20:0, $P = 0.024$; d14:1/18:0, $P = 0.001$; d14:0/22:1, $P = 0.152$; d14:0/20:1, $P = 0.061$) were calculated using two-tailed unpaired Student's t test. Data are represented as mean ± SEM ($n = 4$, biological repeats). (E) Quantification of the fold change in CerPE of nSMase-knockout compared to w$^{1118}$ brains (day 3 females). P values (d16:2/22:0, $P = 0.081$; d16:2/20:0, $P < 0.001$; d16:2/18:0, $P < 0.001$; d16:1/22:0, $P < 0.001$; d16:1/20:0, $P < 0.001$; d16:1/18:0, $P < 0.001$; d14:2/24:0, $P = 0.610$; d14:2/22:0, $P = 0.663$; d14:2/20:0, $P < 0.001$; d14:2/18:1, $P < 0.001$; d14:2/18:0, $P < 0.001$; d14:1/24:0, $P = 0.203$; d14:1/22:0, $P = 0.016$; d14:1/20:0, $P < 0.001$; d14:1/18:0, $P < 0.001$; d14:1/16:0, $P < 0.001$; d14:0/22:1, $P = 0.001$; d14:0/20:1, $P < 0.001$; d14:0/18:1, $P < 0.001$) were calculated using two-tailed unpaired Student's t-test. Data are represented as mean ± SEM ($n = 4$, biological repeats). (F) Anti-HA and anti-Lamin co-staining of the adult brain from dSMPD4-HG line. Lamin is used as a marker of the nuclear envelope (NE). (G) Quantification of dSMPD4-3XHA subcellular localization. Box plot shows the Pearson's coefficient of anti-HA and organellar markers, including anti-Lamin (nuclear envelope), anti-ATP5A (mitochondria), anti-Na$^+$-K$^+$-ATPase (Plasma membrane), and anti-Cathepsin L (Lysosome). Box plots illustrate the data distribution. The center line within the box represents the median, which is the 50th percentile. The lower and upper bounds of the box correspond to the 25th and 75th percentiles, respectively. The ends of the whiskers indicate the minima and maxima. Data points represent biological repeats. P values (Nuclear envelope vs. Mitochondria, $P < 0.001$; Nuclear envelope vs. Plasma membrane, $P < 0.001$; Nuclear envelope vs. Lysosome, $P < 0.001$) were calculated using one-way ANOVA with Tukey's multiple comparisons. Representative images of anti-HA and organellar markers co-stainings are presented in Appendix Fig. S54. (H) Morphology of the nuclear membrane (anti-Lamin, magenta) in adult brains (1 week old) of indicated genotypes. (I) Quantification of the percentage of nuclei with blebs. Data are represented as mean ± SEM ($n = 3$, biological repeats). P values (w$^{1118}$ vs. dSMPD4$^{KO}$, $P < 0.001$; w$^{1118}$ vs. dSMPD4$^{KO}$; actin > dSMPD4-myc, $P = 0.999$; w$^{1118}$ vs. nSMase$^{KO}$, $P = 0.547$) were calculated using one-way ANOVA with Dunnett's multiple comparisons. Data are representative of at least two independent experiments. (J) Olfactory associative memory assay in dSMPD4-knockout, nSMase-knockout, and w$^{1118}$ control flies (1-week-old). Data are represented as mean ± SEM. P values (w$^{1118}$ vs. dSMPD4$^{KO}$, $P = 0.018$; w$^{1118}$ vs. nSMase$^{KO}$, $P = 0.915$) were calculated using one-way ANOVA with Dunnett's multiple comparisons. Data are representative of four independent experiments of biological repeats. Source data are available online for this figure.

and humans. One notable example is the glycosphingolipid pathway. While both mammals and flies depend on glycosphingolipids for essential biological processes, such as development and synaptogenesis (Chen et al, 2007; Huang et al, 2018), mammals exhibit a more intricate network with additional enzymes and more diverse glycosphingolipids. Although our findings in Drosophila may not always be directly extrapolated to humans, SPL metabolism in flies represents a simplified system for studying the regulatory networks in vivo with unprecedented network-wide scope and cell-type resolution.

Another important consideration is that our transcriptional profiling using the GAL4/UAS system may not fully recapitulate all aspects of the endogenous mRNA. For example, although the HG lines reveal temporal gene expression, the stability of fluorescent markers mCD8GFP and nls-mCherry driven by HG lines is independent of the mRNA transcript. Nevertheless, the binary system amplifies the signals for easier detection of endogenous expression and provides a convenient platform for genetic manipulation and functional investigation. Similar caution is warranted in our translational profiling due to known issues associated with HA tagging. We have chosen the HA tag because of its small size, but it may still affect the protein stability and behavior. While we employed multiple experimental approaches to ensure the reliability and relevance of the data from tagged proteins, it is prudent to perform comparative analyses between tagged and endogenous proteins or to conduct genetic rescue experiments for robust utilization of the toolkit in future studies.

While the OlyA$^w$ biosensor is useful to indicate cellular accumulations of CerPE and to reveal membrane raft dynamics, a key consideration when interpreting the OlyA$^w$ signal is the background fluorescence in the cortical ER, which was also observed in Cpes$^{KO}$ cells, where CerPE is mostly depleted. This suggested that a fraction of the biosensor may misfold and become trapped in the ER, resulting in a signal that is independent of CerPE's presence. Moreover, severe cellular phenotypes of Cpes mutants (Kunduri et al, 2018, 2022) may impair protein homeostasis of OlyA$^w$ itself, complicating the interpretation of how OlyA$^w$ reflects CerPE reduction and spatial distribution. Given both the OlyA$^w$ intrinsic localization in the ER and the confounding effects of the mutant background, its signal should not be used to infer the subcellular localization of CerPE without corroborating evidence.

In the OlyA$^w$ experiments for determining cell type-specific CerPE catabolic mechanisms (Fig. 7A,B), the varying efficacies between dSMPD4 RNAi lines and the lack of a second validation line for aSMase raise the possibility of off-target effects. Despite the inherent limitation of RNAi knockdown experiments, the proposed cell-specific functions are substantiated by multiple independent genetic approaches. The function of dSMPD4 in neurons is supported by its specific expression pattern (Figs. 3 and 4), the accumulation of distinct CerPE species in CRISPR/Cas9 knockouts (Fig. 7D), and a nuclear blebbing phenotype caused by targeted protein degradation (Fig. 5; Appendix Fig. S55A,B). Likewise, the function of aSMase in glia was validated by somatic CRISPR/Cas9

knockout, which replicated key pathological phenotypes, including Ref(2)P accumulation and locomotion defects (Fig. EV6C–E). In addition, brain lipidomic analysis revealed a significant increase in CerPE levels following glia-specific aSMase knockdown (Fig. EV6B), further confirming its essential role in glial CerPE metabolism.

# Methods

### Reagents and tools table

| Reagent/resource | Reference or source | Identifier or catalog number |
|---|---|---|
| **Experimental models:** Drosophila melanogaster **strains** | | |
| UAS-mCD8::GFP | Bloomington Drosophila Stock Center | 32184 |
| repo-GAL4 | Bloomington Drosophila Stock Center | 7415 |
| elav-GAL4 | Bloomington Drosophila Stock Center | 8765 |
| UAS-LUC-RNAi | Bloomington Drosophila Stock Center | 31603 |
| UAS-CDase-RNAi | Bloomington Drosophila Stock Center | 36764 |
| UAS-CG6962-RNAi | Bloomington Drosophila Stock Center | 51682 |
| UAS-CG6962-RNAi | Vienna Drosophila Resource Center | 110163 |
| UAS-nSMase-RNAi | Vienna Drosophila Resource Center | 107062 |
| UAS-nSMase-RNAi | Bloomington Drosophila Stock Center | 36759 |
| UAS-aSMase-RNAi | Vienna Drosophila Resource Center | 12227 |
| UAS-aSMase-RNAi | Bloomington Drosophila Stock Center | 36760 |
| UAS-aSMase-sgRNA | Bloomington Drosophila Stock Center | 77103 |
| UAS-Cas9 | Bloomington Drosophila Stock Center | 58986 |
| nsyb-LexA | Bloomington Drosophila Stock Center | 52817 |
| repo-LexA | Bloomington Drosophila Stock Center | 67096 |
| tubulin-GAL4 | Bloomington Drosophila Stock Center | 5138 |

| Reagent/resource | Reference or source | Identifier or catalog number |
|---|---|---|
| actin-GAL4 | Bloomington Drosophila Stock Center | 4414 |
| nSyb-GAL4 | Bloomington Drosophila Stock Center | 51635 |
| UAS-HA-mCherry-GPI | Bloomington Drosophila Stock Center | 94551 |
| UAS-mCherry.CAAX | Bloomington Drosophila Stock Center | 59021 |
| w[1118] | Bloomington Drosophila Stock Center | 3605 |
| 201Y-GAL4 | Bloomington Drosophila Stock Center | 4440 |
| UAS-mCherry.nls | Bloomington Drosophila Stock Center | 38424 |
| ifc CRIMIC-GAL4 | Bloomington Drosophila Stock Center | 92710 |
| CG3376 CRIMIC-GAL4 | Bloomington Drosophila Stock Center | 91346 |
| hs-Cre | Bloomington Drosophila Stock Center | 851 |
| hs-Cre | Bloomington Drosophila Stock Center | 766 |
| P[w + ,cre] | Bloomington Drosophila Stock Center | 1092 |
| UAS-OlyA-mWasabi | This paper | N/A |
| LexAop-OlyA-mWasabi | This paper | N/A |
| LexAop-deGradHA | This paper | N/A |
| UAS-dSMPD4-myc | This paper | N/A |
| UAS-Cpes | This paper | N/A |
| TI{TI}dSMPD4[1st-exon-dsRed] | This paper | N/A |
| TI{TI}nSMase[1st-exon-dsRed] | This paper | N/A |
| TI{TI}Cpes[1st-exon-dsRed] | This paper | N/A |
| TI{TI}lace[3XHA-T2A-GAL4:VP16] | This paper | N/A |
| TI{TI}spt-1[3XHA-T2A-GAL4:VP16] | This paper | N/A |
| TI{TI}CG10425/kdsr[3XHA-T2A-GAL4:VP16] | This paper | N/A |
| TI{TI}ORMDL[3XHA-T2A-GAL4:VP16] | This paper | N/A |
| TI{TI}bwa[3XHA-T2A-GAL4:VP16] | This paper | N/A |
| TI{TI}sk1[3XHA-T2A-GAL4:VP16] | This paper | N/A |
| TI{TI}sk2[3XHA-T2A-GAL4:VP16] | This paper | N/A |

| Reagent/resource | Reference or source | Identifier or catalog number |
|---|---|---|
| TI{TI}cerk[3XHA-T2A-GAL4:VP16] | This paper | N/A |
| TI{TI}wun[3XHA-T2A-GAL4:VP16] | This paper | N/A |
| TI{TI}wun2[3XHA-T2A-GAL4:VP16] | This paper | N/A |
| TI{TI}laza[3XHA-T2A-GAL4:VP16] | This paper | N/A |
| TI{TI}CG11425[3XHA-T2A-GAL4:VP16] | This paper | N/A |
| TI{TI}CG11426[3XHA-T2A-GAL4:VP16] | This paper | N/A |
| TI{TI}CG11437[3XHA-T2A-GAL4:VP16] | This paper | N/A |
| TI{TI}CG11438[3XHA-T2A-GAL4:VP16] | This paper | N/A |
| TI{TI}SMSr[3XHA-T2A-GAL4:VP16] | This paper | N/A |
| TI{TI}nSMase[3XHA-T2A-GAL4:VP16] | This paper | N/A |
| TI{TI}CG3376/aSMase[3XHA-T2A-GAL4:VP16] | This paper | N/A |
| TI{TI}CG15533[3XHA-T2A-GAL4:VP16] | This paper | N/A |
| TI{TI}CG15534[3XHA-T2A-GAL4:VP16] | This paper | N/A |
| TI{TI}CG6962[3XHA-T2A-GAL4:VP16] | This paper | N/A |
| TI{TI}GlcT[3XHA-T2A-GAL4:VP16] | This paper | N/A |
| TI{TI}egh[3XHA-T2A-GAL4:VP16] | This paper | N/A |
| TI{TI}brn[3XHA-T2A-GAL4:VP16] | This paper | N/A |
| TI{TI}Gal[3XHA-T2A-GAL4:VP16] | This paper | N/A |
| TI{TI}Ect3[3XHA-T2A-GAL4:VP16] | This paper | N/A |
| TI{TI}CG33090[3XHA-T2A-GAL4:VP16] | This paper | N/A |
| TI{TI}CG1318/Hexo1[3XHA-T2A-GAL4:VP16] | This paper | N/A |
| TI{TI}Hexo2[3XHA-T2A-GAL4:VP16] | This paper | N/A |
| TI{TI}fdl[3XHA-T2A-GAL4:VP16] | This paper | N/A |
| TI{TI}CG17223/α4GT1[3XHA-T2A-GAL4:VP16] | This paper | N/A |
| TI{TI}CG8536/β4GalNAcTA[3XHA-T2A-GAL4:VP16] | This paper | N/A |
| TI{TI}CG14517/β4GalNAcTB[3XHA-T2A-GAL4:VP16] | This paper | N/A |
| TI{TI}Gba1a[3XHA-T2A-GAL4:VP16] | This paper | N/A |
| TI{TI}Gba1b[3XHA-T2A-GAL4:VP16] | This paper | N/A |

| Reagent/resource | Reference or source | Identifier or catalog number |
|---|---|---|
| TI{TI}mdy[3XHA-T2A-GAL4:VP16] | This paper | N/A |
| TI{TI}Dgat2[3XHA-T2A-GAL4:VP16] | This paper | N/A |
| TI{TI}eloF[3XHA-T2A-GAL4:VP16] | This paper | N/A |
| TI{TI}CG6299[3XHA-T2A-GAL4:VP16] | This paper | N/A |
| TI{TI}CG30392[3XHA-T2A-GAL4:VP16] | This paper | N/A |
| TI{TI}ghi[3XHA-T2A-GAL4:VP16] | This paper | N/A |
| TI{TI}schlank[3XHA-T2A-GAL4:VP16] | This paper | N/A |
| TI{TI}ifc[3XHA-T2A-GAL4:VP16] | This paper | N/A |
| TI{TI}CDase[3XHA-T2A-GAL4:VP16] | This paper | N/A |
| TI{TI}sply[3XHA-T2A-GAL4:VP16] | This paper | N/A |
| TI{TI}cpes[3XHA-T2A-GAL4:VP16] | This paper | N/A |
| TI{TI}Acsl[3XHA-T2A-GAL4:VP16] | This paper | N/A |
| TI{TI}elovl7[3XHA-T2A-GAL4:VP16] | This paper | N/A |
| TI{TI}fa2h[3XHA-T2A-GAL4:VP16] | This paper | N/A |
| TI{TI}cert[3XHA-T2A-GAL4:VP16] | This paper | N/A |
| TI{TI}spin[3XHA-T2A-GAL4:VP16] | This paper | N/A |
| TI{TI}Sap-r [3XHA-T2A-GAL4:VP16] | This paper | N/A |
| **Antibodies** | | |
| Anti-DLG | Developmental Studies Hybridoma Bank | Cat#4F3 |
| Anti-HA | Cell Signaling Technology | Cat# 3724 |
| Anti-HA | Roche | Cat# 12158167001, |
| Anti-Actin | Merck Millipore | Cat# MAB1501 |
| Anti-Repo | Developmental Studies Hybridoma Bank | Cat# 8D12 |
| Anti-Elav | Developmental Studies Hybridoma Bank | Cat# Elav-9F8A9 |
| Anti-lamin | Developmental Studies Hybridoma Bank | Cat# ADL67.10 |
| Anti-ATP5A | Abcam | Cat# ab14748 |
| Anti-Cathepsin L | R and D Systems | Cat# MAB22591 |
| Anti-Na$^+$/K$^+$-ATPase | Developmental Studies Hybridoma Bank | Cat# a5 |

| Reagent/resource | Reference or source | Identifier or catalog number |
|---|---|---|
| Anti-Ref(2)P | Abcam | Cat# ab178440 |
| Anti-GM130 | Abcam | Cat# ab30637 |
| Anti-Golgin245 | Developmental Studies Hybridoma Bank | Cat# Golgin245 |
| Anti-Cnx99A | Developmental Studies Hybridoma Bank | Cat# Cnx99A 6-2-1 |
| Anti-GFP | Abcam | Cat# ab13970 |
| Anti-Calnexin | Abcam | Cat# ab75801 |
| Anti-Rab5 | Abcam | Cat# ab31261 |
| Anti-Ifc | Jung et al, 2017 | N/A |
| Alexa Fluor®-647 anti-Mouse IgG | Jackson ImmunoResearch Laboratories | Cat# 111-605-003 |
| Alexa Fluor®-488 anti-Mouse IgG | Jackson ImmunoResearch Laboratories | Cat# 715-545-150 |
| Alexa Fluor®-488 anti-Rabbit IgG | Jackson ImmunoResearch Laboratories | Cat# 711-545-152 |
| Alexa Fluor® 488 AffiniPure™ Goat Anti-Rat IgG (H + L) | Jackson ImmunoResearch Laboratories | Cat# 112-545-167 |
| Cy™3 AffiniPure™ Goat Anti-Rabbit IgG (H + L) | Jackson ImmunoResearch Laboratories | Cat# 111-165-003 |
| Alexa Fluor® 647 AffiniPure™ Goat Anti-Mouse IgG (H + L) | Jackson ImmunoResearch Laboratories | Cat# 115-605-166 |
| Peroxidase AffiniPure™ Goat Anti-Rat IgG (H + L) | Jackson ImmunoResearch Laboratories | Cat# 112-035-003 |
| Peroxidase AffiniPure™ Goat Anti-Mouse IgG (H + L) | Jackson ImmunoResearch Laboratories | Cat# 115-035-003 |
| Biotin-SP (long spacer) AffiniPure™ Goat Anti-Rabbit IgG (H + L) | Jackson ImmunoResearch Laboratories | Cat# 111-065-003 |
| Alexa Fluor® 647 Streptavidin | Jackson ImmunoResearch Laboratories | Cat# 016-600-084 |

## Fly stocks and genetics

*Drosophila* stocks and crosses were maintained at 25 °C on standard medium following standard fly husbandry. Information on individual fly strains can be found on FlyBase (flybase.org) unless otherwise noted.

## Generation of HG flies

All plasmid vectors were constructed by standard molecular biology procedures. The HG knock-in cassette contains the following components: the 3xHA coding sequence, the T2A sequence derived from *Thosea asigna* virus capsid protein, the GAL4:VP16 sequence with a stop codon, LoxP, hsp70 3'UTR, 3XP3RFP, and LoxP. To construct a HG donor vector for each targeted gene, about 1 kb sequences immediately upstream and downstream of the stop codon were PCR-amplified from the genomic DNA of the w[1118] strain. Two homology arms and the HG knock-in cassette were constructed in the pUC57-Kan vector. The endogenous stop codon will be replaced by the HG-3XP3RFP cassette such that HG was transcribed as an in-frame fusion with the target gene. The selection marker 3XP3RFP contains 3XP3 promoter, RFP, and alpha-Tub 3'UTR, and it facilitates genetic screening and can be flipped out by Cre recombinase. The HG knock-in flies were crossed to hs-Cre stocks, depending on the chromosome where the target gene was located. Because of the leaky expression, the hs-Cre without a heat shock was sufficient to excise the 3XP3RFP marker for all progeny, except for *schlank, egh, brn, Acsl, Elovl7, Sap-r, spin, CG11426*. The transcript is stabilized by *Hsp70Ba* 3'UTR before excision, and it is stabilized by endogenous 3'UTR after excision. Guide RNA (gRNA) vectors were constructed in the pBFv-U6.2 vector. We selected a 20-bp gRNA inside the left homology arm near the stop codon for each gene. gRNA were selected based on their efficiency score and specificity annotated at the UBSC genome browser (https://genome.ucsc.edu/). In order to avoid recurrent cleavage following integration, the homology arm of the donor vector was modified by incorporating silent mutations of the PAM site. Some targeted genes have alternative stop codons, and the targeted isoforms are listed in Table EV1.

## Transgenesis

The deGradHA was constructed in JFRC19-13XLexAop2-IVS-myr::GFP vectors. The deGradHA construct includes an F-Box domain of the Drosophila *slmb* gene and an anti-HA nanobody also known as Frankenbody (Zhao et al, 2019). The F-Box domain was PCR-amplified from the genomic DNA of the *w[*]; P{y[+t7.7] w[+mC]=exu-Nslmb-vhhGFP4}attP40* strain. The anti-HA nanobody is a chimeric of complementarity determining regions of the anti-HA 12CA5-single-chain variable fragments (scFVs) and a 15F11 scFv scaffold, and it was PCR-amplified from the genomic DNA of *w[1118]; UAS-FrankenbodyHA:GFP@attP40* (Murakawa et al, 2022).

The OlyA^w were constructed in pUAST.attB and pJFRC19-13XLexAop2-IVS-myr::GFP vectors for making UAS-OlyA^w and LexAop-OlyA^w transgenic flies, respectively. The OlyA^w construct is composed of a signaling peptide derived from enkephalin in Rat (MAQFLRLCIWLLALGSCLLATVQAD), the OlyA sequence derived from *Pleurotus ostreatus*, and a brighter GFP variant called mWasabi (Ai et al, 2008). Previous studies have shown that a signaling peptide is required for membrane labeling of sphingomyelin-binding probes because sphingomyelin is mainly located on the outer leaflet of the plasma membrane or in the Golgi lumen (Ono et al, 2022; Skočaj et al, 2014). The OlyA sequence was PCR-amplified from the pOlyA(WT) plasmid(Endapally et al, 2019). The mWasabi sequence was PCR-amplified from the mWasabi-C1 plasmid.

The UAS-dSMPD4-myc and the UAS-Cpes were constructed in the pUAST.attB vector. The coding sequence of dSMPD4 and Cpes was PCR-amplified from the cDNA of the *w[1118]* strain.

## CRISPR-knockout flies of *Cpes*, *dSMPD4*, and *nSMase*

The *Cpes*, *dSMPD4*, and *nSMase* were knocked out by replacing the first coding exon with a recombinase-mediated cassette exchange (RMCE) cassette containing two attP sites, a splice acceptor, and STOP codons followed by an SV40 polyadenylation signal and a 3xP3dsRed marker (Appendix Fig. S53A) (Zhang et al, 2014). Since *Cpes* has only one exon, the entire coding sequence of *Cpes* was removed in the *Cpes*-knockout flies. The 1 kb sequences immediately upstream and downstream of the cutting sites were PCR-amplified and constructed in the pJET1.2-STOP-dsRed vector. We selected 20-bp gRNAs upstream and downstream of the first coding exon of the target gene based on the efficiency score and specificity annotated at the UBSC genome browser (https://genome.ucsc.edu/).

The first and the second gRNAs were first constructed in the pBFv-U6.2 and pBFv-U6.2B vectors, respectively. For the construction of a vector expressing both gRNAs, the U6.2 promoter, the second gRNA, and the gRNA scaffold of the pBFv-U6.2B were subcloned into the pBFv-U6.2, consisting of the first gRNA.

## Embryonic microinjection

All transformants were obtained using a standard injection method by Wellgenetics in Taiwan (https://wellgenetics.com/). The injection strain for HG knock-in and gRNA vectors was either *w[1118];;P{nos-Cas9, y+, v+}3 A/TM6B*, Tb[1] or *w[1118];attP40{nos-Cas9}/CyO* depending on the chromosome where the target gene was located. The injection strain for the *dSMPD4*-knockout using RMCE knock-in and gRNA vectors was *w[1118];attP40{nos-Cas9}/CyO*. The pUAST.attB and pJFRC19-13XLexAop2-IVS-myr::GFP vectors of OlyA^W and deGradHA were injected in both *y[1] M{RFP[3xP3.PB]GFP[E.3xP3]=vas-int.Dm}ZH-2A w[*];P{y[+t7.7]=CaryP}attP40* and *y[1] M{RFP[3xP3.PB]GFP[E.3xP3]=vas-int.Dm}ZH-2A w[*];P{y[+t7.7]=CaryP}attP2* strains. The injection strain for the UAS- dSMPD4-myc vector was *y[1] M{RFP[3xP3.PB]GFP[E.3xP3]=vas-int.Dm}ZH-2A w[*];P{y[+t7.7]=CaryP}attP40*.

## Western blotting

For anti-HA immunoblots of the adult fly total lysates, 5 adult flies were homogenized in 200 μl T-PER™ Tissue Protein Extraction Reagent (Thermo, 78510) supplemented with protease inhibitors (Thermo, 88666) and centrifuged to collect the supernatant. The 3 volumes of protein extracts were mixed with 1 volume of 4X Laemmli sample buffer (Bio-Rad, 1610747) and incubated at 95 °C for 15 min. For anti-HA immunoblots of the adult brain extracts, ten adult brains were homogenized in 2× Laemmli sample buffer (Bio-Rad, 1610747) on ice and incubated 10 min at 95 °C. Denatured proteins were separated on 12% SDS-PAGE gels and transferred to a PVDF membrane (Millipore, IPVH00010) overnight at 4 °C. Membranes were blocked in 5% milk in TBST [10 mM Tris (pH 8.0), 150 mM NaCl, 0.1% Tween 20] for 1 h and probed with primary antibodies, anti-HA (1:1000; Roche Cat# 12158167001) and anti-Actin (1:10000; Millipore Cat# MAB1501), diluted in 2% milk in TBST at 4 °C overnight. Following three 10-minute washes with TBST, the membranes were incubated with HRP-conjugated secondary antibodies (1:5000; Jackson ImmunoResearch Labs 112-035-003 and 115-035-003), in 2% milk in TBST at

4 °C overnight. After washing the membrane three times with TBST, the protein signal was detected using ECL Western Blotting detection reagents (Millipore, WBKLS0500) and captured with the MultiGel-21 (TOPBIO, MGIS-21-C2-1M).

## Immunofluorescence and imaging

Adult fly brains were dissected in phosphate-buffered saline (PBS) fixed in 4% paraformaldehyde diluted in PBS at room temperature for 20 min. The fixed samples were washed three times for 10 min in PBS with 1% Triton-X-100 (1% PBST). The brains were permeabilized with 2% PBST for 30 min, then blocked in 0.25% PBST with 1% BSA for 1 h. Next, the samples were incubated with primary antibodies, rat anti-Elav (1:200, Rat-Elav-7E8A10, DSHB), mouse anti-Repo (1:200, 8D12 anti-Repo, DSHB), mouse anti-lamin (1:200, ADL67.10, DSHB), mouse anti-ATP5A (1:500, ab14748, Abcam), mouse anti-Cathepsin L (1:500, MAB22591, R and D Systems), mouse anti-Na⁺/K⁺-ATPase (1:200, a5, DSHB), rabbit anti-Ref2P (1:200, ab178440, abcam), mouse anti-DLG (1:200, 4F3 anti-discs large, DSHB), rabbit anti-GM130 (1:200, ab30637, Abcam), goat anti-Golgin245 (1:500, Golgin245, DSHB), mouse anti-Cnx99A (1:200, Cnx99A 6-2-1, DSHB), mouse anti-Golgin84 (1:200, Golgin84 12-1, DSHB), chicken anti-GFP (1:500, ab13970, Abcam), rabbit anti-Ifc (1:200, Jung et al, 2017) rabbit anti-Calnexin (1:200, ab75801, Abcam), rabbit anti-rab5 (1:200, ab31261, Abcam), diluted in 0.25% PBST with 1% BSA overnight at 4 °C, washed in 1% PBST four times for 30 min, and incubated in secondary antibody (1:500, 111-605-003, 715-545-150, 711-545-152, 112-545-167, 111-165-003, 115-605-166, Jackson ImmunoResearch Laboratories) diluted in 0.25% PBST with 1% BSA overnight at 4 °C. The brains were then washed with 1% PBST four times for 30 min and mounted in Vectashield (H-1000, Vector Laboratories) or RapiClear 1.47 (#RC147002, SunJin Lab) for subsequent confocal imaging. For anti-HA staining, permeabilized brains were blocked in 0.25% PBST with 1% BSA overnight at 4 °C. Next, incubated with preabsorbed rabbit anti-HA antibody (1:200, 3724, Cell Signaling) diluted in 0.25% PBST with 1% BSA overnight at 4 °C, washed in 1% PBST four times for 30 min, and incubated in the Biotin conjugated secondary antibody (1:200, 111-065-003, Jackson ImmunoResearch Laboratories) diluted in 0.25% PBST with 1% BSA overnight at 4 °C. The brains were then washed with 1% PBST four times for 30 min and incubated in Rhodamine Red™-X (RRX) Streptavidin (1:500, 016-290-084, Jackson ImmunoResearch Laboratories) diluted in 0.25% PBST with 1% BSA overnight at 4 °C. The brains were then washed with 1% PBST four times for 30 min and mounted. Images were captured with Zeiss LSM 880, Zeiss LSM780, or Leica Sp8 confocal microscopes.

For OlyA^w live imaging, adult brains were dissected in cold adult hemolymph-like saline (AHL, 108 mM NaCl, 5 mM trehalose, 10 mM sucrose, 5 mM KCl, 2 mM CaCl₂, 8.2 mM MgCl₂, 4 mM NaHCO₃, 1 mM NaH₂PO₄, 5 mM HEPES) and immediately mounted in AHL or AHL with 50 mM KCl (58 mM NaCl, 5 mM trehalose, 10 mM sucrose, 50 mM KCl, 2 mM CaCl₂, 8.2 mM MgCl₂, 4 mM NaHCO₃, 1 mM NaH₂PO₄, 5 mM HEPES) for ex vivo live imaging. Images were acquired using a Zeiss LSM 880 confocal microscope with a 63× NA 1.4 oil objective, and the time interval between frames was 500 ms. The imaging was completed within 5 min starting from the mounting of each brain.

To minimize photobleaching of the OlyA^w, the laser power was kept under 5% during image acquisition.

## 3D brain alignment and regional expression analyses

Confocal image stacks of selected genes exhibiting neuronal projections (*HG* > mCD8GFP) were registered to the reference brain (standard brain template) used in the FlyCircuit database (http://www.flycircuit.tw/). The reference brain includes the segmentation of 29 neuropils on each hemisphere using the anti-DLG channel (Chiang et al, 2011). To efficiently register the anti-DLG images to the reference brain, we use the Computational Morphometry Toolkit (CMTK) (Rohlfing and Maurer, 2003) to transform the images by affine transformation, including translation, rotation, and anisotropic scaling. We then warped the 3D images of mCD8GFP expression based on the resultant transformation matrices to quantify its intensity in each neuropil. For quantification, neuropils in the optic lobes, including LOP, LOP, MED, and OG, were not analyzed due to incomplete imaging. The warped 3D images were overlaid with 25 neuropils from both hemispheres of the reference brain. After the brain warping, the GFP intensity is normalized to the area of each neuropil, and the average intensity of mCD8GFP per voxel was recorded as raw values.

Data were visualized using R (version 4.3.1) in R Studio. To compare expression patterns across genes with different mCD8GFP intensities, both expression and correlation heatmaps were plotted based on the mCD8GFP intensity of the target gene in each neuropil, normalized to the mean of the target gene in all neuropils. Expression heatmap was generated with the pheatmap package (version 1.10.12). A correlation heatmap was generated with the corrplot package (version 0.92). "ward.d2" was used as the clustering method in both figures. RColorBrewer was used for color generation.

## nls-mCherry colocalization

Colocalization of nls-mCherry with Elav and Repo in whole-mount brains of 1-week-old adult flies was analyzed using the Imaris software (Bitplane, Switzerland). Since the expression of 3XP3RFP selection marker in optic lobes interfered with the nls-mCherry signals in pre-Cre HG lines, we excluded the optic lobes. A 3D surface was manually created in Imaris to delineate the boundaries of the central brain region. This surface was then used to generate a mask for all image channels. To improve image resolution, we performed the Imaris ClearView™ Deconvolution using the standard protocol with 10 iterations. Next, we performed the Spot module to detect spots of the deconvoluted nls-mCherry, Elav, and Repo channels. Spots were considered colocalized if they were within 2 μm of each other.

We analyzed a total of 182 adult brains from HG lines of different genotypes and revealed a 9:1 ratio of neurons to glia (89.56% neurons and 10.44% glia), consistent with insect brains constituting approximately 90% neurons (Raji and Potter, 2021). Under these circumstances, a gene with no enrichment in neurons should show approximately 90% of nls-mCherry colocalizing with Elav. Following this rule, the percentage of nls-mCherry colocalizing to Elav in the total number of nls-mCherry is compared to the percentage of Elav-positive cells in the total number of cells (the sum of Elav- and Repo-positive cells) for each HG allele.

## Quantification of anti-HA signals

Images were first batch-processed in ImageJ. Channels for anti-Elav and anti-Repo were used to establish the region of interest (ROI) for HA quantification. Stack of images for anti-Elav and anti-Repo staining were normalized by "Enhance Contrast…", so that 0.35% of pixels are saturated. Binary masks were then made through the "Make Binary" function with the "Otsu calculate black" method. In order to segment connected cells, "Watershed" with the parameter "shed" was applied. Small particles were then removed with an open operation by applying the "Minimum…" and "Maximum…" functions subsequently, both with a radius of 0.5 μm. The result is the "nucleus" mask. Each slice of the "nucleus" mask is then dilated with the "Maximum…" function with a radius of 3 pixels to obtain the "surrounding" mask. The final region of interest is then calculated by subtracting the "surrounding" mask with the "nucleus" mask through the "Calculator Plus" plug-in. Total pixel numbers and the mean HA intensity for the ROI of each slice were finally measured.

The measurements were pasted onto Excel and further calculated. The average HA intensity for a brain were calculated by dividing the sum of the total intensity of all slices, and then divided by the sum of pixels of all slices, namely $\sum(pixels\ within\ ROI\ per\ slice \times mean\ HA\ intensity\ within\ ROI\ per\ slice)/\sum(pixels\ within\ ROI\ per\ slice)$.

## Acceptor photobleaching fluorescence resonance energy transfer (AP-FRET)

AP-FRET was performed on a Zeiss LSM 880 confocal microscope. Images were acquired using a ×63 NA 1.4 oil objective with 5-second intervals. The power of the 488 nm laser was kept under 5% to minimize photobleaching during image acquisition. Between the third and the fourth time point, the 100% of 568 nm laser with 50 iterations was used for photobleaching the acceptor: flo2-RFP or mCherry-CAAX. The mean intensity of the OlyA^w in the region of interest was quantified at each time point using the Zen software.

## OlyA^w quantification in the adult brain

Confocal images of the posterior view of the Calyx from whole-mount adult Drosophila brain were acquired using a Leica Sp8 microscope with a ×63 objective. Fluorescence intensity quantification was performed using ImageJ software. The region of interest (ROI) corresponding to the calyx cortex was defined by creating a binary mask of the anti-Elav staining. This was achieved by applying a Gaussian blur (sigma = 2) to the Elav channel, followed by thresholding. The resulting mask was then overlaid onto the experimental fluorescence channel, and the mean fluorescence intensity within the masked cortical ROI was measured for each image.

## Quantitative real-time PCR

Adult flies were homogenized, and total RNA was extracted using the Quick-RNA Miniprep Kit (Zymo, R1055). Before cDNA

synthesis, the RNA concentration was measured, and normalization was performed to maintain equal concentration. Subsequently, reverse transcription was carried out using SuperScript III Reverse Transcriptase (Invitrogen, 18080093) using Oligo (dT)18 primer. qRT-PCR was performed using the 7300 Real-Time PCR System (Applied Biosystems) and the IQ2 SYBR Green Fast qPCR System Master Mix (Bio Genesis, BB-DBU-006). The average value was normalized to *rp49*.

## Generation of anti-dSMPD4 antibody

Polyclonal rabbit anti-dSMPD4 antibody was generated by Yao-Hong Biotechnology Inc. in Taiwan using a synthetic peptide QKGTEAVLPHSHYFHSG coupled to keyhole limpet hemocyanin (KLH) carrier. The anti-dSMPD4 IgGs were purified on an antigenic peptide-coupled affinity column and checked by ELISA for antigen reactivity.

## Lipidomics and lipid analyses

Lipidomic analysis was performed as previously described (Vaughen et al, 2022). Briefly, fly cohorts of $w^{1118}$ control, $nSMase^{KO}$, and $dSMPD4^{KO}$ mutant (mated female) brains were dissected at day 3 post eclosion. Four sets of 15 brains (omitting the retina) were rapidly dissected from each genotype, cleared of fat, and snap-frozen in 90% methanol before targeted analysis for sphingolipids (using 10 brains) and phospholipids (using 5 brains).

Brain total lipids containing internal standards (ISTD) were extracted as described previously (Tsai et al, 2019). Following ISTD were added: Glucosyl-beta-Ceramide d18:1/8:0 (Avanti #860540), CerPE d18:1/24:0 (Avanti #860067) and Equisplash™ LIPIDOMIX® Quantitative Mass Spec Internal Standard (Avanti #330731). The extracted lipids were then separated using normal phase chromatography (Agilent Zorbax RX-Sil column 3.0 ×100 mm, 1.8 μm particle size) using an Agilent 1260 Infinity LC system (Santa Clara, CA). Finally, an Agilent 6430 Triple-Quad LC/MS system (Santa Clara, CA) obtained MS measurements in positive ion mode. The MS used electrospray ionization (ESI) to generate gas-phase ions from different types of lipids, which were then analyzed and quantified by multiple reaction monitoring (MRM). The collision energies ranging from 15 to 45 eV were used. Data acquisition and analysis were performed using the Agilent Mass Hunter software package.

For quantification of sphingolipid species, we calculated the total quantity per brain in each sample (ng lipid species per brain per tube) as well as the relative percentage of each lipid. We focused on measuring the individual amounts (ng/brain) of different sphingolipids, but only obtained relative percentage values for phospholipid species. Importantly, the total phospholipids (ng/brain) remained the same between the mutant and control groups. We report raw values in Source Data Fig. 7; data were filtered for sphingolipids present at >0.01 ng/brain and phospholipids for species in top 99.5% of all detected phospholipids.

For the Ford Change bar plots, each CerPE and ceramide level (ng/brain) of $dSMPD4^{KO}$ was normalized to the average level of $w^{1118}$ control brain. For principal component analysis (PCA), lipid species were split by headgroup (sphingolipids: ceramide, glucosylceramide, CerPE; phospholipids: PS, PI, PC, PE), z-scored, and plotted using R Studio (version 2023.09.1 + 494). Volcano plots were made using a data frame of lipids *t* tested between control and mutants and multiple-comparison corrected (Benjamini–Hochberg). Species were annotated and plotted as significantly different for $P < 0.05$ (BH corrected) and |fold change| > 1.5 using R Studio.

## Olfactory associative memory assay

Groups of approximately 100 flies were exposed to the first odor (CS + ), which was paired with electric shock (12 instances of a 1.5-second pulse of 60 volts) over a 1-minute period. Flies were exposed to fresh room air for 1-min right after CS+ odor. The same flies were then exposed to a second odor (CS–) without electric shocks for 1-min. After being exposed to the first and second odor, another 1-min break with fresh room air was provided. Flies were then transported to the choice point in the T-maze and chose between the CS+ and CS– odor tubes without any electrical shock for 2 min. The performance index was calculated by subtracting the number of flies choosing the CS– odor from the number of flies choosing the CS+ odor and dividing by the total number of flies. The odors utilized in the olfactory associative memory assay were 4-methylcyclohexanol (MCH; Sigma-Aldrich, product no. 153095) and 3-octanol (OCT, Sigma-Aldrich, product no. 218405).

## Feeding behavior assay

The fly liquid–food interaction counter (FLIC) assay was performed as previously described (Chou et al, 2022). Briefly, one-week-old female flies were anesthetized in ice shortly for sorting and then loaded into feeding chambers between 60 and 30 min before the assay to allow recovery. In all, 10% sucrose solution was provided as food. Flies were raised and tested at 25 °C with a 12/12 light–dark schedule. All experiments were initiated at 12 PM (ZT06). Feeding behaviors were monitored in 24-h sessions, with w1118 and dSMPD4-knockout female flies being tested simultaneously within each session ($n = 4$ flies for each group). The experiments were repeated four times with a total of 16 flies of each group.

## Climbing assay

One-week-old female flies were collected for climbing assays. In total, 5–30 flies were put in a vial and shocked by shaking the vial six times. The number of flies that were above a 10 cm bar line within 10 s after the last shake was then counted. The ratio was then calculated by dividing the number of flies above the bar line by the total number of flies in the vial. Results for the second to sixth trials were then averaged to get the ratio for a single data point.

## Statistical analysis

The quantitative data were analyzed using the two-tailed unpaired Student's *t* test or one-way ANOVA with Dunnett's multiple comparisons. The graphs were expressed as mean ± SEM by using GraphPad Prism 10.

## Inclusion and ethics

We worked to ensure sex balance in the selection of non-human subjects. The author list of this paper includes contributors from the location where the research was conducted who participated in

the data collection, design, analysis, and/or interpretation of the work.

## Data availability

The datasets of gene profiling images are published on the Mendeley Data (nls-mCherry and anti-HA, https://data.mendeley.com/datasets/gg794sznsz/1; mCD8GFP, https://data.mendeley.com/datasets/fty6n2yjxz/1).

The source data of this paper are collected in the following database record: biostudies:S-SCDT-10_1038-S44319-025-00632-0.

## Peer review information

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

## Acknowledgements

We thank Drs. Thomas Clandinin, Hugo Bellen, Robin Hiesinger, Tso-Pang Yao, and people from the labs of Chan, Clandinin, and Bellen for critical discussions. We thank the Imaging and the Biomedical Resource Cores at the First Core Labs at the National Taiwan University (NTU) College of Medicine, Drs. Ya-Wen Chen and Chien-Hsiang Wang at WellGenetics Inc., and Mr. Jian-Quan Lee at the National Health Research Institute of Taiwan (NHRI) for technical assistance. We thank WellGenetics Inc. in Taiwan for the microinjection service. We thank the Brain Research Center at National Tsing Hua University for providing the standard brain template. We thank Dr. Naonobu Fujita, the Bloomington Drosophila Stock Center; the Drosophila Genetic Resource Center at the Kyoto Institute of Technology; the Vienna Drosophila Resource Center; the Developmental Studies Hybridoma Bank; the Drosophila Transgenic RNAi Project at Harvard Medical School for fly strains and reagents. C-YY, K-YH, and C-CL thank Dr. Chao-Chun Chuang for providing consultation on image registration and warping. MRP and YHC are partially supported by the National Science and Technology Council of Taiwan (NSTC) (112-2311-B-001-036-MY3). This work was supported by grants from NTU Hospital (113-L3003) to S-YH, NSTC (113-2320-B-002-022-MY3, 114-2311-B-002-006-), NHRI (EX114-11228NI, NHRI-13A1-CG-CO-08-2325-2), and NTU (114L8522, 114L891303, 114L910203) to C-CC.

## Author contributions

**Fei-Yang Tzou**: Conceptualization; Data curation; Formal analysis; Validation; Investigation; Visualization; Writing—original draft; Project administration; Writing—review and editing. **Cheng-Li Hong**: Data curation; Validation; Investigation; Visualization. **Kai-Hung Chen**: Data curation; Validation; Investigation. **John P Vaughen**: Data curation; Formal analysis; Validation; Investigation; Visualization; Writing—review and editing. **Wan-Syuan Lin**: Data curation; Validation; Investigation. **Chia-Heng Hsu**: Data curation; Validation; Investigation. **Irma Magaly Rivas-Serna**: Data curation; Investigation. **Kai-Yi Hsu**: Formal analysis; Investigation; Visualization; Methodology. **Shuk-Man Ho**: Data curation; Formal analysis; Investigation; Writing—review and editing. **Michael Raphael Panganiban**: Data curation; Formal analysis; Investigation. **Hsin-Ti Hsieh**: Data curation; Formal analysis; Investigation. **Yi-Jhan Li**: Data curation; Formal analysis; Investigation. **Yi Hsiao**: Data curation; Investigation. **Hsin-Chun Yeh**: Data curation; Investigation. **Cheng-Yu Yu**: Formal analysis; Investigation. **Hong-Wen Tang**: Conceptualization; Funding acquisition; Writing—review and editing. **Ya-Hui Chou**: Supervision; Project administration; Writing—review and editing. **Chia-Lin Wu**: Supervision; Project administration; Writing—review and editing. **Chung-Chuan Lo**: Supervision; Project administration; Writing—review and editing. **Vera C Mazurak**: Supervision; Project administration. **M Thomas Clandinin**: Supervision; Project administration. **Shu-Yi Huang**: Conceptualization; Funding acquisition; Writing—review and editing. **Chih-Chiang Chan**: Conceptualization; Supervision;

Validation; Writing—original draft; Project administration; Writing—review and editing.

Source data underlying figure panels in this paper may have individual authorship assigned. Where available, figure panel/source data authorship is listed in the following database record: biostudies:S-SCDT-10_1038-S44319-025-00632-0.

## Disclosure and competing interests statement

The authors declare no competing interests.

# Expanded View Figures

**Figure EV1. Validation of GAL4 expression in 3XHA-T2A-GAL4 (HG) knock-in flies (related to Fig. 2).**

(A) Anti-HA and anti-Cnx99A (ER membrane marker) co-stainings of heterozygous *schlank*-HG L3 salivary glands. (Left) The colocalization of anti-HA and anti-Cnx99A indicated ER localization of the Schlank-3XHA protein. (Right) Punctate anti-HA signals in the nucleus (circled by dashed line) indicate nuclear localization of Schlank-3XHA protein. (B) Comparison of *gba1b*-HG and *gba1b* CRIMIC-GAL4 expressions. The GAL4 expressions in the adult brain of 1-week-old flies are visualized with UAS-nls-mCherry, and their colocalization with a glial marker (anti-Repo; Cyan) indicates glia-enriched expression of *gba1b*. Quantification and statistics of cell-type expressions are presented in Fig. EV2B,C. (C) Quantification and statistical analyses of numbers of nls-mCherry-positive cells. Bar graphs showed the number of nls-mCherry-positive cells in the central brains (1-week-old adult flies) of HG lines before and after Cre-mediated excision and CRIMIC-GAL4s of *CDase*, *gba1b*, and *sk1*. Data were representative of at least 2 independent experiments. Dots represent individual brains. Data are represented as mean ± SEM ($n \geq 3$). $P$ values (CDase-HG Before Cre vs. CDase-HG After Cre, $P = 0.852$; CDase-HG Before Cre vs. CDase-CRIMIC, $P = 0.034$; CDase-HG After Cre vs. CDase-CRIMIC, $P = 0.007$; gba1b-HG Before Cre vs. gba1b-HG After Cre, $P = 0.869$; gba1b-HG Before Cre vs. gba1b-CRIMIC, $P = 0.004$; gba1b-HG After Cre vs. gba1b-CRIMIC, $P = 0.008$; sk1-HG Before Cre vs. sk1-HG After Cre, $P = 0.189$; sk1-HG Before Cre vs. sk1-HG CRIMIC, $P = 0.002$; sk1-HG After Cre vs. sk1-CRIMIC, $P = 0.006$) were calculated using one-way ANOVA with Dunnett's multiple comparisons. (D) Quantification and statistical analyses of nls-mCherry colocalized to neuronal and glia markers. Bar graphs showed the percentage of nls-mCherry spots colocalizing to neuronal nuclei (anti-Elav; black bars) and glial nuclei (anti-Repo; white bars) in the total number of labeled nls-mCherry (colocalizing with Elav or Repo). The percentage of anti-Elav (88.21%) or anti-Repo (11.79%) spots in the total number of labeled nuclei (the sum of anti-Elav and anti-Repo spots) is shown on the left, and the red dashed line indicates the percentage of anti-Elav spots. Data were presented as mean percentages of at least two independent experiments. $P$ values (sk1-HG Before Cre, $P < 0.001$; sk1-HG After Cre, $P < 0.001$; sk1-CRIMIC, $P < 0.001$; gba1b-HG before Cre, $P < 0.001$; gba1b-HG After Cre, $P < 0.001$; gba1b-CRIMIC, $P < 0.001$; CDase-HG Before Cre, $P < 0.001$; CDase-HG After Cre, $P < 0.001$; CDase-CRIMIC, $P = 0.009$) were calculated using two-tailed unpaired Student's $t$-test, comparing the means between the percentage of nls-mCherry colocalizing with Elav in the total number of labeled nls-mCherry (the sum of Elav- and Repo-colocalizing nls-mCherry spots) and the percentage of Elav spots in the total number of labeled nuclei (the sum of Elav and Repo spots).

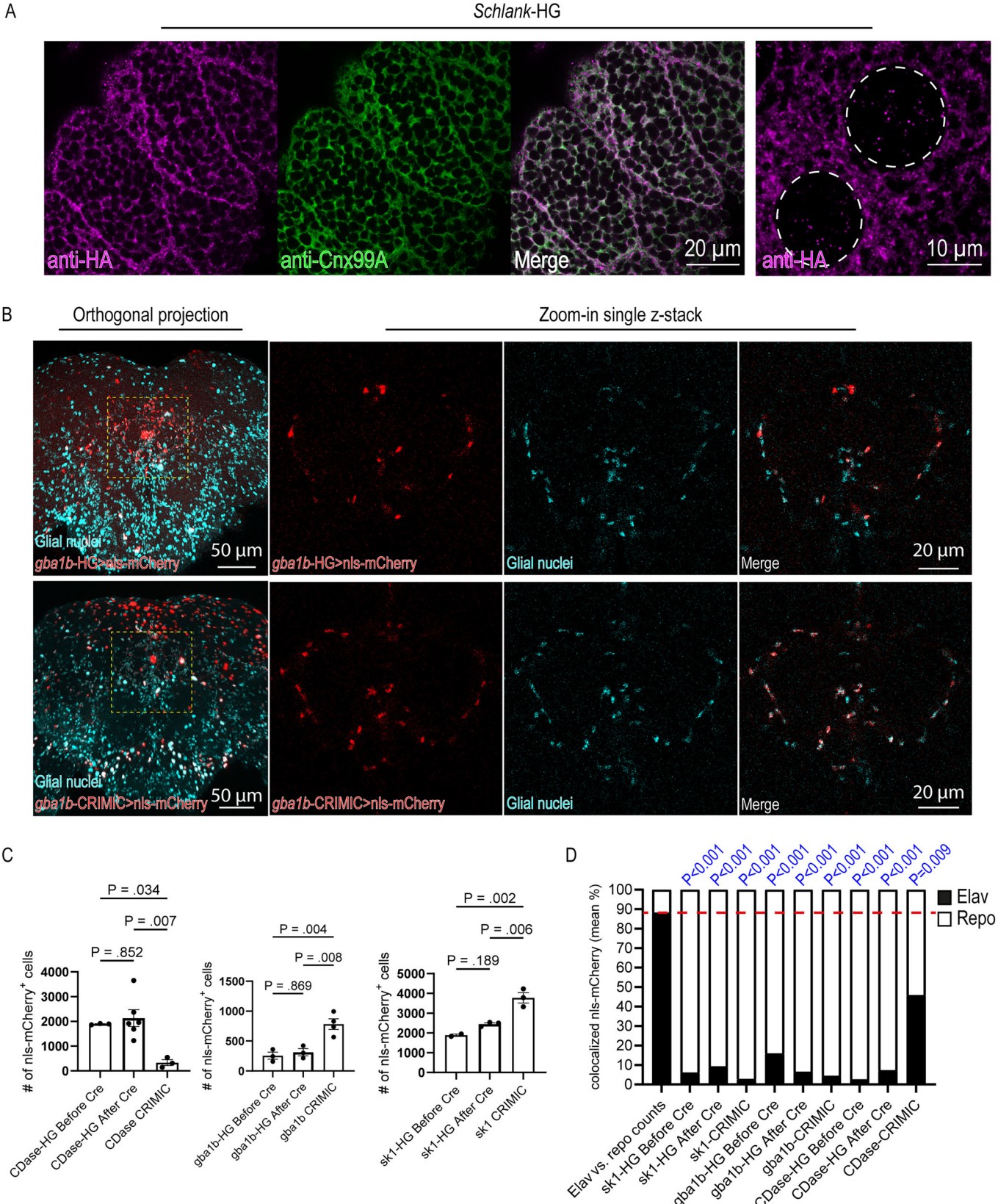

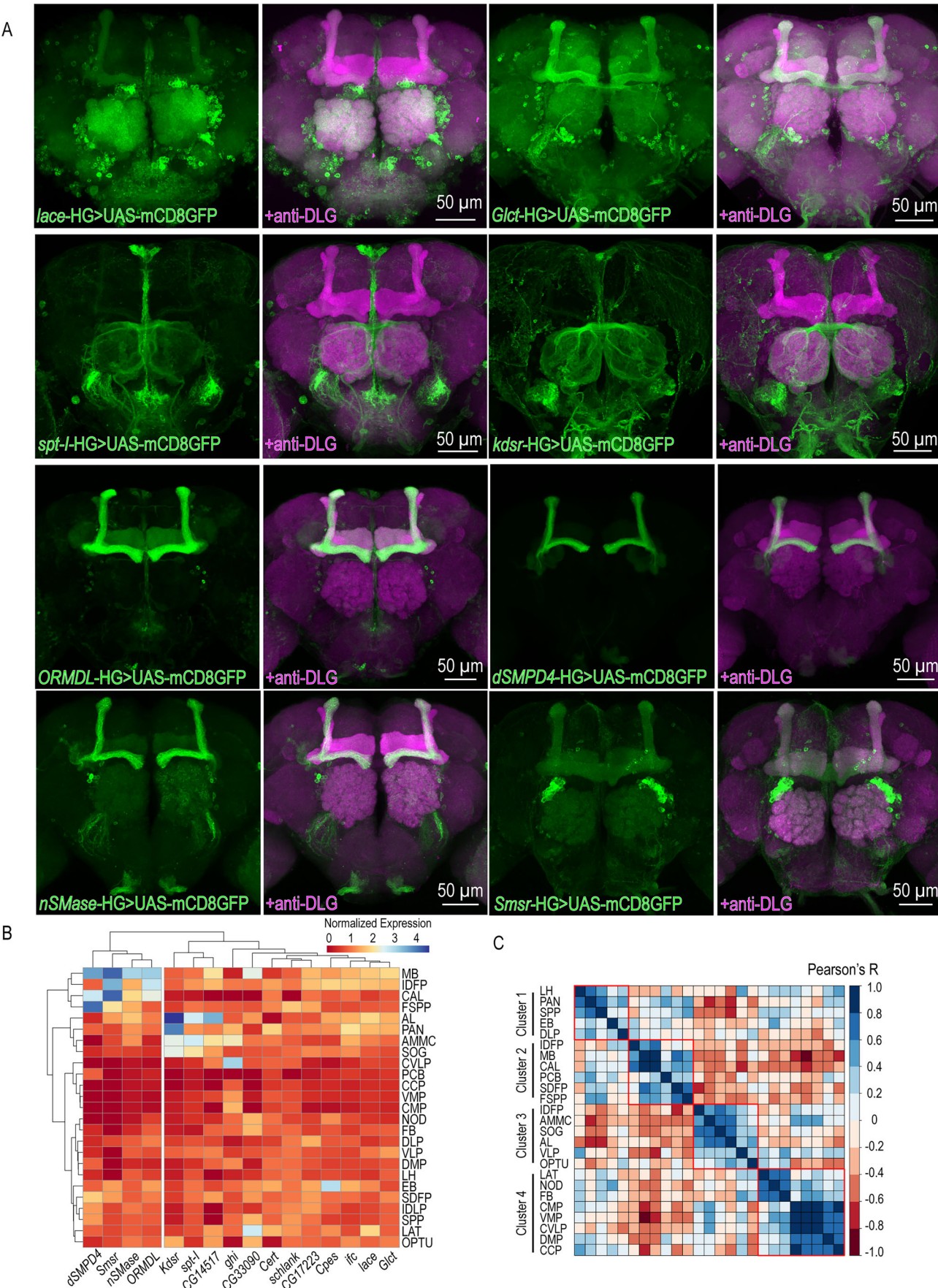

◀ **Figure EV2. Transcriptional profiling with HG revealed highly diverse expression patterns of SPL regulators in different brain regions (related to Fig. 3).**

(A) The expression pattern of targeted genes in the central brain of young adult flies (1-week-old) is visualized with UAS-mCD8GFP (green) and neuropil staining (magenta; anti-DLG). (B) A hierarchical heatmap of SPL metabolism gene expression by genes (columns) and neuropils (rows). The mean value of mCD8GFP intensity of each gene in each neuropil was normalized to the mean value of mCD8GFP of each gene in all neuropils. Red indicates high expression, and blue indicates low expression. Rows and columns are clustered based on gene expression similarity. (C) Correlation matrix displays Pearson correlation coefficients between gene expression profiles of different neuropil. Blue indicates high positive correlation, Red indicates high negative correlation, and white indicates no correlation. Representative figures of mCD8GFP images of all SPL regulators are presented in Appendix Figs. S22–30. The data in (B, C) represent the mCD8GFP image of one brain for individual genes. The representative figure of the brain alignment is presented in Appendix Fig. S31.

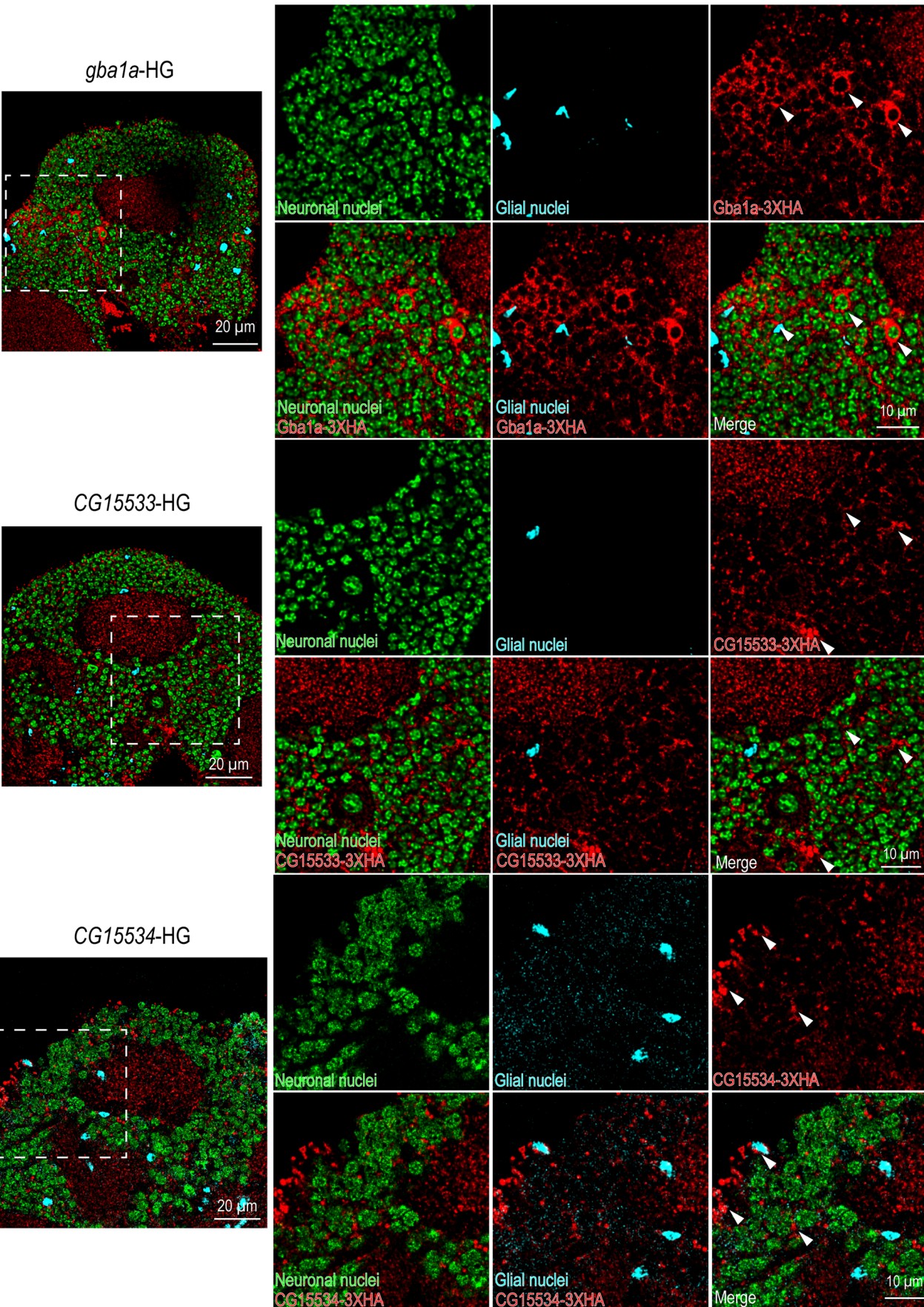

**Figure EV3. Anti-HA immunostainings of adult brains from HG lines *gba1a*, *CG15533*, *CG15534* (related to Fig. 4).**

The protein distribution of Gba1a-3XHA, CG15533-3XHA, and CG15534-3XHA are visualized by anti-HA immunostaining (red) with co-stainings of neuronal (green; anti-Elav) and glial (cyan; anti-Repo) nuclei of young adult brains (1-week-old). Arrowheads indicate cells showing positive anti-HA signals.

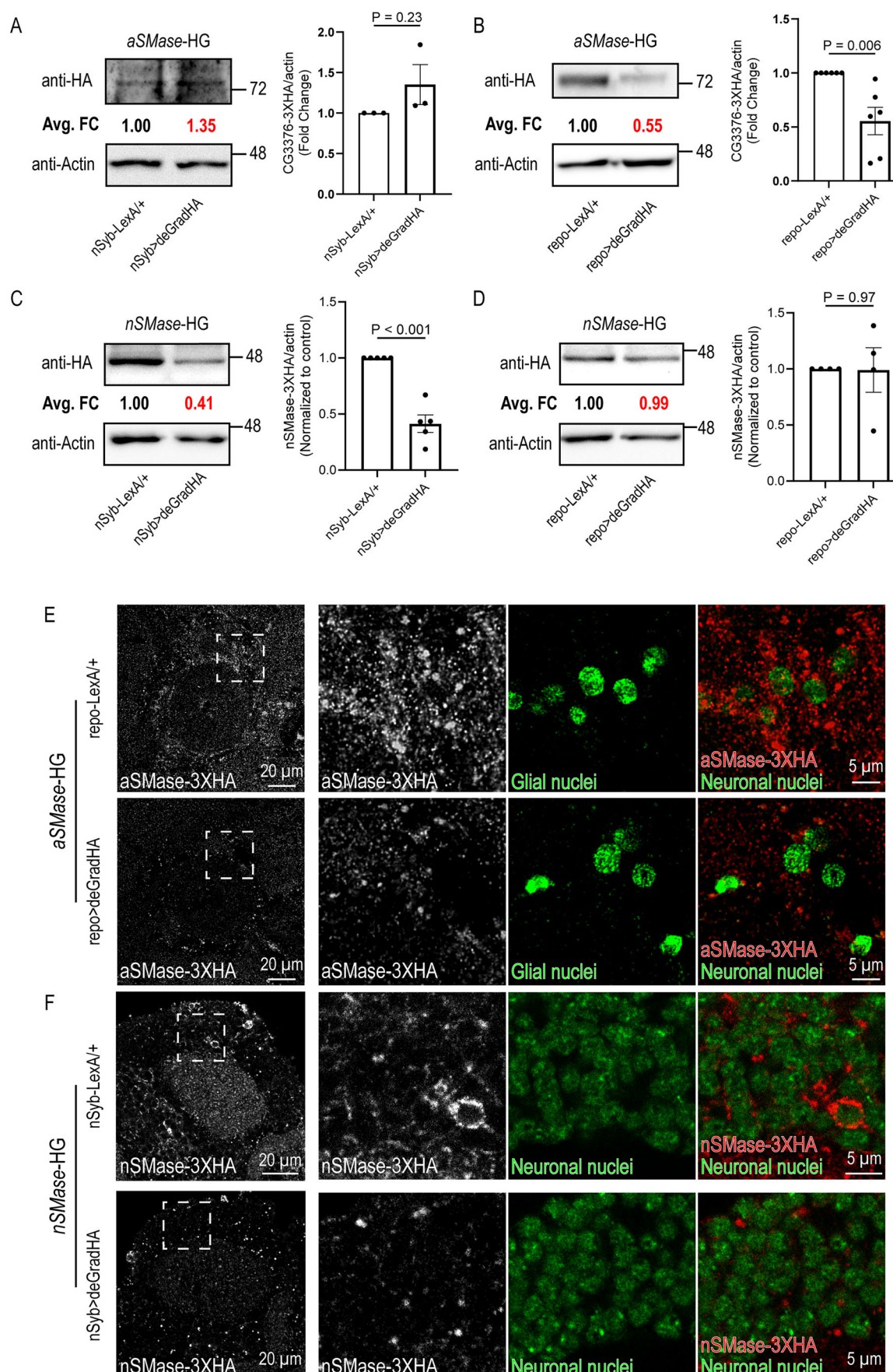

**Figure EV4.  The deGradHA induced cell type-specific protein degradation of aSMase-3XHA and nSMase-3XHA in the brain (related to Fig. 5).**

(A) Representative immunoblot of the adult head extract of *aSMase*-HG line with (*aSMase*-HG/+; *nSyb*-LexA/*LexAop-deGradHA*) or without (*aSMase*-HG/+; *nSyb*-LexA/+) neuronal expression of deGradHA. The average fold change (Avg. FC) of aSMase-3XHA immunoreactivity normalized to loading control (anti-Actin) is shown in the figure. (Right) Quantification of the deGradHA protein degradation efficiency are shown on the right. Data are represented as mean ± SEM of 3 independent experiments. The *P* value (*P* = 0.23) was calculated using two-tailed unpaired Student's *t* test. (B) Representative immunoblot of the adult head extract of *aSMase*-HG line with (*Repo*-LexA/+ or y; *aSMase*-HG/+; *LexAop-deGradHA*/+) or without (*Repo*-LexA/+ or y; *aSMase*-HG/+) glial expression of deGradHA. (Right) Quantification of the deGradHA protein degradation efficiency are shown on the right. Data are represented as mean ± SEM of 6 independent experiments. The *P* value (*P* = 0.006) was calculated using two-tailed unpaired Student's *t*-test. (C) Representative immunoblot of the adult brain extract of *nSMase*-HG line with (*nSyb*-LexA/+; *nSMase*-HG/*LexAop-deGradHA*) or without (*nSyb*-LexA/+; *nSMase*-HG/+) neuronal expression of deGradHA. The average fold change (Avg. FC) of nSMase-3XHA immunoreactivity normalized to loading control (anti-Actin) is shown in the figure. (Right) Quantification of the deGradHA protein degradation efficiency are shown on the right. Data are represented as mean ± SEM of 5 independent experiments. The *P* value (*P* < 0.001) was calculated using two-tailed unpaired Student's *t* test. (D) Representative immunoblot of the adult brain extract of *nSMase*-HG line with (*Repo*-LexA/+ or y;; *nSMase*-HG/*LexAop-deGradHA*) or without (*Repo*-LexA/+ or y;; *nSMase*-HG/+) glial expression of deGradHA. Quantification of the deGradHA protein degradation efficiency are shown on the right. Data are represented as mean ± SEM of 4 independent experiments. The *P* value (*P* = 0.97) was calculated using two-tailed unpaired Student's *t* test. (E) Glial-specific degradation of aSMase-3XHA protein in the adult brain. The protein distribution of aSMase-3XHA is visualized by anti-HA immunostaining with co-stainings of glial nuclei (green; anti-Repo) of young adult brains (1-week-old). (Left) The representative image is taken from the posterior view of the Calyx, and the dashed-line square indicates the region of zoom-in images shown on the right. (Right) Zoom-in images show anti-HA signals (gray in the single-channel image; red in the overlay image) surrounding glial nuclei (green). (F) Neuron-specific degradation of nSMase-3XHA protein in the adult brain. The protein distribution of nSMase-3XHA is visualized by anti-HA immunostaining with co-stainings of neuronal nuclei (green; anti-Elav) of young adult brains (1-week-old). (Left) The representative image is taken from the posterior view of the Calyx, and the dashed-line square indicates the region of zoom-in images shown on the right. (Right) Zoom-in images show anti-HA signals (gray in the single-channel image; red in the overlay image) surrounding neuronal nuclei (green).

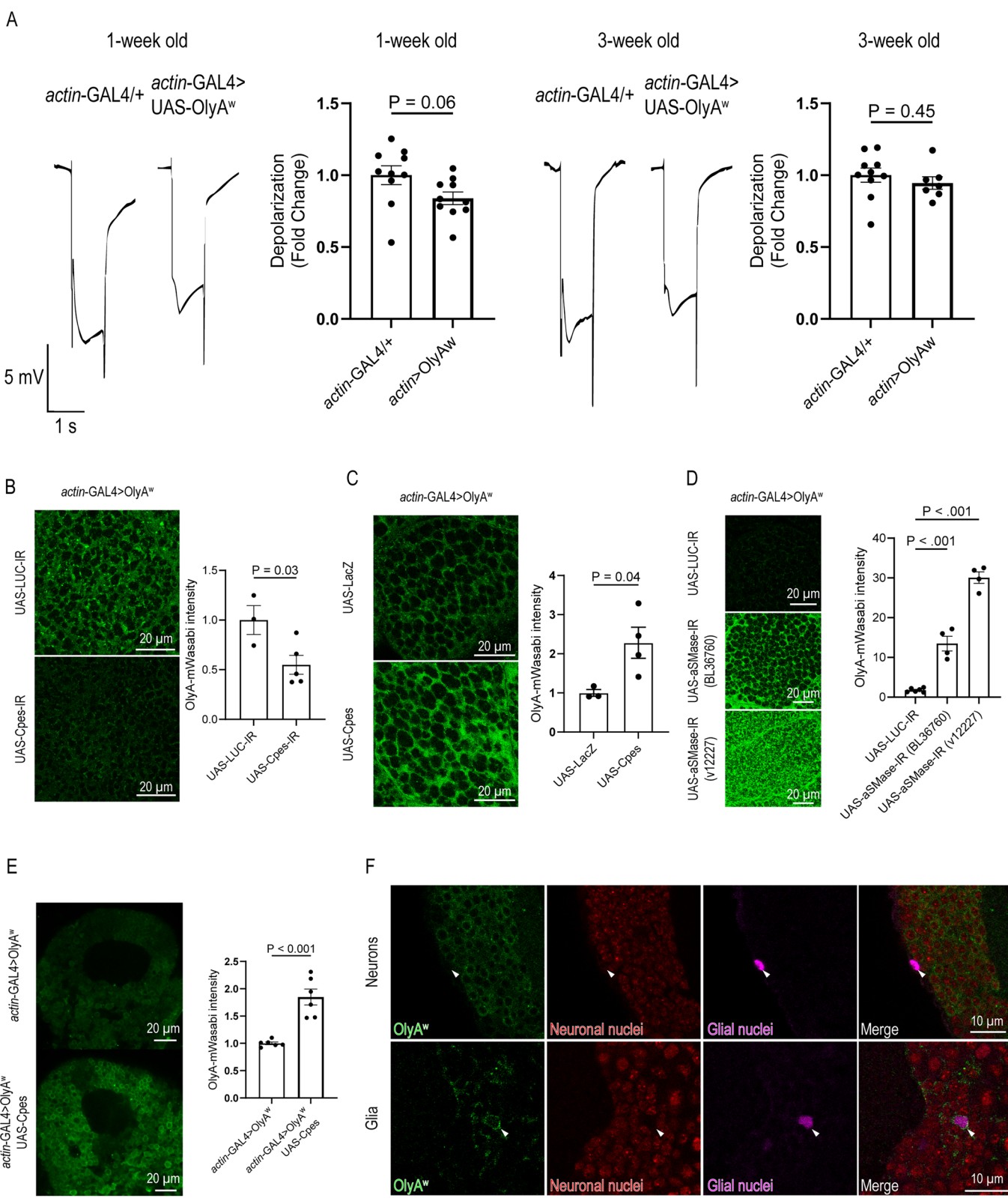

◄ **Figure EV5. Correlation between OlyA^w intensity and CerPE levels; Cell type-specific expression of OlyA^w in the brain; the effect of OlyA^w on neuronal function (related to Fig. 6).**

(A) Representative electroretinogram (ERG) traces of control (*Actin*-GAL4/+) and OlyA^w-overexpressing (*Actin*-GAL4/+; *UAS-OlyA^w*/+) flies at 1 week and 3 weeks of age. Bar graphs show the quantification of the depolarization (normalized to the control). Data are representative of 3 independent experiments. Data are represented as mean ± SEM. The $P$ value (1-week old, $P = 0.06$; 3-week old, $P = 0.45$) was calculated using two-tailed unpaired Student's $t$ test. (B) OlyA^w expression in the L3 salivary glands upon control (*Actin*-GAL4 > UAS-OlyA^w + *UAS-LUC-IR*) and *Cpes* (*Actin*-GAL4 > *UAS-OlyA^w* + *UAS-Cpes-IR*) RNAi knockdowns. Data are representative of at least 2 independent experiments. Data are represented as mean ± SEM ($n \geq 3$). The $P$ value ($P = 0.03$) were calculated using two-tailed unpaired Student's $t$ test. (C) OlyA^w expression in the L3 salivary glands upon control (*Actin*-GAL4 > *UAS-OlyA^w* + *UAS-LacZ*) and *Cpes* (*Actin*-GAL4 > *UAS-OlyA^w* + *UAS-Cpes*) overexpression. Data are representative of at least 2 independent experiments. Data are represented as mean ± SEM ($n \geq 3$). The $P$ value ($P = 0.04$) was calculated using two-tailed unpaired Student's $t$ test. (D) OlyA^w expression in the L3 salivary glands upon control (*Actin*-GAL4 > *UAS-OlyA^w* + *UAS-LUC-IR*) and *aSMase* (*Actin*-GAL4 > *UAS-OlyA^w* + *UAS-aSMase-IR*) RNAi knockdowns. Data are representative of at least 2 independent experiments. Data are represented as mean ± SEM ($n \geq 4$). $P$ values [aSMase-IR (BDSC 36760), $P < 0.001$; aSMase-IR (VDRC 12227), $P < 0.001$] were calculated using two-tailed unpaired Student's $t$ test. (E) OlyA^w expression in the adult brain in control (*Actin*-GAL4 > *UAS-OlyA^w*) and *Cpes*-overexpressing (*Actin*-GAL4 > *UAS-OlyA^w* + *UAS-Cpes*) flies. Data are representative of at least 2 independent experiments. Data are represented as mean ± SEM ($n \geq 3$). The $P$ value ($P < 0.001$) was calculated using two-tailed unpaired Student's $t$ test. (F) (top) OlyA^w expression driven by a pan-neuron driver (*nSyb*-GAL4) in the adult brains. Neuronal nuclei are labeled by anti-Elav (red), and the glial nucleus is labeled by anti-Repo (magenta; arrowhead). (Bottom) OlyA^w expression driven by a pan-glia driver (*repo*-GAL4) in the adult brains. Neuronal nuclei are labeled by anti-Elav (red), and the glial nucleus is labeled by anti-Repo (magenta; arrowhead).

A

repo-GAL4>UAS-OlyA^w

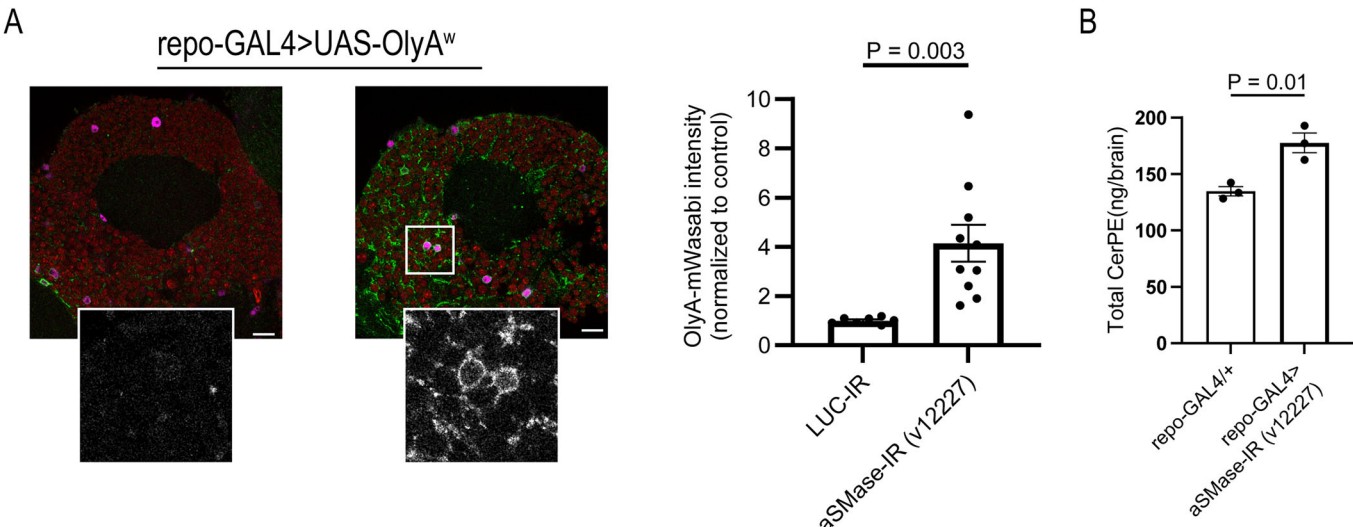

B

C

*aSMase-sgRNA+Cas9*

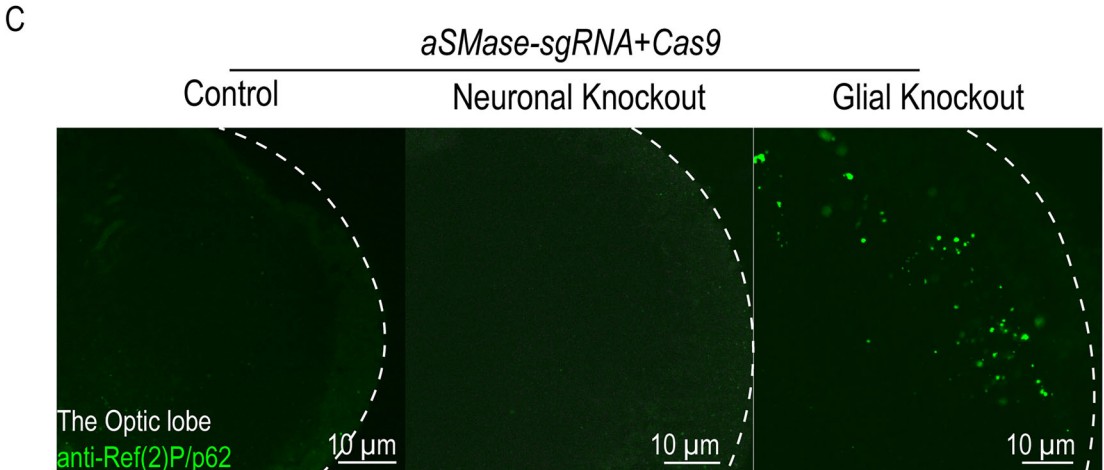

D

E

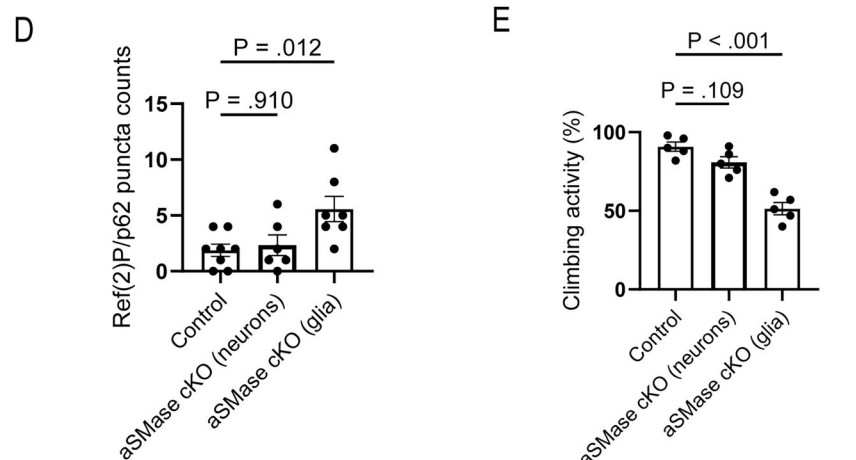

**Figure EV6. aSMase loss-of-function phenotypes (related to Fig. 7).**

(A) OlyA^w expression and the *aSMases RNAi* knockdown in glia. UAS-OlyA^w expression and aSMases RNAi were driven by *repo*-GAL4 (*repo*-GAL4 > *UAS-OlyA^w*+*aSMase-RNAi*). Confocal images were captured from the posterior view of the Calyx in the adult brain of young adult flies (1 week old). (Right) Quantifications of OlyA^w intensity in the cortical region. The P value (*P* = 0.003) was calculated using two-tailed unpaired Student's *t* test. Data are represented as mean ± SEM (*n* ≥ 7). Data are representative of 4 independent experiments. (B) Quantification of the total CerPE levels in control (*repo*-GAL4/+) and *aSMase* knockdown (*repo*-GAL4/UAS-aSMase-RNAi) flies (day 10 female). The P value (*P* = 0.01) were calculated using two-tailed unpaired Student's *t* test. Data are represented as mean ± SEM (n > 3). Data are representative of 3 independent experiments. (C) Ref(2)P/p62 immunostainings in the optic lobe of adult brains from control or *aSMase* conditional knockout flies (1-week-old). Cell type-specific *aSMase* knockout was achieved by using small guide RNA targeting *aSMase* and Cas9 driven by neuronal (*nSyb*-GAL4) or glial (*repo*-GAL4) drivers. Control: *UAS-Cas9/+*; *UAS-aSMase-sgRNA/+*; neuronal cKO: *nSyb>Cas9+aSMase-sgRNA*; glial cKO: *repo>Cas9+aSMase-sgRNA*. (D) Quantification of Ref(2)P/p62 puncta counts. Data are represented as mean ± SEM (*n* ≥ 6). P values [Control vs. aSMase cKO (neuron), *P* = 0.910; Control vs. aSMase cKO (glia), *P* = 0.012] were calculated using one-way ANOVA with Tukey's multiple comparisons. Data are representative of >5 independent experiments. (E) Climbing assay of control or *aSMase* conditional knockout flies (1-week-old). Cell type-specific *aSMase* knockout was achieved by using small guide RNA targeting *aSMase* and Cas9 driven by neuronal (*nSyb*-GAL4) or glial (*repo*-GAL4) drivers. Data are represented as mean ± SEM (*n* = 5). P values [Control vs. aSMase cKO (neuron), *P* = 0.109; Control vs. aSMase cKO (glia), *P* < 0.001] were calculated using one-way ANOVA with Tukey's multiple comparisons. Data are representative of 5 independent experiments. Control: *UAS-Cas9/+*; *UAS-aSMase-sgRNA/+*; neuronal cKO: *nSyb>Cas9+aSMase-sgRNA*; glial cKO: *repo>Cas9+aSMase-sgRNA*.

