## [Peer Review File · EMBO Reports]

Functional profiling and visualization of the sphingolipid metabolic network in vivo

Fei-Yang Tzou, Cheng-Li Hong, Kai-Hung Chen, John Vaughen, Wan-Syuan Lin, Chia-Heng Hsu, Irma Rivas-Serna, Kai-Yi Hsu, Shuk-Man Ho, Michael Panganiban, Hsin-Ti Hsieh, Yi-Jhan Li, Yi Hsiao, Hsin-Chun Yeh, Cheng-Yu Yu, Hong-Wen Tang, Ya-Hui Chou, Chia-Lin Wu, Chung-Chuan Lo, Vera Mazurak, M. Clandinin, Shu-Yi Huang, and Chih-Chiang Chan

Corresponding author(s): Chih-Chiang Chan (chancc1@ntu.edu.tw)

Review Timeline:

Transfer Date:	30th Jun 25
Editorial Decision:	18th Aug 25
Revision Received:	29th Aug 25
Accepted:	24th Oct 25

Editor: Deniz Senyilmaz Tiebe / Kurt Weir

Transaction Report: This manuscript was transferred to EMBO reports following peer review at The EMBO Journal.

Referee #1:

Tzou et al. developed a genetic toolkit by inserting a 3xHA-T2A-Gal4 at the stop codon of genes that are implicated in sphingolipid metabolism, hereafter called HG. This allows to monitor the expression patterns of genes of the sphingolipid pathway components and allows elegant genetic manipulations. Using this toolkit, the authors claim an astonishing array of cell-type specific expression patterns for different components of the sphingolipid pathway. The authors then combine their genetic toolkit with OlyA-mWasabi to visualize CerPE dynamics in vivo. Additionally, the authors use the deGradHA strategy to specifically degrade HA tagged proteins in combination with the HG toolkit to validate the cell-type specific and organelle specific expression and function of the three CerPE catabolism enzymes: CG6962 (a SMase), the nSMase and the aSMase. Finally, the authors dissect the biological functions of CG6962 using a combination of knockout lines and HG toolkit. The authors have done an enormous amount of work to create all these lines and test their expression patterns. Unfortunately, the quality of the presented data, the lack of co-stainings with nls-mCherry with neuronal and glial markers and the size and resolution of the images throughout the paper do not allow to assess the expression patterns of most genes. The manuscript is focused on presenting huge amounts of low-quality data. It would very much benefit from reducing the number of experiments and improving the quality of the data.

I will make suggestions to improve the quality of a few figures as this study has merits and it would be valuable if the reagents are all made available publicly via the BDSC or the Kyoto stock center. This should also be a prerequisite for publication. The authors should also very much reduce the number of pictures presented in the manuscript and setup an open data resource/website where they deposit high quality images of expression pattern for rest of the Sphingolipid pathway components, as is typically expected for a resource.

Indeed, this resource will allow a thorough analysis of the sphingolipid pathway in *Drosophila melanogaster*. If the claims of the authors can be substantiated with high quality data I would be strongly in favor of publication. However, given that there are so many issues with quality of the data I will only comment on the first few figures as the message should be obvious.

Here are my suggestions.

Major comments:

The authors should clearly mention the number and age of animals used for each experiment in this study. Also, they need to provide detailed genotypes for all the different experiments in the figure legends. The genotypes of control genotypes are not spelled out.

Figure 1: The image quality in Figure 1c is very poor.

Extended data figure 1b: The resolution of these images is very low and no clear pattern of

HA staining can be identified. The authors could easily reduce the number of figures by 50% or more and show high resolution images. Additional co-stainings with other markers would be important.

Extended data figure 2a: The images are low quality and making it difficult to draw any reliable inferences from these images. Additionally, the authors use only three CRIMIC lines without any explanation why only these lines were chosen.

Extended data figure 2b: ORMDL, nSMase and GlcT show very different expression pattern before and after Cre excision. This raises significant concern about the expression pattern shown throughout the paper. To show that the difference in expression pattern is indeed a result of 3'UTRs the authors must validate the expression pattern of more genes with available CRIMICS.

Additionally, the authors should clarify if they tested all the lines with or without the Cre excision and comment on the pattern of expression of the two.

Extended Figure 2d: The image resolution is very low to confirm the neuronal. vs glial expression pattern.

Figure 2d: Adult fly brain has about 90% neurons and 10% glia. However, the *repo>mCD8GFP* shows a much broader staining pattern. The *nls-mCherry* stainings will solve that issue.

Figure 2e-o: The image resolution is very poor. This makes it very difficult to identify any clear expression pattern. The authors should also use alternative markers to better visualize the patterns of expression. Additionally, add a comparative analysis with currently known and experimentally validated expression patterns for sphingolipid pathway components. For example, *Gba1b* expression pattern shown in Wang et al., 2022 is significantly different from the one seen in this manuscript because the authors did not use *nls-mCherry*

Extended data figure 3: The authors indicate that the *mCD8-GFP* pattern recapitulates the endogenous pattern of expression. However, the images are low resolution and very small, making it difficult to see any clear cell-type specific patterns. They should select about 6-10 of the most important genes and show high resolution images and use *nls-mCherry* and co-label with *repo* and *elav*.

Extended data figure 4: The glial staining is very inconsistent across the different genes and therefore it is difficult to visualize the data. The associated pie charts also do not match the representative images. Please provide clear representative images for better understanding and clarity.

Extended data figure 5: The age-associated reduction in *ifc>mCD8:GFP* may be validated using anti-HA western blot at 1, 3 and 6-weeks.

Please provide raw data for the depolarization quantifications.

Figure 3: The GFP staining looks weaker and more confined than the HA staining. However,

GFP is driven by gene-specific Gal4 and is an amplified signal as compared to HA staining that indicates endogenous protein levels. Please explain these issues or repeat experiments. The authors use GFP and HA staining patterns of *kdsr* to imply that the protein is made in glia and travel to neurons for function. This is a significant claim and really needs to be supported with other data.

Referee #2:

Functional profiling and visualization of the sphingolipid metabolic network in vivo

Fei-Yang Tzou et al 2024

In this study the authors have identified 52 genes as direct regulators of sphingolipid metabolism and generated knockin transgenic flies with engineered HA tag followed by T2A self-cleaving peptide followed by a Gal4 sequence immediately proximal to the stop codon of these genes using Crispr-Cas9 mediated genome engineering (called HG lines). They evaluated the HA-tagging, Gal4 expression and Western analysis of CDase as a proof of principle for demonstrating the utility of these lines. They compared the transcriptional profiling of the HG lines utilizing Gal4 driven mCD8-GFP fluorescence staining to evaluate expression pattern and compared it to staining driven by previously established CRIMIC-Gal4 lines that cover about a third of the current HG lines. They compared expression abundance with published single cell RNA seq data. They find both similarities and differences between the various methods of evaluation of gene expression. The SPL genes segregated across 25 regions in the brain. Based on the expression of HG line Gal4 driven mCD8GFP reporter components of the SPT complex that catalyzes the first rate-limiting step of the de novo sphingolipid biosynthetic pathway including the *ORMDL* gene is enriched in the neurons, the immediate downstream gene was expressed preferentially in the glia. The gene that acts immediately downstream of this reaction in this pathway was observed to be enriched in the neurons and enrichment of the subsequent enzyme in the pathway (dihydroceramide desaturase, *infantile crescent*) in the glia. They also report differential expression between the neurons and glia for several of the salvage pathway genes. The authors focused on the posterior view of Calyx to visualize gene expression and protein between neuronal cell bodies, neuropil, and different glial cells. They probed the metabolism of ceramide phosphoethanolamine (CerPE) using a mWasabi tagged *Ostreolysin A* (*OlyAw*) and validated its ability to bind using the *cpes* mutant. They report that overexpression of *OlyAw* was not toxic to flies. They used this reporter to examine cellular CerPE metabolism regulated by sphingomyelinase family of proteins. They utilized

deGradHA that targets HA tagged proteins for degradation. Neuronal expression of deGradHA decreased CG6962 and nSMase-3xHA whereas aSMase-3xHA was targeted by glial deGradHA, as observed by fluorescence staining. They show increased OlyAw upon neuronal degradation of neutral sphingomyelinase homolog and surprisingly increased neuronal OlyAw staining upon glial degradation of the acid sphingomyelinase homolog. The authors perform several knockdown experiments using neuronal and glial drivers to target putative lipoprotein transfer proteins and conclude that neuronal apoLTP transported CerPE from neurons into glia. Glial knockdown of apolpp, GLaz, MTP, apoLTP and LpR2 significantly increased OlyAw staining.

Based on these observations' authors suggested that the de novo sphingolipid biosynthetic pathway may not be cell autonomous in the brain. Overall, authors performed large volume of work covering transcriptional, translational profiling of sphingolipid metabolic enzymes and CerPE trafficking between neuron and glia, and characterization of CG6962 knockout animals.

There is a potential that the tools developed in this study could be useful to the Drosophila community. However, many conclusions drawn by the authors are premature and are insufficiently demonstrated. Very many of them are inconsistent with the known literature.

Major comments.

1. Since the paper is a pathway wide analysis, it is imperative that the authors provide comprehensive coverage of all the probes alluded to in the study. The expression data for all 52 tagged proteins should be provided. The protein expression data is not informative. It should at least be correlated with respect to glia/neuronal and nuclear colocalization data. The information should be shown for all 52 tagged proteins. It is very likely that several of the genes do not tolerate tagging at the C-terminus and the authors would have some information about the states of such proteins from their Western and or immunofluorescence data. It is critical they include this negative data and discuss this in the text of the manuscript. This will also provide consistency between the Western blot and immunofluorescence data that is currently lacking (Extended Data Figure 1a and b).

2. Based on immunostaining pattern of sphingolipid specific HG knock-in lines authors suggested that the de novo biosynthetic pathway is split between neuron and glia. However, this conclusion has not been substantially tested. Enrichment of the de novo sphingolipid biosynthetic enzymes in different subset of neurons or glia may simply mean those enzymes have different stabilities in different cell types or may have roles other than sphingolipid biosynthesis. For instance, when examined closely, Fig.2e and f, it appears that even subunits of same enzyme, SPT complex (Lace, and SPT-1) that catalyze the first

step of de novo sphingolipid biosynthesis are not expressed in same neuronal cell type. This immunostaining could simply mean that the transcriptional activities of Lace and Spt-1 subunits are different in different subset of neurons. On a similar note, detection of IFC protein in glia more strongly than in neurons does not necessarily mean that initial 3 steps of de novo sphingolipid biosynthesis do not occur in glial cells. For instance, an elegant genetic screening study by Ghosh, A et al 2013 PMID: 24348263, have shown that glial specific knockdown of Lace, Spt-1, Schlank, and Des1 induced severe glial membrane swelling phenotype, and this phenotype was not observed when these genes including lace and Spt-1 were knocked down in neurons, these results clearly argue for a cell autonomous function for sphingolipid biosynthetic pathway genes in Drosophila glia. According to the authors' hypothesis, neuronal specific knockdown of Lace or Spt-1 should abolish CerPE production in glia as its precursors are originating from neurons. However, data from Ghosh et al 2013 suggest that this is not the case. In this context, it is hard to accept authors' claim that the de novo sphingolipid biosynthetic pathway is split between neurons and glia, especially in the absence of substantial genetic, cell biological and biochemical evidence.

3. Lines 152-154 - the authors indicate there are minor differences between CRIMIC-Gal4s and HG lines. One could easily argue that there are huge differences. To consolidate their claims that authors are advised to perform higher resolution imaging with colocalization studies and neuronal/glial specific knockdowns to show that both drivers are reporting expression in similar group of cells.
4. Figure 3. In several of the panels it is difficult to recognize the neuropil. For example, in figure 3i the Gal4 driver seems to be driving the expression in Glia while in the same brain HA tagged protein shows a neuronal cell body pattern?
5. Line 560-561 - The authors claim despite knocking down aSMase in glia it led to CerPE accumulations in neurons - This is a loaded sentence. The authors make no effort to evaluate how an enzyme that acts in glia causes accumulation of CerPE in neurons since the transport of CerPE from neurons is not compromised and thus the authors should see increased CerPE in glia before seeing any effect on neuron and how do neurons end up accumulating CerPE?
6. Recent studies by Wang et 2022 (PMID: 35857503) and Vaughen et al 2022 (PMID: 35961319) have demonstrated that GlcCer is trafficked from neurons to glia via exosomes for lysosomal degradation in the glia. In similar lines authors in this study suggest CerPE is also trafficked from neurons to glia via lipoprotein particles to degrade in glial lysosomes. Although this finding is interesting, the evidence shown in this manuscript is not strong enough to demonstrate this finding. I suggest, overexpress UAS CPES and OlyAw in control and cpes mutant neurons and demonstrate that CPE and OlyAw accumulates in glial specific endosomes/lysosomes but not in neuronal specific endosomes/lysosomes.
7. OlyAw staining in Fig.4b & c is not convincing, as it appears that general expression of

OlyAw is reduced in cpes mutants that may be independent of OlyAw protein's ability to binding to CPE in vivo. Expression of OlyAw in Fig.4b is restricted to neuropile compartment and excluded in cortical compartment, in contrast, expression of OlyAw in Fig.4d & e is restricted to cortical compartment and excluded in neuropile compartment, which cannot be explained by differences in Tubulin Gal4 and Actin Gal4, as both express ubiquitously. It appears from Fig.4e, Fig. 5h-l, and Fig.6a & b, OlyAw protein is predominantly localized to neuronal ER or cytoplasm and that is independent of whether it is expressed by neuronal specific Gal4 (*nsyb*) or glial specific Gal4 (*repo*). Previous study by Bhat, H.B. et al 2015 (PMID: 26060215) suggested that only pleurotolysin A2 (*plyA2*) but not OlyA bound to CPE independent of cholesterol. This has also been validated in the cpes mutant by Kunduri et al 2022 (PMID: 36170207). In this context, it raises concerns that the in vivo OlyAw biosensor described in this study is detecting CPE/cholesterol complexes? and not any cholesterol enriched compartments such as rafts and ER membranes? Taken together better in vivo validation of the OlyAw is required, I suggest, validate OlyAw in cells that are larger in size such as spermatocytes, male accessory glands, or salivary glands or photoreceptor neurons etc., to demonstrate OlyAw is secreted into extracellular space and labels the plasma membrane/rafts/endocytic compartments of same cell or neighboring cells. For instance, express in the neurons and detect them in glia or vice versa. In addition to the discrepancies outlined above, the authors should have seen a generalized OlyAw staining even in cpes mutant flies since the fluorescent probe is being synthesized and secreted from all cells that the Gal4 is expressed in. In addition, this reviewer is surprised that the authors have not observed intense Golgi staining in all cells of the control flies. Secreted proteins transit through the lumen of Golgi while being secreted out. Luminal leaflet is the major site of CPE synthesis. Thus, fluorescence in Golgi should be consistent across all cells irrespective of the state of CerPE degradation. The authors should examine cells from several tissues to examine Golgi staining with Wasabi and examine the pattern and document it in the manuscript.

8. Previous studies from the Suzanne Eaton laboratory have demonstrated that almost all circulating apoLTP, apoLPP and largely MTP is synthesized by the fat body. Brankatsch M et al., 2014 (PMID 25275323) showed that neuronal LTP was exclusively derived by fat body generated LTP that crossed the blood brain barrier. This reviewer is very intrigued by the claim by the authors that apoLTP is expressed in the neuron and apoLPP is expressed in the glia. Also, Yin et al., (PMID: 33893307) study that the authors cite clearly show that astrocyte specific knockdown of apoLTP and LPP did not work for them (as they are not expressed by any cells in the brain, supplementary data 2). Since the current data contradicts a previously well-established finding, the authors will have to provide decisive proof other than CerPE binding of OlyAw to establish beyond reasonable doubt that these two lipoproteins are indeed expressed in neurons and glia. Also, they need to provide why

and how do the apoLTP and apoLPP that will be still in abundance after neuron and glia specific knockdown (since fat body is the major site for synthesis and secretion of these two lipoproteins) are unable to assist in transfer of the sphingolipids.

Minor comments:

1. The authors have been generally agnostic of previous studies on sphingolipid metabolic pathway genes in *Drosophila*. While tangential references have been made in many instances, a proper contextual reference should be made about these studies either in the result sections or in the discussion.
2. Please include SPT protein staining in Fig.3.
3. Since several laboratories have generated cpes mutants and UAS-CPES flies, the source of the mutant and transgenic should be cited in the protocol section of the manuscript.
4. Fig.7f-h. Please add neuronal specific degradation of CG6962 to compare with knockout model.
5. Lines 284-285 - Therefore, intercellular transport of dihydroceramide from neuron to glia is required for ceramide de novo synthesis initiated in neurons to be completed. Such statements lack scholarship. This needs to be validated by showing knockdown of Schlank in neuron results in lack of CerPE in the glia.
6. Line 385-386 - the authors write 'Accumulation of p62 protein aggregate is a hallmark of SPL accumulation in Lysosomes' - There is no evidence for this statement. Kinghorn et al., in their 2016 manuscript show that p62, are markers of lysosomal-autophagic degradation, linking ubiquitinated proteins to the autophagy machinery. If autophagy is inhibited, then p62 accumulates. In some specific instance of trafficking defects p62 accumulates as suggested in the three manuscripts that the author cites. But it is no indicator of sphingolipid or CerPE accumulation as the authors claim.
7. A more convincing experiment would be to test memory deficit and abnormal feeding behavior in neuronal vs glial degradHA flies and validate the specific role in neurons.

Referee #3:

Summary:

In this study, the authors generated tools to visualise the expression of enzymes involved in sphingolipid metabolism in the CNS. Using these GAL4 lines and a SM/CerPE biosensor generated in this paper, the authors showed that SM is shuttled between neurons and glia via apolipoproteins for degradation. They also discovered that CG6962 is autonomously

required in neurons to sustain brain function. In general, the tools generated in this paper are valuable resources for the community. However, they need to be more carefully validated. In many cases, the images did not clearly support the conclusions made in the results section. Furthermore, the introduction part only addresses the technical hurdle for studying sphingolipid metabolism, without providing sufficient background on what has been known about sphingolipid metabolism in *Drosophila* brains. The way the manuscript is written needs to be improved. The authors should have linked the 4 parts (GAL4 tools, OlyAw biosensor, Glial neuron SM shuttle and CG6962 function) better, so that it is more coherent.

Major Comments:

The HG Knock in animals of SPL metabolism genes reveal endogenous gene expression and protein localization

Are the proteins tagged with 3XHA functional? With CDase-3XHA-T2-GAL4, do you get homozygotes? Are they embryonic lethal?

134-135

Are the patterning of 3XHA-tagged sphingolipid regulators shown in extended data Fig. 1b matched with what have been shown previously with antibody staining? I would like to see how the staining using antibody against the protein colocalises with the 3XHA tagged one in a heterozygote.

Extended data Fig 2b. What about glia? Does Cre excision changed the expression pattern in glia and how's that compared to the CRIMIC as you showed in a?

The HG Knockin animals revealed heterogeneous gene expression of the SPL metabolic network in the brain

197-201 Fig 2. Can you explain how you aligned your 3D confocal images to the brain wiring map? With the heatmap, did you normalise the GFP intensity to the area of each neuropil region based on the anti-DLG staining? If so, I think you should show the DLG staining of the same focal plane next to your image showing GFP pattern. d-h, to claim a certain gene is expressed in neurons or glia, you have to label neuron or glia in the same experiment where you drove mCD8GFP using specific HG. elavQF, repoQF and QUAS-RFP are all commercially available.

Extended Fig 4.

You quantified what percentage of mcherry+ cells are glial cells and what percentage are

neurons. First, I am not sure what area of the brain you showed. Second, it makes more sense to me if you quantify what percentage of neurons is mcherry+ and what percentage of glial cells is mcherry+. E.g. for ORMDL, on the surface of the brain, XX% of glia is mcherry+ . In CVLP neuropil region (based on DLG staining), XXX% of neurons is mcherry+. And other regions... You also need to show statistics (n=3).

226 I am not convinced by your statement that ifc exhibited a glia-enriched pattern.

235-236 where's the image for GBA2?

245: What do you mean by "These findings suggested that SPL catabolism occurs in distinct subcellular compartments in different cell types"? I haven't seen any data showing these enzymes are expressed in specific subcellular compartments.

246-258 I am not convinced that the expression patterns of these genes remain unchanged except for ifc. Lacc, nSmase, Kdsr, gba1b, Schlank and ifc all looked different upon aging.

Immunostaining of the endogenous 3xHA-tagged proteins revealed cell non-autonomous regulation of the SPL de novo synthesis and catabolism

268-299: I am not convinced by Fig. 3, because glial and neuronal markers are lacking.

SM/CerPE homeostasis is maintained by neuronal neutral SMase and glial acidic SMase

382 Fig. 5F Most of aSMase-3XHA did not colocalise with anti-Cathepsin L. So, you can only claim some aSMase localised to lysosome.

In what subtype of glia does aSMase knockdown causes an increase of OlyAw?

Glial lipoprotein mediates neuronal CerPE levels

412-415: Where do you show glial knockdown of aSMase led to neuronal accumulation of CerPE? In fig. 5 I, you only showed upregulation of OlyA in the brain, there's no neuronal marker in the image. Also, does aSMase knockdown in glia cause neurodegeneration? e.g. does the animal die eventually or have some behavioural defects?

Minor comments:

The HG Knock in animals of SPL metabolism genes reveal endogenous gene expression and protein localization

Extended data Fig 2d. You need to quantify what percentage of mcherry+ cells are positive for glial marker and neuron marker. Also, it's good to add a zoom-in image showing there's indeed no colocalization between glial marker or neuronal marker and mcherry.

160. I thought you are comparing your data with public single cell transcriptome? Not your transcriptional profiling, right? Or you have this data generated in your lab?

161-162. What are these genes with cell type-specific expression pattern that were not observed in the transcriptome? Can you include those in Extended Data Fig. 2e or as a supplemental table?

The HG Knockin animals revealed heterogeneous gene expression of the SPL metabolic network in the brain

220 The allosteric inhibitor for the SPT complex, showed neuron-specific expression (Fig. 2e-f). This should be Fig. 2g. Kdsr ... (Fig. 2h) These figure images are cited wrong in your result session. Reference is needed when you explain the function of each enzyme.

231. Fig 2j-p should be K-0

255-258: Can you explain how you measured the neuron function in the text?

Extended Data fig. 5C. What is the sample size of ifcKO? Is ifcKO mutant or mushroom body specific knockout?

The Ostreolysin A transgene associates with membrane rafts and indicates the abundance and dynamics of CerPE

322-325: Fig4 b-e: Please include quantification.

Fig 4 f-h: Please include quantification (colocalization analysis).

Extended Data Fig. 6 b: p value?

335-337: what's your conclusion? Is CerPE more enriched in nuclear membrane and ER?

SM/CerPE homeostasis is maintained by neuronal neutral SMase and glial acidic SMase
387-388 Fig. 5g quantification.

Gal4 should be GAL4

Referee #4:

The authors have generated a toolkit of transgenic lines for *Drosophila* sphingolipid research. CRISPR-engineered knock-in HG lines for 52 sphingolipid metabolism genes allow GAL4 expression and HA tagging. These lines can also be combined with degradHA for conditional protein degradation. An XFP (mWasabi) based sensor for CerPE/sphingomyelin was also developed and it labels membrane rafts. The new tools are then used to provide evidence that neuronal neutral SMases and glial acidic SMase maintain CerPE levels and that neurons release CerPE, which is then taken up by glia.

The 52 CRISPR-engineered HG lines will be very useful to the fly community for defining the expression patterns as well as the physiological roles of sphingolipid enzymes in many different biological contexts. These HG lines are rather similar in structure and function to the Bellen lab's CRIMIC lines, although, as the authors show, there are neural expression pattern differences that, in some cases, may be due to whether or not the endogenous 3'UTRs are included in the reporter mRNA. Recapitulation of endogenous expression patterns by HG lines is validated in the adult brain using scRNAseq data sets. There are, however, numerous major criticisms (listed below) of both the tool validation and the mechanistic biology sections that must be addressed.

MAJOR CRITICISMS:

- 1) The authors should provide evidence as to which of the HG knock-ins maintain gene function and which disrupt it e.g. are the HG knock-ins homozygous viable or lethal?
- 2) The HG knock-in driver expression patterns and protein localizations are not described in sufficient detail to be useful to the community. To have a better appreciation of the cell-type specificity of expression of each enzyme it is important to display elav and repo channels separately from mCherryNLS and keep the same scale between images (extended Fig. 4). The repo staining seems variable between the different lines. For instance, *ifc* and *schlank* seem to have a higher number of glial cells compared to *lace* and *kdsr* (extended Fig. 4). Also, it should be clarified whether the neuron/glia ratio is calculated only in the area displayed or in the whole brain. *Kdsr*-HA protein seems to have a much wider expression than its transcription pattern (Fig. 3), why is this? Further testing with an independent method would be required to support the authors surprising conclusion that *Kdsr* protein is transported from neurons to glia. Merged channel overlay panels should be added to Fig. 3 to aid interpretation. CG6962-HA also seems to have a broader expression pattern than the CD8:GFP reporter, why is this?
- 3) The usefulness of the HG lines for conditional protein degradation relies on how efficient

the protein knockdown with degradHA is. Functional validation of this appears to rely on a non-quantitative comparison between a few confocal panels, from a region that is not specified i.e no low power "overview" images shown (Fig 5b-c and 7e). A reliable quantitative method is required for these new genetic tools, which are being reported here for the first-time e.g similar to the western blot readout for protein levels used in Ext data Fig 1a.

4) The lipid selectivity/specificity of the new OlyA[w] sensor for CerPE needs further validation using a reliable quantitative method. Showing non-quantitative confocal images of sensor expression in control, Cpes knockout and O/E flies side-by-side is not sufficient (Fig. 4). Also, what is the weak and diffuse signal remaining in the Cpes knockouts and what percentage does this apparently non-specific signal contribute to the signal detected in control genotypes?

5) The localisation of the tubulin-GAL4 driven OlyA[w] signal seems mostly in neurons in the larval brain (Fig. 4b-c) can the authors explain this? Is the OlyA[w] signal also visible in glia?

6) There are problems with the results section on "Glial lipoprotein mediates neuronal CerPE levels". The evidence supporting the two-way shuttling model in Figure 6 is very incomplete. The author knockdown several gene involved in lipid transport, some of them, such as apolpp, apoltp and Mtp, have never been described as being expressed in glia or neurons to our knowledge. It is critical to show evidence that Mtp, apolpp and apoltp are indeed expressed in the cells that are being targeted for knockdown. At present the effects of glial and neuronal knockdowns have been assessed with only one readout in nSyb neurons, the CerPE sensor, and this has yet to be properly validated (see point 4 above). Therefore, the CerPE sensor should also be expressed in glia with both the neuronal and glial RNAi manipulations and an independent readout for non-cell autonomous effects on lipid metabolism would be required to support the two-way shuttling model.

Alternatively, this biological mechanism section could be deleted. This would be acceptable as according to the title and abstract, the main point of the manuscript appears to be to provide a new resource not to advance new biological mechanism in detail.

7) Related to the incomplete evidence for the Fig. 6 model and for the Fig.5 results, a second independent genetic method is needed to validate all RNAi results. e.g a non-overlapping RNAi line or a mutant. This is important as the effect sizes observed with all of the RNAi lines (except neuronal knockdown of apoltp) are modest (less than two fold).

8) Similarly, the authors refer to UAS-Luc-RNAi as the RNAi control, which in the Materials Table is given the BDSC ID of #35788. This line is not an RNAi line but a luciferase overexpression line. So, either the ID number is wrong, or the authors did not use an RNAi control, which would mean that they cannot rule out non-specific effects of activating the RNAi machinery. Also, what is the control genotype in Fig 5, is it the same or different form

Fig 6?

9) In Fig.5h, the example image of nSMase-RNAi shows a higher OlyAw signal than CG6962-RNAi (especially in the neuropil region), but the quantification shows that the increase is not as significant as CG6962-RNAi. Could the authors also clarify how the quantification is made (see point below).

10) In addition to the previous point, lipidomic analysis (Fig. 7a-c) shows that nSMaseKO leads to accumulation of CerPE, which seems inconsistent with the OlyA[w] result (Fig. 5h). Could the authors explain this discrepancy?

11) p62 seems to accumulate in the optic lobe of glial knock-out of aSMase (Fig. 5g). However, it is important to confirm the finding of p62 accumulation independently, using at least one of tools/methods used by the authors in this study e.g glial RNAi of aSMase or the whole body CRISPR mutant. In addition, quantification of fluorescent signals is lacking and should be performed for Fig. 5g.

12) The method for confocal image fluorescent intensity quantifications (currently in Fig. 5 and 6) needs to be added to the Methods section, specifying the region of analysis (which part of cell bodies or neuropil), the software used, parameters being quantified, and the normalisation methods used to account for sample-to-sample intensity differences e.g due to mounting. In addition, are the fold changes shown for the RNAi knockdowns in Fig 6a-b relative to "UAS-Luc-RNAi"? Please specify.

13) At present, the reactions catalysed by many Drosophila enzymes including those of sphingolipid metabolism are not known. Instead, they are inferred from sequence similarity/orthology to mammalian or yeast enzymes, where the actual enzymology has been done. This type of inference can lead to mistakes as the exact substrates and products of some enzyme classes are challenging to predict from only their sequence. The authors should describe if/how their new tools will assist with filling this important knowledge gap in the Drosophila metabolism field. In addition, it would be very useful to add an additional column to Extended Table 1, indicating whether the stated enzyme activity is known by direct assay or inferred by orthology.

MINOR CRITICISMS:

1) Because of the stronger sphingolipid phenotype of nSMaseKO, it would be interesting to characterize the possible brain phenotypes in the same way as CG6962KO (Fig. 7f-h).

2) Magnifications and scale bars would be better if they were kept the same between images within the same figure (Figure 3, Figure 6, Extended Data Figure 1,4,5).

3) Introduction, lines 79-90 wrongly imply that lipidomics can only capture steady-state-profiles not dynamics. However, it is well established that lipidomics, in combination with stable isotope tracing, is a very powerful method for probing lipid dynamics. The authors

may want to mention this in the Introduction.

4) There are multiple discrepancies in the text figure citations, some of which refer to the wrong figures.

5) A key is required to show blue corresponds to CG6962KO and grey to nSMaseKO in Fig. 7a

6) In Fig.7b, the red line is not on "1".

7) In Fig.7d-h, black dots are not visible on black bars.

Referee #1:

We have carefully reviewed the revised version of the manuscript "Functional profiling and visualization of the sphingolipid metabolic network in vivo", along with the authors' response to reviewers. We appreciate the thorough effort the authors have put into addressing the previous concerns.

The authors have addressed our major concern of better resolution images, and they have made significant edits to the manuscript as suggested. Additionally, the authors have conducted many of the requested experiments and the new data significantly strengthen the manuscript.

We find the authors' responses to the reviewer comments to be detailed and satisfactory. In our assessment, the revised figures and analyses are appropriate and contribute meaningfully to the overall clarity and robustness of the work. Overall, we consider the manuscript to be in a significantly improved state and believe it presents a solid body of work with clear scientific merit.

Referee #2:

Fei-Yang Tzou et al., 205 (revised manuscript)

In the revised manuscript, the authors have streamlined the resource section and withdrawn several claims made in earlier version of the manuscript. This revision primarily focuses on the resource aspect of the study and the authors have made endeavors to deposit the data and provide access to the tools through a public resource center. Barring one major concern requiring the authors to edit the results and discussion section of the manuscript, we have no other serious objection to the publication of the manuscript.

Major concern

The surprising finding is the fact that the authors do not find robust plasma membrane staining of OlyAw. If the protein was indeed mostly secreted as the authors intended, and it was binding to CPE, we would anticipate and expect strong and universal plasma membrane staining across cells and tissues. The findings and the data provided by the authors most likely indicate that the designed OlyAw probe results in substantial fraction of the protein not folding properly during the synthesis. This is the reason the authors see a predominant ER staining {which shows a Pearson correlation coefficient of 0.4 in the control and close to 0.3 in the cpes mutant (should have been down to almost zero in the

absence of endogenous CPE, if the staining was specific)}. Generally, secreted proteins rarely show an intracellular pattern of localization unless under very defined special circumstances. In the absence of existence of data to prove specific staining of ER due to binding to CPE, it is important that statements such as "OlyAw fluorescence is mainly found in cortical ER in the L3 salivary glands" do not misguide the field to conclude that CPE is found abundantly in the ER in *Drosophila* tissues. We would not like to belabor this point further other than require the authors to unequivocally indicate such staining patterns could result from proteins undergoing misfolding during synthesis in the ER.

Referee #3:

I am impressed with the revision, and think the ms is sufficiently improved to be published.

Referee #4:

There is no doubt that the authors extensive revisions involving several additional experiments have improved the manuscript. Many of our initial criticisms have now been satisfactorily addressed, including deleting the overinterpreted section on "Glial lipoprotein mediates neuronal CerPE levels" and adding some text to the Discussion about possible reasons for the GFP versus HA discrepancy in staining. However, there remain three of our original points that do not appear to have been adequately addressed and where further experimental data are still needed:

RESPONSE 4-4:

The author stated that "OlyAw did not colocalize with Golgi markers GM130 (cis-Golgi) and Golgin245 (trans-Golgi) (Appendix Fig. S49A)" but in the quantification in Fig. S49 and S50, Golgi markers gave a relatively high Pearson's coefficient, suggesting co-localisation. The authors need to explain this clearly.

The change in OlyAw intensity with Cpes KO is minimal (Fig 6B, 6D) but much greater than with Cpes RNAi (Fig. EV5B). The authors need to explain this and provide validation that the Cpes KO line is effective.

RESPONSE 4-7:

The authors have now used two independent RNAi lines for dSMPD4 and nSMase to show

consistent changes in OlyAw intensity with neuronal or glial drivers. However, the huge difference between the Bloomington and VDRC dSMPD4 RNAi lines (Fig. 7A) suggests that the VDRC line is not effective. So, with only 1 of 2 independent RNAi lines working, that still leaves open the possibility of an off-target effect. Therefore, the previous criticism still stands: "a second independent genetic method is needed to validate all RNAi results. e.g a non-overlapping RNAi line or a mutant.". Unfortunately the same is also true for aSMase.

RESPONSE 4-10:

The authors have clarified the discrepancies between OlyAw and lipidomic experiments with the effectiveness of the RNAi checked by PCR. The PCR data provided for Reviewer #2 should be included in the supplementary data as it is important information. It would also be preferable but not essential to include OlyAw results using the KO lines for both nSMase and dSMPD4 to see if they show the same OlyAw or lipidomic trends as the RNAi lines.

Referee #1:

Tzou et al. developed a genetic toolkit by inserting a 3xHA-T2A-Gal4 at the stop
codon of genes that are implicated in sphingolipid metabolism, hereafter called HG.
This allows to monitor the expression patterns of genes of the sphingolipid pathway
components and allows elegant genetic manipulations . Using this toolkit, the
authors claim an astonishing array of cell-type specific expression patterns for
different components of the sphingolipid pathway. The authors then combine their
genetic toolkit with OlyA-mWasabi to visualize CerPE dynamics in vivo. Additionally,
the authors use the deGradHA strategy to specifically degrade HA tagged proteins in
combination with the HG toolkit to validate the cell-type specific and organelle
specific expression and function of the three CerPE catabolism enzymes: CG6962 (a
SMase), the nSMase and the aSMase. Finally, the authors dissect the biological
functions of CG6962 using a combination of knockout lines and HG toolkit.
The authors have done an enormous amount of work to create all these lines and
test their expression patterns. Unfortunately, the quality of the presented data, the
lack of co-stainings with nls-mCherry with neuronal and glial markers and the size
and resolution of the images throughout the paper do not allow to assess the
expression patterns of most genes. The manuscript is focused on presenting huge
amounts of low-quality data. It would very much benefit from reducing the number
of experiments and improving the quality of the data.

I will make suggestions to improve the quality of a few figures as this study has
merits and it would be valuable if the reagents are all made available publicly via the
BDSC or the Kyoto stock center. This should also be a prerequisite for publication.
The authors should also very much reduce the number of pictures presented in the
manuscript and setup an open data resource/website where they deposit high
quality images of expression pattern for rest of the Sphingolipid pathway
components, as is typically expected for a resource.
Indeed, this resource will allow a thorough analysis of the sphingolipid pathway in
Drosophila melanogaster. If the claims of the authors can be substantiated with high
quality data I would be strongly in favor of publication. However, given that there
are so many issues with quality of the data I will only comment on the first few
figures as the message should be obvious.

We thank the Referee for the constructive comments and for recognizing the
value of this study. We have reduced the number of images in each figure to allow
for larger views and uploaded high-resolution images of gene profiling, including
mCD8GFP, nls-mCherry, and anti-HA stainings, to Mendeley Data, which will be
published alongside the article. We have also re-examined the whole set of HG lines
with nls-mCherry in combination with markers for neurons and glia for the
quantitative documentation of gene expression in these two cell types. Finally, we
have contacted the Kyoto Drosophila Stock Center to deposit the reagents generated
in this study, including all HG lines, deGradHA, and OlyA^w transgenic flies.

Overall, the revision involves a significant overhaul to focus on the validation
of the genetic tools including the HG lines, deGradHA, and OlyA^w, and, as a proof-of-
principle example, the demonstration of cell type-specific functions of SMases in the

brain. To refine the scope and enhance the coherence of this paper, we have
removed the data on changes in the aging brain and the mechanism of lipoprotein
shuttling. We have also improved the review of current literature by including
relevant sphingolipid studies using the *Drosophila melanogaster* in the Introduction,
Results, and Discussion sections. We believe these changes significantly improve the
quality of this paper and better demonstrate the utility of the genetic toolkits to the
fly community.

Below are the point-by-point responses to the reviewer's comments.

Here are my suggestions.

1. The authors should clearly mention the number and age of animals used for
each experiment in this study. Also, they need to provide detailed genotypes for
all the different experiments in the figure legends. The genotypes of control
genotypes are not spelled out.

**Response 1-1**

We have added the numbers, age, and genotype information in the figure
legends of the revised manuscript.

2. Figure 1: The image quality in Figure 1c is very poor.

**Response 1-2**

The resolution of the original Figure 1c suffered from the small image sizes and
software auto-compression during submission. This issue is resolved in the
revised manuscript by having fewer but larger images and uploading figures
separately from the main text.

3. Extended data figure 1b: The resolution of these images is very low and no clear
pattern of HA staining can be identified. The authors could easily reduce the
number of figures by 50% or more and show high resolution images. Additional
co-stainings with other markers would be important.

**Response 1-3**

In the revised manuscript, we have reduced the number of images to allow for
larger views. We have also improved the quality of anti-HA staining images
presented in the main figure. We performed anti-Elav and anti-Repo co-
stainings to visualize the protein distribution in neurons and glia. In the New Fig.
4, we showed representative images of SPL regulators that showed either
neuronal or glial protein distribution (New Fig. 4A-B) and quantified the neuron-
to-glia ratio of anti-HA signals (New Fig. 4C).

We have moved the representative anti-HA staining images of the remaining
SPL regulatory genes to the Appendix Fig. S32-47, and the raw data of z-stacks
of anti-HA confocal images will be published at the Mendeley Data (Reserved
DOI: 10.17632/gg794sznsz.1) (Reserved DOI: 10.17632/fty6n2yjsx.1).

4. Extended data figure 2a: The images are low quality and making it difficult to
draw any reliable inferences from these images. Additionally, the authors use
only three CRIMIC lines without any explanation why only these lines were
chosen.

Response 1-4

In addition to improving the quality of the images, we agree that three CRIMIC lines were insufficient for drawing a conclusion. We have expanded our analysis by comparing 11 publicly available CRIMIC lines in our stocks, including three CRIMIC-GAL4s that have been validated in the literature (Wang et al., 2022; John et al., 2022; Chung et al., 2023). Furthermore, we quantified nls-mCherry signals colocalizing with anti-Elav and anti-Repo signals and showed consistent expression patterns of most HG and CRIMIC-GAL4 lines.

Line184-210: *“We next validated GAL4 expression patterns by comparing HG lines with two independent methods of gene expression profiling: CRIMIC-GAL4 (Lee et al, 2018) and single-cell transcriptomics of the adult brain (Davie et al, 2018). CRIMIC is a large collection of GAL4 lines that insert an artificial exon containing splice acceptor, T2A-GAL4, and a polyadenylation signal into the first coding intron of the gene (Lee et al, 2018). The CRIMIC-GAL4s thereby recapitulate gene expression by using the promotor activity while causing loss of function by arresting the transcription of targeted genes, and we obtained 11 of the CRIMIC lines in SPL genes. First, we compared the adult brain expression patterns of our HG lines with 3 genes, sk1, gba1b, and CDase, from the CRIMIC-GAL4 collection that have been validated in the literature (Wang et al, 2022b; Vaughen et al, 2022; Chung et al, 2023). For a quantitative comparison, we labeled neuronal and glial nuclei by antibodies and tallied the percentage of HG or CRIMIC-GAL4-driven nls-mCherry that colocalized with neurons and glia. Both HG and CRIMIC-GAL4 revealed glia-enriched expressions of sk1, gba1b, and CDase, despite variations in the numbers of nls-mCherry positive cells (Fig. 2C-E, Fig. EV1B-D). We further compared the 8 other CRIMIC-GAL4s with HG for cell-type expression patterns in the brain (Appendix Fig. S1). Five out of the 8 genes showed consistent cell-type expression patterns, except for lace, mdy, and sply (Appendix Table S1). HG lines of lace and sply revealed neuron-enriched expression, while the corresponding CRIMIC-GAL4 showed glial enrichment (Appendix Fig. S1F-G). In contrast, mdy HG lines showed glial enrichment, while mdy CRIMIC-GAL4 indicated higher neuronal expressions (Appendix Fig. S1H). We further compared the cell-type expression patterns of these genes with single-cell transcriptomics of the adult brain (Appendix Fig. S1I-K; <https://scope.aertslab.org/> (Davie et al, 2018)). Clustering revealed both glial and neural expression of sply and lace (broader than CRIMIC and HG lines), and strong glial enrichment of mdy (consistent with HG lines).”*

5. Extended data figure 2b: ORMDL, nSMase and GlcT show very different expression pattern before and after Cre excision. This raises significant concern about the expression pattern shown throughout the paper. To show that the difference in expression pattern is indeed a result of 3'UTRs the authors must validate the expression pattern of more genes with available CRIMICS. Additionally, the authors should clarify if they tested all the lines with or without the Cre excision and comment on the pattern of expression of the two.

Response 1-5

We have compared HG lines before and after Cre excision using nls-mCherry (Appendix Fig. S1) and mCD8GFP (Appendix Fig. S2-3). While 9 out of 11 HG

lines showed a similar number of nls-mCherry positive cells before and after Cre
excision, two genes, *ifc*, and *sply*, showed significantly different results. After
Cre excision, *ifc*-HG exhibited fewer nls-mCherry positive cells, while *sply*-HG
showed an increased number (Appendix Fig. S1). We also quantified the
percentage of nls-mCherry positive cells that colocalized with neuronal or glial
nuclei to examine the expression of the targeted gene in different cell types.
Most HG lines (9 out of 11) displayed consistent cell-type expression patterns
before and after Cre excision (Appendix Fig. S1). However, the post-Cre HG
alleles of *ifc* and *bwa* exhibited different cell-type expression patterns
compared to their pre-Cre counterparts (Appendix Fig. S1D-E). Particularly,
while the pre-Cre *ifc* HG line showed neuronal expression, the post-Cre *ifc*
exhibited glia-enriched expression. The latter is consistent with a recent report
that *ifc* is primarily expressed and required in glia (Zhu et al., 2024), suggesting
that Cre-mediated excision may be necessary for HG to recapitulate
endogenous gene expression. Furthermore, we observed a difference in
mCD8GFP expression between some pre- and post-Cre HG alleles, including
*Cpes*, *ifc*, *nSMase*, *ORMDL*, and *Glct*, showing distinct cellular projections in
various brain regions (Appendix Fig. S2-3).

The transcripts use *Hsp70Ba* 3'UTR before Cre excision and endogenous 3'UTR
after excision. However, whether the difference in expression pattern is a result
of *Hsp70Ba* versus gene-specific 3'UTRs cannot be definitely demonstrated by
comparing HG to CRIMIC-GAL4s, which use another SV40 3'UTR. Because the
regulatory role of 3'UTRs in these specific genes has not been investigated, we
have removed any speculation on their contribution to the observed expression
differences in the revised manuscript.

Since the HG lines were aimed to recapitulate gene transcription and protein
translation of targeted genes by keeping the promoter and 3'UTR intact, we
used post-Cre HG lines for subsequent gene profiling except for *schlank*, *acsl*,
*elovl*, *Spin*, *Sap-r*, and *CG11426*, whose post-Cre HG lines could not be obtained.
Although the different expression patterns between pre- and post-Cre HG
alleles are critical to the utility of the whole collection, detailed descriptions do
not meet the general interest of most readers. Therefore, in the revised
manuscript, we briefly described that we tested 11 HG lines to discover that Cre
excision changes expression patterns in some cases and that we use post-Cre
HG alleles.

Line157-164: "Given that 3'UTRs can regulate cell type-specific protein
expression (Merritt et al, 2008), we evaluated whether the *Hsp70Ba* 3'UTR
introduced with the HG cassette affects *SPL* gene expression (Fig. 1B). We
tested 11 HG lines and observed some discrepancies in a few of the HG lines
before and after Cre excision (Appendix Fig. S1-S3, Appendix Table S1).
Therefore, subsequent gene profiling used HG lines after Cre excision, where
only the endogenous 3'UTRs were present. Of the 52 HG lines, we used pre-
Cre alleles of *schlank*, *acsl*, *elovl*, *Spin*, *Sap-r*, and *CG11426* for gene profiling,
as their post-Cre alleles could not be obtained."

With that said, if the reviewer feels that detailed descriptions for the
comparison of pre- and post-Cre alleles should be clarified in the main text, we
will elaborate on the abovementioned difference in expression patterns,

including the numbers of expressing cells, cell-type enrichments, and
regionalization.

6. Extended Figure 2d: The image resolution is very low to confirm the neuronal.
vs glial expression pattern.

**Response 1-6**

We apologize for the resolution issue of the figures. In the revised manuscript,
we have reduced the number of images and added enlarged images of a single
slice from the z-stack image to better present the expression patterns (New Fig.
3A-B, Appendix Fig. S5-21).

7. Figure 2d: Adult fly brain has about 90% neurons and 10% glia. However, the
repo>mCD8GFP shows a much broader staining pattern. The nls-mCherry
stainings will solve that issue.

**Response 1-7**

We agree that nls-mCherry will provide a better resolution for this purpose. We
have replaced all panels of mCD8GFP images with nls-mCherry stainings for
documenting gene expression in specific cell types (New Fig. 3A-B, Appendix S5-
21). Furthermore, we have quantified the percentages of nls-mCherry-
expressing cells colocalizing with anti-elav or anti-repo signals in all HG lines
(New Fig. 3C). From the 52 SPL regulators, we found 21 genes exhibited neuron-
enriched patterns, and 18 genes were enriched in the glia.

8. Figure 2e-o: The image resolution is very poor. This makes it very difficult to
identify any clear expression pattern. The authors should also use alternative
markers to better visualize the patterns of expression. Additionally, add a
comparative analysis with currently known and experimentally validated
expression patterns for sphingolipid pathway components. For example, Gba1b
expression pattern shown in Wang et al., 2022 is significantly different from the
one seen in this manuscript because the authors did not use nls-mCherry

**Response 1-8**

Thank you for the insightful comment. As mentioned in Response 1-7, we have
replaced the mCD8GFP images with the nls-mCherry colocalization data for
documenting gene expressions in specific cell types (New Fig. 3). We have
moved the mCD8GFP images to the Expanded View (New Fig. EV2) and added a
neuropil staining using anti-the DLG antibody for visualizing gene expression in
different brain regions.

Additionally, as mentioned in Response 1-4, we used nls-mCherry to compare
11 pairs of HG- and CRIMIC-GAL4s, including *gba1b* (Wang et al., 2022). HG and
CRIMIC-GAL4s of *gba1b* showed consistent glia-enriched expression patterns
(Fig. EV1B).

9. Extended data figure 3: The authors indicate that the mCD8-GFP pattern
recapitulates the endogenous pattern of expression. However, the images are
low resolution and very small, making it difficult to see any clear cell-type
specific patterns. They should select about 6-10 of the most important genes
and show high resolution images and use nls-mCherry and co-label with repo

and elav.

**Response 1-9**

As described in Response 1-7 and 1-8, we have replaced the mCD8GFP images
with the nls-mCherry colocalization data for demonstrating gene expressions in
specific cell types (New Fig. 3) and moved the mCD8GFP images to the
Expanded View (New Fig. EV2) for visualizing gene expression in different brain
regions. In order to show larger images, we have now selected 2 genes to
demonstrate cell-type-specific expressions (New Fig. 3A-B) and 8 genes to show
the heterogeneity expression patterns in distinct brain regions (New Fig. EV2A)
with high-resolution images. The remaining images are presented in the
Appendix Figures and the raw data of z-stack images will be published at
Mendeley Data (Reserved DOI: 10.17632/gg794sznsz.1) (Reserved DOI:
10.17632/fty6n2yjxz.1).

10. Extended data figure 4: The glial staining is very inconsistent across the
different genes and therefore it is difficult to visualize the data. The associated
pie charts also do not match the representative images. Please provide clear
representative images for better understanding and clarity.

**Response 1-10**

We apologize for the inconsistent presentation of the glial nuclei stainings in
the single slice of the z-stack images. In the revised manuscript, we quantified
3D images of the whole-mount brain and showed that glia account for 10% of
all brain cells from 182 adult brains analyzed, which is consistent with the
literature (Raji and Potter, 2021). We now showed the representative image of
whole-mount brains for each channel and a zoom-in single slice of the z-stack
images with consistent immunostainings to demonstrate the cell type-specific
expression patterns (New Fig 3, Appendix Fig. S5-21).

11. Extended data figure 5: The age-associated reduction in *ifc*>mCD8:GFP may be
validated using anti-HA western blot at 1, 3 and 6-weeks.
Please provide raw data for the depolarization quantifications.

**Response 1-11**

We thank the reviewer for the great suggestion, which will significantly improve
the quality of the data. However, we have decided to remove the data on the
aging brain data to better focus the paper. We will incorporate the suggestion
in our future work on the role of *ifc* in the aging brain.

12. Figure 3: The GFP staining looks weaker and more confined than the HA staining.
However, GFP is driven by gene-specific Gal4 and is an amplified signal as
compared to HA staining that indicates endogenous protein levels. Please
explain these issues or repeat experiments. The authors use GFP and HA
staining patterns of *kdsr* to imply that the protein is made in glia and travel to
neurons for function. This is a significant claim and really needs to be supported
with other data.

**Response 1-12**

We thank the reviewer for highlighting the discrepancy between the GFP and
HA staining patterns in Figure 3. While it's true that GFP expression is driven by

a gene-specific GAL4 line and can amplify the signal, it's also important to note
that the anti-HA signals were also amplified through two rounds of antibody
hybridization and the biotin-streptavidin staining (rabbit-anti-HA, anti-rabbit-
biotin, streptavidin-Alexa fluorophore). We have repeated these experiments
multiple times and consistently observed the HA signal to be broader than the
GFP signal in most cases. We considered several potential explanations for this
observation, including:

- 1. Protein stability: The HA-tagged protein has a longer half-life than GAL4 and
mCD8GFP, resulting in the detection of protein where the gene was no longer
actively transcribed.
 - 2. Protein transport: The protein may be transported to non-expressing cells.
 - 3. Detection sensitivity: The transcription activity detected by the GAL4/UAS
system may be below the detection limit, while the protein staining through
antibody hybridization and the biotin-streptavidin amplification reveals a signal.

While these explanations are plausible, further investigation is needed to
elucidate the underlying mechanism. We have added a paragraph in the
Discussion section to discuss plausible cell non-autonomous activities of SPL
regulators and will explore this intriguing observation in future studies.

*Line529-540: "Furthermore, we consistently observed a broader distribution*
*of anti-HA signals compared to fluorescent markers driven by HG, which*
*raises the intriguing possibility of intercellular transport of SPL-regulating*
*enzymes within the brain. Evidently, the cause of this phenomenon is likely*
*gene-dependent, and the discrepancy may also be attributed to other factors.*
*For one, the differential stability of 3XHA-tagged proteins and fluorescent*
*markers might lead to a temporal mismatch in signal detection. Or, low levels*
*of transcription in certain cells may result in detectable protein expression, as*
*previous studies also observed the discrepancy between the levels of cognate*
*protein and mRNA molecules (Buccitelli & Selbach, 2020). Altogether, while*
*these findings could reveal novel insights into the complex regulation of SPL*
*pathways in the brain, further investigation is needed to elucidate the*
*underlying mechanisms."*

Referee #2:

Functional profiling and visualization of the sphingolipid metabolic network in vivo

Fei-Yang Tzou et al 2024

In this study the authors have identified 52 genes as direct regulators of sphingolipid
metabolism and generated knockin transgenic flies with engineered HA tag followed
by T2A self-cleaving peptide followed by a Gal4 sequence immediately proximal to
the stop codon of these genes using Crispr-Cas9 mediated genome engineering
(called HG lines). They evaluated the HA-tagging, Gal4 expression and Western
analysis of CDase as a proof of principle for demonstrating the utility of these lines.

They compared the transcriptional profiling of the HG lines utilizing Gal4 driven
mCD8-GFP fluorescence staining to evaluate expression pattern and compared it to
staining driven by previously established CRIMIC-Gal4 lines that cover about a third
of the current HG lines. They compared expression abundance with published single
cell RNA seq data. They find both similarities and differences between the various
methods of evaluation of gene expression. The SPL genes segregated across 25
regions in the brain. Based on the expression of HG line Gal4 driven mCD8GFP
reporter components of the SPT complex that catalyzes the first rate-limiting step of
the de novo sphingolipid biosynthetic pathway including the ORMDL gene is
enriched in the neurons, the immediate downstream gene was expressed
preferentially in the glia. The gene that acts immediately downstream of this
reaction in this pathway was observed to be enriched in the neurons and enrichment
of the subsequent enzyme in the pathway (dihydroceramide desaturase, infantile
crescent) in the glia. They also report differential expression between the neurons
and glia for several of the salvage pathway genes. The authors focused on the
posterior view of Calyx to visualize gene expression and protein between neuronal
cell bodies, neuropil, and different glial cells. They probed the metabolism of
ceramide phosphoethanolamine (CerPE) using a mWasabi tagged Ostreolysin A
(OlyAw) and validated its ability to bind using the cpes mutant. They report that
overexpression of OlyAw was not toxic to flies. They used this reporter to examine
cellular CerPE metabolism regulated by sphingomyelinase family of proteins. They
utilized deGradHA that targets HA tagged proteins for degradation. Neuronal
expression of deGradHA decreased CG6962 and nSMase-3xHA whereas aSMase-
3xHA was targeted by glial deGradHA, as observed by fluorescence staining. They
show increased OlyAw upon neuronal degradation of neutral sphingomyelinase
homolog and surprisingly increased neuronal OlyAw staining upon glial degradation
of the acid sphingomyelinase homolog. The authors perform several knockdown
experiments using neuronal and glial drivers to target putative lipoprotein transfer
proteins and conclude that neuronal apoLTP transported CerPE from neurons into
glia. Glial knockdown of apolpp, GLaz, MTP, apoLTP and LpR2 significantly increased
OlyAw staining.

Based on these observations' authors suggested that the de novo sphingolipid
biosynthetic pathway may not be cell autonomous in the brain. Overall, authors
performed large volume of work covering transcriptional, translational profiling of
sphingolipid metabolic enzymes and CerPE trafficking between neuron and glia, and
characterization of CG6962 knockout animals.

There is a potential that the tools developed in this study could be useful to the
Drosophila community. However, many conclusions drawn by the authors are
premature and are insufficiently demonstrated. Very many of them are inconsistent
with the known literature.

We thank the Referee for the insightful comments. We agree that
conclusions of the *de novo* synthesis and lipoprotein-mediated CerPE shuttling
necessitate additional evidence and reconciliation with the literature, but the
elucidation of these mechanisms is beyond the scope of this genetic toolkit paper.

Therefore, we have removed the speculations on the *de novo* synthesis and data on
the lipoprotein shuttling mechanism. Instead, we have added a paragraph on the cell
non-autonomous regulation of SPL metabolism in the Discussion section of the
revised manuscript and hope to address these questions in future studies. We have
also enhanced the review of current literature by including relevant sphingolipid
studies using the *Drosophila melanogaster* in the Introduction, Results, and
Discussion sections.

Overall, the revision involves a significant overhaul to focus on the validation
of genetic tools including the HG lines, deGradHA, and OlyA^w, and, as a proof-of-
principle example, the demonstration of cell type-specific functions of SMases in the
brain. We believe these revisions significantly improve the quality of this paper and
better demonstrate the utility of the genetic toolkits to the fly community.

Below are the point-by-point responses to the reviewer's comments.

Major comments.

1. Since the paper is a pathway wide analysis, it is imperative that the authors
provide comprehensive coverage of all the probes alluded to in the study. The
expression data for all 52 tagged proteins should be provided. The protein
expression data is not informative. It should at least be correlated with respect to
glia/neuronal and nuclear colocalization data. The information should be shown for
all 52 tagged proteins. It is very likely that several of the genes do not tolerate
tagging at the C-terminus and the authors would have some information about the
states of such proteins from their Western and or immunofluorescence data. It is
critical they include this negative data and discuss this in the text of the manuscript.
This will also provide consistency between the Western blot and
immunofluorescence data that is currently lacking (Extended Data Figure 1a and b).

**Response 2-1**

To correlate the patterns of gene activity and protein distribution in specific cell
types, we have presented the anti-HA immunostainings of all HG lines with the co-
staining of neuronal and glial markers (New Fig. 4, Appendix S30-45; raw data of z-
stack images will be published at the Mendeley Data (Reserved DOI:
10.17632/gg794sznsz.1).

*Line265-274: "To correlate protein distributions with HG transcript expression, we*
*quantified the intensity of the anti-HA signals surrounding neuronal or glial nuclei*
*and analyzed the neuron-to-glia ratio of anti-HA immunoreactivity (Fig. 4C). We*
*found that neuronally enriched genes, including schlank, dSMPD4, kdsr, CG33090,*
*CG30392, lace, spin, acsl, egh, cerk, SMSr, and fdl, have significantly higher*
*neuron/glia ratios of anti-HA immunoreactivity than the w¹¹¹⁸ control (Fig. 4C). In*
*contrast, the neuron/glia ratio of anti-HA immunoreactivity of the glial enriched*
*genes, including sk1 and CDase, was significantly lower than the w¹¹¹⁸ control,*
*consistent with the cell type-enriched gene expression (Fig. 3C)."*

In the revised manuscript, we have now included all results from the protein
profiling:

*Line172-175: "The anti-HA immunoblots of the total lysate of adult flies detected a*
*dominant band at the predicted molecular weight for 41 of the 52 HG flies,*

*whereas 11 lines were below the detection limit (Fig. 2A; Appendix Fig. S4; Table*
*EV1)."*

*Line262-265: "We co-stained the brain of each HG line with HA, neuronal, and glial*
*markers to visualize the protein distribution (Fig. 4A-B, Appendix Fig. S32-47). Most*
*HG lines (40 out of 52) showed significantly higher anti-HA immunoreactivity than*
*the w¹¹¹⁸ control (Table EV1)."*

**By incorporating these changes, the revised manuscript provided a more**
**comprehensive and informative presentation of the protein expression data.**

2. Based on immunostaining pattern of sphingolipid specific HG knock-in lines
authors suggested that the de novo biosynthetic pathway is split between neuron
and glia. However, this conclusion has not been substantially tested. Enrichment of
the de novo sphingolipid biosynthetic enzymes in different subset of neurons or glia
may simply mean those enzymes have different stabilities in different cell types or
may have roles other than sphingolipid biosynthesis. For instance, when examined
closely, Fig.2e and f, it appears that even subunits of same enzyme, SPT complex
(Lace, and SPT-1) that catalyze the first step of de novo sphingolipid biosynthesis are
not expressed in same neuronal cell type. This immunostaining could simply mean
that the transcriptional activities of Lace and Spt-1 subunits are different in different
subset of neurons. On a similar note, detection of IFC protein in glia more strongly
than in neurons does not necessarily mean that initial 3 steps of de novo
sphingolipid biosynthesis do not occur in glial cells. For instance, an elegant genetic
screening study by Ghosh, A et al 2013 PMID: 24348263, have shown that glial
specific knockdown of Lace, Spt-1, Schlank, and Des1 induced severe glial membrane
swelling phenotype, and this phenotype was not observed when these genes
including lace and Spt-1 were knocked down in neurons, these results clearly argue
for a cell autonomous function for sphingolipid biosynthetic pathway genes in
Drosophila glia. According to the authors' hypothesis, neuronal specific knockdown
of Lace or Spt-1 should abolish CerPE production in glia as its precursors are
originating from neurons. However, data from Ghosh et al 2013 suggest that this is
not the case. In this context, it is hard to accept authors' claim that the de novo
sphingolipid biosynthetic pathway is split between neurons and glia, especially in the
absence of substantial genetic, cell biological and biochemical evidence.

**Response 2-2**

**We thank the reviewer for the insightful comments on the plausible explanations of**
**gene profiling regarding the *de novo* synthesis. We agree with the reviewer's**
**comment that our data on neuron-glia cooperation in the *de novo* synthesis was**
**insufficient to support the conclusion. Given that multiple reviewers suggested that**
**we reduce the number of experiments to focus on the tool validation and complete**
**profiling with high-quality, coherent, and quantitative results, we have removed the**
**part regarding protein transport and neuron-glia interactions in the *de novo***
**synthesis from the Results section. We will address the mechanism and functional**
**implication of cell type-enriched distribution of the *de novo* synthesis genes in future**
**studies.**

3. Lines 152-154 - the authors indicate there are minor differences between CRIMIC-

Gal4s and HG lines. One could easily argue that there are huge differences. To
consolidate their claims that authors are advised to perform higher resolution
imaging with colocalization studies and neuronal/glial specific knockdowns to show
that both drivers are reporting expression in similar group of cells.

**Response 2-3**

We agree that quantitative analysis is necessary to compare expression patterns of
HG and CRIMIC-GAL4s. The question is also raised by Referee #1 (Response 1-4). In
the revised manuscript, we have expanded the comparison of HG and CRIMIC-GAL4s
by expressing nls-mCherry in 11 pairs of HG and CRIMIC-GAL4s and quantified the
percentage of nls-mCherry colocalized with anti-Elav-or anti-Repo signals (New Fig.
2C-E; New Fig. EV1B-D; New Appendix Fig. S1).

*Line184-210: “We next validated GAL4 expression patterns by comparing HG lines*
*with two independent methods of gene expression profiling: CRIMIC-GAL4 (Lee et*
*al, 2018) and single-cell transcriptomics of the adult brain (Davie et al, 2018).*
*CRIMIC is a large collection of GAL4 lines that insert an artificial exon containing*
*splice acceptor, T2A-GAL4, and a polyadenylation signal into the first coding intron*
*of the gene (Lee et al, 2018). The CRIMIC-GAL4s thereby recapitulate gene*
*expression by using the promotor activity while causing loss of function by*
*arresting the transcription of targeted genes, and we obtained 11 of the CRIMIC*
*lines in SPL genes. First, we compared the adult brain expression patterns of our*
*HG lines with 3 genes, sk1, gba1b, and CDase, from the CRIMIC-GAL4 collection*
*that have been validated in the literature (Wang et al, 2022b; Vaughen et al, 2022;*
*Chung et al, 2023). For a quantitative comparison, we labeled neuronal and glial*
*nuclei by antibodies and tallied the percentage of HG or CRIMIC-GAL4-driven nls-*
*mCherry that colocalized with neurons and glia. Both HG and CRIMIC-GAL4*
*revealed glia-enriched expressions of sk1, gba1b, and CDase, despite variations in*
*the numbers of nls-mCherry positive cells (Fig. 2C-E, Fig. EV1B-D). We further*
*compared the 8 other CRIMIC-GAL4s with HG for cell-type expression patterns in*
*the brain (Appendix Fig. S1). Five out of the 8 genes showed consistent cell-type*
*expression patterns, except for lace, mdy, and sply (Appendix Table S1). HG lines of*
*lace and sply revealed neuron-enriched expression, while the corresponding*
*CRIMIC-GAL4 showed glial enrichment (Appendix Fig. S1F-G). In contrast, mdy HG*
*lines showed glial enrichment, while mdy CRIMIC-GAL4 indicated higher neuronal*
*expressions (Appendix Fig. S1H). We further compared the cell-type expression*
*patterns of these genes with single-cell transcriptomics of the adult brain*
*(Appendix Fig. S1I-K; <https://scope.aertslab.org/> (Davie et al, 2018)). Clustering*
*revealed both glial and neural expression of sply and lace (broader than CRIMIC*
*and HG lines), and strong glial enrichment of mdy (consistent with HG lines).”*

4. Figure 3. In several of the panels it is difficult to recognize the neuropil. For
example, in figure 3i the Gal4 driver seems to be driving the expression in Glia while
in the same brain HA tagged protein shows a neuronal cell body pattern?

**Response 2-4**

We apologize for the confusion of cell type-specific gene expression and protein
distribution caused by the lack of markers. In Figure 3i of the submitted manuscript,
the CG6962/dSMPD4 HG line showed a neuronal expression pattern but not a glial
expression. In the revised manuscript, we use co-staining of neuronal or glial

markers to present gene expression and protein distribution in specific cell types
(New Fig. 3-4). Our data supported that *CG6962/dSMPD4* exhibited neuronal gene
expression (New Fig. 3C) and protein distribution (New Fig. 4A and 4C).

5. Line 560-561 - The authors claim despite knocking down aSMase in glia it led to
CerPE accumulations in neurons - This is a loaded sentence. The authors make no
effort to evaluate how an enzyme that acts in glia causes accumulation of CerPE in
neurons since the transport of CerPE from neurons is not compromised and thus the
authors should see increased CerPE in glia before seeing any effect on neuron and
how do neurons end up accumulating CerPE?

**Response 2-5**

We agree that the knockdown of glial aSMase should also lead to glial CerPE
accumulation. Indeed, we found that RNAi knockdown of *aSMase* in glia led to a
more significant increase in OlyA^w signals in glia (New Fig. EV6A; mean difference
4.15, $P = 0.003$) than that in neurons (New Fig. 7B; mean difference 1.90, $P < 0.001$).
Consistently, the brain lipidomic data showed CerPE accumulation upon glial
knockdown of aSMase (New Fig. EV6B). In contrast, neuronal knockdown of aSMase
had no such effect (New Fig. 7A). These data supported that CerPE homeostasis in
both neurons and glia depends on *aSMase* expression in glia. Since we did not
examine whether glial accumulation precedes that of neurons, we edited the Result
section to focus on the requirements of SMases in specific cell types:

*Line383-391: "We next examined CerPE catabolism in neurons and glia by*
*combining OlyA^w and cell type-specific RNAi knockdowns of different SMases.*
*Neuronal knockdowns of dSMPD4 using two independent RNAi lines led to a*
*significant increase in the intensity of neuronal OlyA^w (Fig. 7A), while glial*
*knockdown caused no significant changes (Fig. 7B). In contrast, RNAi knockdown*
*of aSMase in glia significantly elevated the intensity of neuronally expressed OlyA^w*
*(Fig. 7B), and the intensity of glial-derived OlyA^w was also significantly increased*
*(Fig. EV6A). Indeed, lipidomic analysis confirmed a significant increase in total*
*CerPE level in the brain upon glia-specific knockdown of aSMase (Fig. EV6B)."*

6. Recent studies by Wang et 2022 (PMID: 35857503) and Vaughen et al 2022 (PMID:
35961319) have demonstrated that GlcCer is trafficked from neurons to glia via
exosomes for lysosomal degradation in the glia. In similar lines authors in this study
suggest CerPE is also trafficked from neurons to glia via lipoprotein particles to
degrade in glial lysosomes. Although this finding is interesting, the evidence shown
in this manuscript is not strong enough to demonstrate this finding. I suggest,
overexpress UAS CPES and OlyAw in control and cpes mutant neurons and
demonstrate that CPE and OlyAw accumulates in glial specific endosomes/lysosomes
but not in neuronal specific endosomes/lysosomes.

**Response 2-6**

We thank the Referee for suggesting the experiment of Cpes overexpression and
examining the endolysosomal compartments. While we would hope to address the
reviewer's concern with experiments, this part of the data is beyond the scope of the
current project, which focused on the development and validation of the HG lines.
Therefore, after discussing with the editor, we removed the lipoprotein-mediated
CerPE shuttling from the Results section and only discussed the hypothesis in the

Discussion section of the revised manuscript. We hope to elucidate the cell non-
autonomous mechanism of CerPE catabolism in future studies.

Line508-522: *“The neural-glia transport of glucosylceramide impacts neuronal*
*proteostasis and structural plasticity (Wang et al, 2022b; Vaughen et al, 2022). It is*
*possible that a similar transport pathway exists for CerPE/SM. While in vivo*
*evidence for CerPE/SM transport between neurons and glia remains elusive, in*
*vitro studies have shown that neurons secrete SM (Chigorno et al, 2006), whereas*
*glial cells uptake exogenous SM via endocytosis and primarily degrade it within*
*lysosomes (Riboni et al, 1994). Notably, SM bound to lipoproteins is present in the*
*cerebrospinal fluid (CSF) (Pitas et al, 1987), and CSF SM levels are reduced in*
*APOE4 carriers (den Hoedt et al, 2023), suggesting a potential role for lipoproteins*
*in brain SM transport. This aligns with the reported role of lipoproteins in SM*
*transport in other tissues, such as the digestive tracts and plasma (Nilsson & Duan,*
*2006), where apoB/E lipoprotein receptor mediates endocytosis to facilitate SM*
*uptake and lysosomal degradation (Levade et al, 1991). Future studies are*
*warranted to interrogate whether lipoproteins facilitate the transport of CerPE/SM*
*between neurons and glia in the brain.”*

7. OlyAw staining in Fig.4b & c is not convincing, as it appears that general
expression of OlyAw is reduced in cpes mutants that may be independent of OlyAw
protein's ability to binding to CPE in vivo. Expression of OlyAw in Fig.4b is restricted
to neuropile compartment and excluded in cortical compartment, in contrast,
expression of OlyAw in Fig.4d & e is restricted to cortical compartment and excluded
in neuropile compartment, which cannot be explained by differences in Tubulin Gal4
and Actin Gal4, as both express ubiquitously. It appears from Fig.4e, Fig. 5h-l, and
Fig.6a & b, OlyAw protein is predominantly localized to neuronal ER or cytoplasm and
that is independent of whether it is expressed by neuronal specific Gal4 (nsyb) or
glial specific Gal4 (repo). Previous study by Bhat, H.B. et al 2015 (PMID: 26060215)
suggested that only pleurotolysin A2 (plyA2) but not OlyA bound to CPE independent
of cholesterol. This has also been validated in the cpes mutant by Kunduri et al 2022
(PMID: 36170207). In this context, it raises concerns that the in vivo OlyAw biosensor
described in this study is detecting CPE/cholesterol complexes? and not any
cholesterol enriched compartments such as rafts and ER membranes? Taken
together better in vivo validation of the OlyAw is required, I suggest, validate OlyAw
in cells that are larger in size such as spermatocytes, male accessory glands, or
salivary glands or photoreceptor neurons etc., to demonstrate OlyAw is secreted into
extracellular space and labels the plasma membrane/rafts/endocytic compartments
of same cell or neighboring cells. For instance, express in the neurons and detect
them in glia or vice versa.

**Response 2-7-1**

We thank the reviewer for the constructive feedback and suggestions on the
validation of the OlyA^w probe. As the reviewer suggested, we have validated the
Cpes-dependent localization in the ER membrane and showed that OlyA^w
fluorescence correlated with CerPE levels in L3 salivary glands.

Line 328-346: *“We then examined the subcellular distribution of OlyA^w using both*
*the brain and L3 salivary glands, which contain larger cells more amenable to*
*subcellular localization. While the CerPE synthase (Cpes) located mainly in the*

*Golgi apparatus (Vacaru et al, 2013), OlyA^w did not colocalize with Golgi markers*
*GM130 (cis-Golgi) and Golgin245 (trans-Golgi) (Appendix Fig. S49A). Instead, the*
*OlyA^w fluorescence is mainly found in cortical ER in the L3 salivary glands*
*(Appendix Fig. S49B). In both adult brains and L3 salivary glands, OlyA^w preferably*
*colocalized with the ER membrane marker calnexin over other organelles*
*(Appendix Fig. S49-50). To determine whether OlyA^w fluorescence can detect CerPE,*
*we expressed OlyA^w in control and a CRISPR-knockout line of Cpes. In Cpes-*
*knockout salivary glands, the OlyA^w fluorescence became diffuse (Fig. 6B), and its*
*colocalization with the ER membrane marker was significantly reduced (Fig. 6C).*
*The fluorescent intensity of OlyA^w on the ER membrane also significantly decreased*
*in Cpes-knockout salivary glands (Fig. 6D). Moreover, knockdown of Cpes also*
*reduced OlyA^w intensity in the salivary gland (Fig. EV5B), while overexpressing*
*Cpes significantly increased OlyA^w intensity (Fig. EV5C). In addition, aSMase*
*knockdown led to a drastic increase in the OlyA^w signal (Fig. EV5D). Importantly,*
*overexpression of Cpes also significantly increased OlyA^w intensity in the adult*
*brain (Fig. EV5E). These data demonstrate that OlyA^w fluorescence intensity*
*correlates with CerPE levels.”*

Moreover, in adult photoreceptors and brains, we showed that OlyA^w primarily
labeled the expressing cells even though it may be secreted.

*Line347-359: “Because OlyA^w contains a secretory peptide, we utilized the unique*
*organization of the Drosophila compound eyes to test if the OlyA^w probe is*
*secreted into extracellular spaces and can label neighboring cells. In the cross-*
*section view of the compound eye, seven photoreceptors can be seen in each*
*ommatidium, with pigment glia in the inter-ommatidial space (Fig. 6E). When*
*driven by the pigment glia-specific 54C-GAL4, OlyA^w fluorescence primarily labeled*
*the interommatidial space, with only a few OlyA^w puncta in the photoreceptors*
*(Fig. 6F, arrowhead). Conversely, when using the photoreceptor-specific rh1-GAL4,*
*most OlyA^w signals were observed in photoreceptors with some puncta in the*
*interommatidial space (Fig. 6G, arrowhead). In the adult brain, OlyA^w labels the*
*glial nuclei and membrane processes using a glial driver, while it labels neurons in*
*the cortex when using the neuronal driver (Fig. EV5F). These results showed that*
*OlyA^w predominantly labels the expressing cells even though it may potentially be*
*secreted.”*

In addition to the discrepancies outlined above, the authors should have seen a
generalized OlyA^w staining even in cpes mutant flies since the fluorescent probe is
being synthesized and secreted from all cells that the Gal4 is expressed in.

**Response 2-7-2**

Indeed, we observed a diffuse signal of OlyA^w in the L3 salivary gland of Cpes
mutants. This is spelled out in the revised Results section:

*Line337-339: “In Cpes-knockout salivary glands, the OlyA^w fluorescence became*
*diffuse (Fig. 6B), and its colocalization with the ER membrane marker was*
*significantly reduced (Fig. 6C).”*

In addition, this reviewer is surprised that the authors have not observed intense
Golgi staining in all cells of the control flies. Secreted proteins transit through the
lumen of Golgi while being secreted out. Luminal leaflet is the major site of CPE

synthesis. Thus, fluorescence in Golgi should be consistent across all cells
irrespective of the state of CerPE degradation. The authors should examine cells
from several tissues to examine Golgi staining with Wasabi and examine the pattern
and document it in the manuscript.

**Response 2-7-3**

Indeed, Golgi staining is an important aspect of OlyA^w validation. Therefore, we
examined OlyA^w localization in the adult brain and L3 larval salivary glands. Our
observations consistently revealed low colocalization of OlyA^w with the *cis*-Golgi
marker GM130 and the *trans*-Golgi marker Golgin245 (New Appendix Fig. S49A).
Instead, we observed significant colocalization with the ER membrane marker
Calnexin (New Appendix Fig. S49B).

*Line328-335: "We then examined the subcellular distribution of OlyA^w using both*
*the brain and L3 salivary glands, which contain larger cells more amenable to*
*subcellular localization. While the CerPE synthase (Cpes) located mainly in the*
*Golgi apparatus (Vacaru et al, 2013), OlyA^w did not colocalize with Golgi markers*
*GM130 (cis-Golgi) and Golgin245 (trans-Golgi) (Appendix Fig. S49A). Instead, the*
*OlyA^w fluorescence is mainly found in cortical ER in the L3 salivary glands*
*(Appendix Fig. S49B). In both adult brains and L3 salivary glands, OlyA^w preferably*
*colocalized with the ER membrane marker calnexin over other organelles*
*(Appendix Fig. S49-50)."*

8. Previous studies from the Suzanne Eaton laboratory have demonstrated that
almost all circulating apoLTP, apoLPP and largely MTP is synthesized by the fat body.
Brankatsch M et al., 2014 (PMID 25275323) showed that neuronal LTP was
exclusively derived by fat body generated LTP that crossed the blood brain barrier.
This reviewer is very intrigued by the claim by the authors that apoLTP is expressed
in the neuron and apoLPP is expressed in the glia. Also, Yin et al., (PMID: 33893307)
study that the authors cite clearly show that astrocyte specific knockdown of apoLTP
and LPP did not work for them (as they are not expressed by any cells in the brain,
supplementary data 2). Since the current data contradicts a previously well-
established finding, the authors will have to provide decisive proof other than CerPE
binding of OlyA^w to establish beyond reasonable doubt that these two lipoproteins
are indeed expressed in neurons and glia.

Also, they need to provide why and how do the apoLTP and apoLPP that will be still
in abundance after neuron and glia specific knockdown (since fat body is the major
site for synthesis and secretion of these two lipoproteins) are unable to assist in
transfer of the sphingolipids.

**Response 2-8**

We understand that the cellular expression and function of lipoproteins in the brain
should be carefully examined to reveal the mechanism of CerPE shuttling. However,
as mentioned in Response 2-6, considering the scope of the revised manuscript, we
decided to remove the section on lipoprotein-mediated CerPE shuttling after
discussing it with the editor. We agree that it is imperative to address this issue in
future investigations of the mechanism. We have added a paragraph to discuss the
cell non-autonomous regulation of SPL metabolism in the Discussion section of the
revised manuscript.

Line508-522: “The neural-glia transport of glucosylceramide impacts neuronal
proteostasis and structural plasticity (Wang et al, 2022b; Vaughen et al, 2022). It is
possible that a similar transport pathway exists for CerPE/SM. While in vivo
evidence for CerPE/SM transport between neurons and glia remains elusive, in
vitro studies have shown that neurons secrete SM (Chigorno et al, 2006), whereas
glial cells uptake exogenous SM via endocytosis and primarily degrade it within
lysosomes (Riboni et al, 1994). Notably, SM bound to lipoproteins is present in the
cerebrospinal fluid (CSF) (Pitas et al, 1987), and CSF SM levels are reduced in
APOE4 carriers (den Hoedt et al, 2023), suggesting a potential role for lipoproteins
in brain SM transport. This aligns with the reported role of lipoproteins in SM
transport in other tissues, such as the digestive tracts and plasma (Nilsson & Duan,
2006), where apoB/E lipoprotein receptor mediates endocytosis to facilitate SM
uptake and lysosomal degradation (Levade et al, 1991). Future studies are
warranted to interrogate whether lipoproteins facilitate the transport of CerPE/SM
between neurons and glia in the brain.”

Minor comments:

1. The authors have been generally agnostic of previous studies on sphingolipid
metabolic pathway genes in *Drosophila*. While tangential references have been
made in many instances, a proper contextual reference should be made about these
studies either in the result sections or in the discussion.

**Response 2-10**

We apologize for the insufficient referencing of studies on sphingolipid metabolism
in *Drosophila*. In the revised manuscript, we have added references to the
Introduction, Results, and Discussion sections wherein important findings of fly
sphingolipid metabolism should be mentioned.

2. Please include SPT protein staining in Fig.3.

**Response 2-11**

To show images of high resolution in large sizes, we have reduced the number of s
anti-HA staining images presented in the main figure (New Fig. 4). The remaining
anti-HA staining images are presented in the Appendix figures. The SPT protein
staining is shown in the New Appendix Fig. S45.

3. Since several laboratories have generated *cpes* mutants and UAS-CPES flies, the
source of the mutant and transgenic should be cited in the protocol section of the
manuscript.

**Response 2-12**

We apologize for the not disclosing the origin of the *Cpes* mutants and transgenes in
the initial submission. The CRISPR-engineered *Cpes*^{KO} and the *UAS-Cpes* transgenic
flies were generated in our lab. The cloning and transgenesis of these two reagents
are now included in the Methods section of the revised manuscript.

4. Fig.7f-h. Please add neuronal specific degradation of CG6962 to compare with
knockout model.

**Response 2-13**

We agree that comparing neuronal-specific degradation to knockout phenotypes

would strongly support the cell type-specific role of *CG6962/dSMPD4* in
neurophysiology. Since the genetic background for neuron-specific degradation
requires homozygous *CG6962/dSMPD4* HG alleles, a nSyb-LexA, and a LexAop-
deGradHA, it is very difficult to obtain sufficient flies for a single batch of behavior
assays (Figure 7f-g in the original manuscript). Given the limitation in time and
resources, we have prioritized tool validation and comprehensive SPL brain profiling
during the revision. As an alternative approach, we performed RNAi knockdown of
*CG6962/dSMPD4* using neuronal or glial drivers and found that neither were
sufficient to induce feeding abnormalities or learning defects (Figure to Reviewers. 1).
The discrepancy between *CG6962/dSMPD4*^{KO} and RNAi knockdown may be due to
incomplete reduction of gene product.

*Figure for referee with unpublished data has been removed upon request by
the authors.*

5. Lines 284-285 - Therefore, intercellular transport of dihydroceramide from neuron
to glia is required for ceramide de novo synthesis initiated in neurons to be
completed. Such statements lack scholarship. This needs to be validated by showing
knockdown of Schlank in neuron results in lack of CerPE in the glia.

**Response 2-14**

We agree that the conclusion of the *de novo* synthesis components was not
supported by sufficient evidence. We have removed the speculations regarding *de*
*novo* synthesis from the revised manuscript.

6. Line 385-386 - the authors write 'Accumulation of p62 protein aggregate is a

hallmark of SPL accumulation in Lysosomes¹ - There is no evidence for this statement.
Kinghorn et al., in their 2016 manuscript show that p62, are markers of lysosomal-
autophagic degradation, linking ubiquitinated proteins to the autophagy machinery.
If autophagy is inhibited, then p62 accumulates. In some specific instance of
trafficking defects p62 accumulates as suggested in the three manuscripts that the
author cites. But it is no indicator of sphingolipid or CerPE accumulation as the
authors claim.

**Response 2-15**

We agree with the reviewer that p62 accumulation is a marker of lysosomal
degradation rather than a specific indicator of sphingolipid accumulation in
lysosomes. We have edited the text to emphasize that dysregulation of sphingolipids
induces lysosomal defects (Tang et al., 2022), thus results in the accumulation of p62,
such as the case of p62 protein accumulation in the *Drosophila* model of Gaucher
disease (Kinghorn et al., 2016; Vaughen et al., 2022). In the revised manuscript, we
have rephrased the rationale of the anti-p62 staining.

*Line442-446: "We investigated whether ref(2)P/p62, an autophagy adaptor, was*
*affected, as p62 accumulation was reported following SPL accumulation in the*
*contexts of lysosome storage diseases (Vaughen et al, 2022; Kinghorn et al, 2016;*
*Davis et al, 2016). We showed that glial knockout of the aSMase led to p62*
*aggregates, while neuronal knockout had no such effect (Fig. EV6C-D)."*

7. A more convincing experiment would be to test memory deficit and abnormal
feeding behavior in neuronal vs glial degradHA flies and validate the specific role in
neurons.

**Response 2-16**

As described in Response 2-13, we performed RNAi knockdown experiments and
obtained negative results, likely due to incomplete reduction of gene product.

**Referee #3:**

Summary:

In this study, the authors generated tools to visualise the expression of enzymes
involved in sphingolipid metabolism in the CNS. Using these GAL4 lines and a
SM/CerPE biosensor generated in this paper, the authors showed that SM is shuttled
between neurons and glia via apolipoproteins for degradation. They also discovered
that CG6962 is autonomously required in neurons to sustain brain function. In
general, the tools generated in this paper are valuable resources for the community.
However, they need to be more carefully validated. In many cases, the images did
not clearly support the conclusions made in the results section. Furthermore, the
introduction part only addresses the technical hurdle for studying sphingolipid
metabolism, without providing sufficient background on what has been known about
sphingolipid metabolism in *Drosophila* brains. The way the manuscript is written
needs to be improved. The authors should have linked the 4 parts (GAL4 tools,
OlyAw biosensor, Glial neuron SM shuttle and CG6962 function) better, so that it is

more coherent.

We thank the referee for the insightful comments. Overall, the revision
involves a significant overhaul to focus on the validation of genetic tools, including
the HG lines, deGradHA, and OlyA^w, and, as a proof-of-principle example, the
demonstration of cell type-specific functions of SMases in the brain. To refine the
scope and enhance the coherence of this paper, we have removed the data on
changes in the aging brain and the mechanism of lipoprotein shuttling. We have also
enhanced the review of current literature by including relevant sphingolipid studies
using the *Drosophila melanogaster* in the Introduction, Results, and Discussion
sections. We believe these revisions significantly improve the quality of this paper
and better demonstrate the utility of the genetic toolkits to the fly community.

Below are the point-by-point responses to the reviewer's comments.

Major Comments:

The HG Knock in animals of SPL metabolism genes reveal endogenous gene
expression and protein localization

Are the proteins tagged with 3XHA functional? With CDase-3XHA-T2-GAL4, do you
get homozygotes? Are they embryonic lethal?

**Response 3-1**

We acknowledge that some proteins may not tolerate the 3XHA tag at the C-
terminus. As the reviewer suggested, we now provide the homozygous
viability/lethality of all HG lines. In the revised manuscript, every HG line was
backcrossed to the *w¹¹¹⁸* strain for three generations before the Cre excision, and the
homozygous viability of the isogenized and Post-Cre HG lines is stated in the Results
section and documented in Table EV1. We found that 47 out of the 52 HG alleles
were homozygously viable, including *CDase*.

134-135

Are the patterning of 3XHA-tagged sphingolipid regulators shown in extended data
Fig. 1b matched with what have been shown previously with antibody staining? I
would like to see how the staining using antibody against the protein colocalises with
the 3XHA tagged one in a heterozygote.

**Response 3-2**

We agree that 3XHA-tagging at the C-terminus may alter the subcellular localization
of targeted proteins. To address this, we compared the localization of the 3XHA-
tagged protein in heterozygous *ifc*-HG and *schlank*-HG lines to the antibody staining
results from the literature, as examples of endogenous protein labeling. We found
that the 3XHA-tagged *Ifc* and *Schlank* showed consistent patterns as that of the
endogenous protein.

Line175-183: "To examine whether the localization of the 3XHA-tagged protein in
HG lines resembles that of the endogenous protein, we compared the anti-HA
immunostainings to antibody staining results that have been reported in the
literature. We showed the colocalization (Pearson's $R^2 = 0.65$) of anti-HA and anti-
*Ifc* immunofluorescence in the L3 salivary glands of the heterozygote of the *ifc*-HG

*line (Fig. 2B). Moreover, anti-HA immunostaining of the schlank-HG revealed both*
*ER and nuclear localizations, consistent with the known subcellular distribution of*
*Schlank (Bauer et al, 2009; Sociale et al, 2018) (Fig. EV1A). These data indicate*
*that HG knockins can reveal endogenous protein localization.”*

Extended data Fig 2b. What about glia? Does Cre excision changed the expression
pattern in glia and how's that compared to the CRIMIC as you showed in a?

**Response 3-3**

We agree that Cre excision may affect gene expression in both neuronal and glial
cells. In the revised manuscript, we used nls-mCherry with co-stainings of neuronal
or glial markers and quantified the percentage nls-mCherry colocalized with either
neuronal or glial markers. We then compared the HG lines before and after Cre
excision with 11 publicly available CRIMIC-GAL4s, including 5 glia-enriched genes, *sk1*,
*gba1b*, *CDase*, *aSMase*, and *ect3* (New Fig. EV1B-D; New Appendix Fig. S1). Pre- and
post-Cre HGs and CRIMIC-GAL4s showed consistent glia-enrichment, indicating that
the expression patterns are not changed by Cre excision. In addition, we found that
cellular projections of glia-enriched genes remained similar before and after Cre
excision using mCD8GFP (New Appendix Fig. S2). Together, both nls-mCherry and
mCD8GFP data indicated that Cre excision does not change glial expressions of HG,
and CRIMIC-GAL4s showed consistent results of glia-enrichment.

The HG Knockin animals revealed heterogeneous gene expression of the SPL
metabolic network in the brain

197-201 Fig 2. Can you explain how you aligned your 3D confocal images to the brain
wiring map? With the heatmap, did you normalise the GFP intensity to the area of
each neuropil region based on the anti-DLG staining? If so, I think you should show
the DLG staining of the same focal plane next to your image showing GFP pattern.

**Response 3-4**

For the brain warping, we align the anti-DLG signal of each whole-mount brain image
to the anti-DLG signal of the reference brain. Each 3D brain image of mCD8GFP
driven by HG-GAL4, was transformed by the Computational Morphometry Toolkit
(CMTK)(Rohlfing & Maurer, 2003) based on the anti-DLG alignment, including
transformation includes translation, rotation, and anisotropic scaling.

We did not normalize the GFP intensity to anti-DLG staining. The heatmaps were
plotted based on mCD8GFP intensity of the target gene in each neuropil normalized
to the mean of the target gene in all neuropils.

In the revised manuscript, we further explain the brain warping technique in the
Results section and the method detailed in the Methods sections. Additionally, we
showed the DLG staining of the same focal plane next to the image showing the GFP
pattern (New Fig. EV2).

d-h, to claim a certain gene is expressed in neurons or glia, you have to label neuron
or glia in the same experiment where you drove mCD8GFP using specific HG. elavQF,
repoQF and QUAS-RFP are all commercially available.

**Response 3-5**

We agree that neuronal and glial markers are required to claim cell-type expression
patterns. As also suggested by Referee #1, we have performed the nls-mCherry
stainings of all HG lines with neuronal and glial markers. We have quantified the
percentage of nls-mCherry colocalizing with neuronal or glial markers (New Fig. 3C).
In the revised manuscript, the conclusion of cell-type expressions is made from nls-
mCherry stainings with neuronal and glial markers (New Fig. 3), while the mCD8GFP
images are only used to describe regional expression profiles (New Fig. EV2).

Extended Fig 4. You quantified what percentage of mcherry+ cells are glial cells and
what percentage are neurons. First, I am not sure what area of the brain you showed.
Second, it makes more sense to me if you quantify what percentage of neurons is
mcherry+ and what percentage of glial cells is mcherry+. E.g. for ORMDL, on the
surface of the brain, XX% of glia is mcherry+ . In CVLP neuropil region (based on DLG
staining), XXX% of neurons is mcherry+. And other regions... You also need to show
statistics (n=3).

**Response 3-6**

We apologize for the missing details on data quantifications. In the original
manuscript, we show the posterior view of the calyx. In the revised manuscript, we
quantified the number of mCherry+ cells and the percentages of colocalization with
neuronal and glial markers using the whole-mount brain images (New Fig. 3).
Additionally, we quantified the percentage of mCherry+ neurons and glial cells as the
reviewer suggested. The quantification details are described in the Methods section
and the quantification of each HG line is reported in the Source Data of Figure 3 in
the revised manuscript.

226 I am not convinced by your statement that *ifc* exhibited a glia-enriched pattern.

**Response 3-7**

We agree that mCD8GFP expression is not convincing for examining gene expression
in specific cell types. As mentioned in Response 3-6, we acquired cell type expression
data from whole-mount brain using HG driven nls-mCherry and co-stainings of
neuronal and glial markers (New Fig. 3). We found that 35.7% of nls-Cherry driven by
*ifc*-HG colocalized to the glial marker (New Fig.3C), which is significantly higher ($P <$
0.001) than the percentage of glia (approximately 10 %) in total brain cells. Therefore,
we concluded that *ifc* exhibits a glia-enriched expression pattern. Moreover, the glial
expression of *ifc* is consistent with a recent report showing that *ifc* functions in glia
to promote neuronal survival (Zhu et al., 2024; Tzou et al., 2025).

235-236 where's the image for GBA2?

**Response 3-8**

We apologize for the mislabeling of Figure 2k-l in the text. In the original manuscript,
Figure 2k should be *gba1b/GBA1*, whereas *CG33090/GBA2* should be Figure 2l. The
*CG33090* profiling data is now moved to the Appendix Figures (nls-mCherry,
Appendix Fig. S10; mCD8GFP, Appendix Fig. S25; anti-HA, Appendix Fig. S36)

245: What do you mean by "These findings suggested that SPL catabolism occurs in
distinct subcellular compartments in different cell types"? I haven't seen any data
showing these enzymes are expressed in specific subcellular compartments.

**Response 3-9**

We apologize for the overstatement in our conclusion due to a jumping logic.
Previous studies have characterized the subcellular localization of enzymes in the
salvage pathway using cell cultures, and in Figure 2 of the original manuscript, we
showed that the expression of these enzymes was enriched in certain cell types.
Collectively, it could be suggested that “SPL catabolism occurs in distinct subcellular
compartments in different cell types”. This conclusion is not justified by the data
presented in Figure 2 of the original manuscript. In the revised manuscript, we have
toned down our description by focusing on the subcellular localization of SMases
with supported by experimental results in the New Fig. 7F-G and New Appendix Fig.
S54 of the revised manuscript.

*Line414-418: “We further characterize the subcellular distribution of SMases in the*
*brain by co-staining 3XHA-tagged proteins with organelle markers in HG lines. We*
*found that nSMase localized to the mitochondria, aSMase was partially associated*
*with lysosomes (Appendix Fig. S54), and dSMPD4 was present on the nuclear*
*membrane (Fig. 7F-G).”*

246-258 I am not convinced that the expression patterns of these genes remain
unchanged except for ifc. Lace, nSmase, Kdsr, gba1b, Schlank and ifc all looked
different upon aging.

**Response 3-10**

We thank the reviewer for the observation regarding the potential changes in the
expression patterns of these genes during aging. This is one of the advantages of
generating these HG lines and making them available to the public. However, a more
rigorous quantitative analysis would be necessary to draw meaningful conclusions.
With the refining of the scope of the manuscript on tool validation and the core
findings related to SMases, we have removed the section on expression profiling of
aging brains after discussing it with the editor. We think this decision aligns with the
suggestions from Referee #3 to improve the coherence of the paper and that from
Referees #1 & #4 to reduce the number of experiments and focus on tool validation.
We hope to investigate age-related changes in brain sphingolipid metabolism in
future studies.

Immunostaining of the endogenous 3xHA-tagged proteins revealed cell non-
autonomous

regulation of the SPL de novo synthesis and catabolism

268-299: I am not convinced by Fig. 3, because glial and neuronal markers are lacking.

**Response 3-11**

We agree that the data would be more convincing with glial and neuronal markers.

In the revised manuscript, we repeated anti-HA staining of all HG lines with anti-elav
and anti-repo co-stainings to visualize the HA-tagged proteins and conducted a
quantitative analysis of the protein distribution in neurons and glia (New Fig. 4).

*Line262-274: “We co-stained the brain of each HG line with HA, neuronal, and glial*
*markers to visualize the protein distribution (Fig. 4A-B, Appendix Fig. S32-47). Most*
*HG lines (40 out of 52) showed significantly higher anti-HA immunoreactivity than*
*the w¹¹¹⁸ control (Table EV1). To correlate protein distributions with HG transcript*
*expression, we quantified the intensity of the anti-HA signals surrounding neuronal*

*or glial nuclei and analyzed the neuron-to-glia ratio of anti-HA immunoreactivity*
*(Fig. 4C). We found that neuronally enriched genes, including schlank, dSMPD4,*
*kdsr, CG33090, CG30392, lace, spin, acsl, egh, cerk, SMSr, and fdl, have*
*significantly higher neuron/glia ratios of anti-HA immunoreactivity than the w^{1118}*
*control (Fig. 4C). In contrast, the neuron/glia ratio of anti-HA immunoreactivity of*
*the glial enriched genes, including sk1 and CDase, was significantly lower than the*
*w^{1118} control, consistent with the cell type-enriched gene expression (Fig. 3C)."*

SM/CerPE homeostasis is maintained by neuronal neutral SMase and glial acidic
SMase

382 Fig. 5F Most of aSMase-3XHA did not colocalise with anti-Cathepsin L. So, you
can only claim some aSMase localised to lysosome.

**Response 3-12**

Indeed, the majority of the aSMase-3XHA signals did not colocalize with anti-
Cathepsin L. Therefore, we toned down our conclusion in the revised manuscript.

Line414-418: *"We further characterize the subcellular distribution of SMases in the*
*brain by co-staining 3XHA-tagged proteins with organelle markers in HG lines. We*
*found that nSMase localized to the mitochondria, aSMase was partially associated*
*with lysosomes (Appendix Fig. S54), and dSMPD4 was present on the nuclear*
*membrane (Fig. 7F-G)."*

In what subtype of glia does aSMase knockdown causes an increase of OlyA^w?

**Response 3-13**

While pinpointing the subtypes of glia where aSMase controls brain CerPE
degradation is an important task, we believe it is beyond the scope of this study. In
the manuscript, we have compared the function of SMases in neurons versus glia by
profiling the mRNA expression, protein distribution, and their regulation in CerPE
catabolism. We have shown that OlyA^w indicates a CerPE levels in cell type-specific
manner (New Fig. 6F-G, Fig. EV5F). Pan-glial knockdown of aSMase led to increased
OlyA^w in both neurons and glia (New Fig. 7B, Fig. EV6A). Moreover, lipidomic data
showed that glial aSMase knockdown significantly elevated CerPE levels in the brain
(New Fig. EV6B). These findings strongly support that aSMase is required in glia to
maintain brain CerPE catabolism. We hope to address the functions and mechanisms
of SMases in glia/neuron subtypes in future studies.

Glial lipoprotein mediates neuronal CerPE levels

412-415: Where do you show glial knockdown of aSMase led to neuronal
accumulation of CerPE? In fig. 5 I, you only showed upregulation of OlyA in the brain,
there's no neuronal marker in the image.

**Response 3-14**

We repeated the experiments with co-staining of neuronal and glial markers (New
Fig. 7). Moreover, we further validated in the adult brain and eyes that OlyA^w
predominantly located in the cells where it was expressed, thus allowing for cell
type-specific labeling when driven by cell type-specific GAL4 drivers.

Line347-359: *"Because OlyA^w contains a secretory peptide, we utilized the unique*
*organization of the Drosophila compound eyes to test if the OlyA^w probe is*

secreted into extracellular spaces and can label neighboring cells. In the cross-
section view of the compound eye, seven photoreceptors can be seen in each
ommatidium, with pigment glia in the inter-ommatidial space (Fig. 6E). When
driven by the pigment glia-specific 54C-GAL4, OlyA^w fluorescence primarily labeled
the interommatidial space, with only a few OlyA^w puncta in the photoreceptors
(Fig. 6F, arrowhead). Conversely, when using the photoreceptor-specific *rh1*-GAL4,
most OlyA^w signals were observed in photoreceptors with some puncta in the
interommatidial space (Fig. 6G, arrowhead). In the adult brain, OlyA^w labels the
glial nuclei and membrane processes using a glial driver, while it labels neurons in
the cortex when using the neuronal driver (Fig. EV5F). These results showed that
OlyA^w predominantly labels the expressing cells even though it may potentially be
secreted.”

In addition, we have added multiple new pieces of evidence that the fluorescence
intensity of OlyA^w correlated with CerPE levels. Therefore, the increased intensity of
neuronal GAL4-driven OlyA^w can be used to indicate increased CerPE levels in
neurons.

*Line335-346: “To determine whether OlyA^w fluorescence can detect CerPE, we*
*expressed OlyA^w in control and a CRISPR-knockout line of Cpes. In Cpes-knockout*
*salivary glands, the OlyA^w fluorescence became diffuse (Fig. 6B), and its*
*colocalization with the ER membrane marker was significantly reduced (Fig. 6C).*
*The fluorescent intensity of OlyA^w on the ER membrane also significantly decreased*
*in Cpes-knockout salivary glands (Fig. 6D). Moreover, knockdown of Cpes also*
*reduced OlyA^w intensity in the salivary gland (Fig. EV5B), while overexpressing*
*Cpes significantly increased OlyA^w intensity (Fig. EV5C). In addition, aSMase*
*knockdown led to a drastic increase in the OlyA^w signal (Fig. EV5D). Importantly,*
*overexpression of Cpes also significantly increased OlyA^w intensity in the adult*
*brain (Fig. EV5E). These data demonstrate that OlyA^w fluorescence intensity*
*correlates with CerPE levels.”*

Also, does aSMase knockdown in glia cause neurodegeneration? e.g. does the animal
die eventually or have some behavioural defects?

**Response 3-15**

We examined the locomotor activity of adult flies with conditional CRISPR knockouts
in neurons or glia. Our results showed that knocking out *aSMase* in glia but not
neurons caused climbing defects, indicating the *aSMase* is required in glia to
maintain neuronal functions (New Fig. EV6E). We did not examine whether *aSMase*
knockout in glia affects lifespan.

Minor comments:

The HG Knock in animals of SPL metabolism genes reveal endogenous gene
expression and protein localization

Extended data Fig 2d. You need to quantify what percentage of mcherry+ cells are
positive for glial marker and neuron marker. Also, it's good to add a zoom-in image
showing there's indeed no colocalization between glial marker or neuronal marker
and mcherry.

**Response 3-16**

We have quantified the percentage of mCherry+ cells colocalizing with neuronal or
glial markers in all HG lines (New Fig. 3C). Furthermore, we have added enlarged
images of the co-staining results to demonstrate the colocalization of mCherry and
the cell type markers (New Fig. 3A-B).

160. I thought you are comparing your data with public single cell transcriptome?
Not your transcriptional profiling, right? Or you have this data generated in your lab?

**Response 3-17**

Yes, we used publicly available data from a single-cell transcriptome study of the
adult fly brain (Davie et al., 2018). We originally cited this reference in the Result
section and Figure Captions, and now we have also cited the reference on the Figure
(New Fig. 2F).

161-162. What are these genes with cell type-specific expression pattern that were
not observed in the transcriptome? Can you include those in Extended Data Fig. 2e
or as a supplemental table?

**Response 3-18**

In the original manuscript, we categorized the cell type-specific expression from
HG>mCD8GFP and the single-cell transcriptome (Davie et al., 2018) by visual
inspection of the confocal images and t-SNE plots, respectively. We recognized that
quantitative analysis is required for drawing convincing conclusions, so we have
repeated the whole set of experiments with nls-mCherry stainings for documenting
gene expression in specific cell types (New Fig. 3A-B, Appendix S5-21). We quantified
the percentages of nls-mCherry-expressing cells colocalizing with anti-elav or anti-
repo signals in all HG lines (New Fig. 3C). From the 52 SPL regulators, we found 21
genes exhibited neuron-enriched patterns, and 18 genes were enriched in the glia
(New Fig. 3C). The cell type expression pattern of each gene is shown on the New Fig.
3C and summarized in Table EV1.

The HG Knockin animals revealed heterogeneous gene expression of the SPL
metabolic network in the brain

220 The allosteric inhibitor for the SPT complex, showed neuron-specific expression
(Fig. 2e-f). This should be Fig. 2g. Kdsr ...(Fig. 2h) These figure images are cited wrong
in your result session. Reference is needed when you explain the function of each
enzyme.

231. Fig 2j-p should be K-0

**Response 3-19**

We apologize for the error. Since we have revised the manuscript extensively, we
have reorganized the figures and made sure each panel is labeled correctly.
Moreover, we have added references for the enzyme functions in the Table EV1 of
the revised manuscript.

255-258: Can you explain how you measured the neuron function in the text?
Extended Data fig. 5C. What is the sample size of ifcKO ? Is ifcKO mutant or
mushroom body specific knockout?

**Response 3-20**

We measured the depolarization of the *ifc*-KO whole-eye clone using
electroretinograms as an indicator of the light-sensing function of photoreceptors.
However, as explained in Response 3-10, we have removed the data regarding the
effect of aging in the revised manuscript after discussion with the editor.

The Ostreolysin A transgene associates with membrane rafts and indicates the
abundance

and dynamics of CerPE

322-325: Fig4 b-e: Please include quantification.

Fig 4 f-h: Please include quantification (colocalization analysis).

Extended Data Fig. 6 b: p value?

335-337: what's your conclusion? Is CerPE more enriched in nuclear membrane and
ER?

**Response 3-21**

We agree that quantitative analysis is necessary for the validation of the OlyA^W
probe. In the revised manuscript, we now provide a quantitative analysis of OlyA^W
pattern changes in *Cpes*^{KO} (New Fig. 6B-C), OlyA^W subcellular distributions (New
Appendix Fig. S49-50), and correlation of OlyA^W fluorescence with CerPE levels (New
Fig. EV5A-D). We conclude that OlyA^W is mostly distributed to the ER membrane,
while the punctate signals in the brain likely indicate CerPE-enriched lipid rafts.

*Line328-346: "We then examined the subcellular distribution of OlyA^W using
both the brain and L3 salivary glands, which contain larger cells more amenable
to subcellular localization. While the CerPE synthase (Cpes) located mainly in the
Golgi apparatus (Vacaru et al, 2013), OlyA^W did not colocalize with Golgi
markers GM130 (cis-Golgi) and Golgin245 (trans-Golgi) (Appendix Fig. S49A).
Instead, the OlyA^W fluorescence is mainly found in cortical ER in the L3 salivary
glands (Appendix Fig. S49B). In both adult brains and L3 salivary glands, OlyA^W
preferably colocalized with the ER membrane marker calnexin over other
organelles (Appendix Fig. S49-50). To determine whether OlyA^W fluorescence can
detect CerPE, we expressed OlyA^W in control and a CRISPR-knockout line of Cpes.
In Cpes-knockout salivary glands, the OlyA^W fluorescence became diffuse (Fig.
6B), and its colocalization with the ER membrane marker was significantly
reduced (Fig. 6C). The fluorescent intensity of OlyA^W on the ER membrane also
significantly decreased in Cpes-knockout salivary glands (Fig. 6D). Moreover,
knockdown of Cpes also reduced OlyA^W intensity in the salivary gland (Fig. EV5B),
while overexpressing Cpes significantly increased OlyA^W intensity (Fig. EV5C). In
addition, aSMase knockdown led to a drastic increase in the OlyA^W signal (Fig.
EV5D). Importantly, overexpression of Cpes also significantly increased OlyA^W
intensity in the adult brain (Fig. EV5E). These data demonstrate that OlyA^W
fluorescence intensity correlates with CerPE levels."*

SM/CerPE homeostasis is maintained by neuronal neutral SMase and glial acidic
SMase

387-388 Fig. 5g quantification.

**Response 3-22**

We have added the quantification of p62 staining in the New Figure EV6D of the
revised manuscript.

Gal4 should be GAL4

Response 3-23

We have replaced Gal4 with GAL4 throughout the revised manuscript.

Referee #4:

The authors have generated a toolkit of transgenic lines for *Drosophila* sphingolipid
research. CRISPR-engineered knock-in HG lines for 52 sphingolipid metabolism genes
allow GAL4 expression and HA tagging. These lines can also be combined with
degradHA for conditional protein degradation. An XFP (mWasabi) based sensor for
CerPE/sphingomyelin was also developed and it labels membrane rafts. The new
tools are then used to provide evidence that neuronal neutral SMases and glial acidic
SMase maintain CerPE levels and that neurons release CerPE, which is then taken up
by glia.

The 52 CRISPR-engineered HG lines will be very useful to the fly community for
defining the expression patterns as well as the physiological roles of sphingolipid
enzymes in many different biological contexts. These HG lines are rather similar in
structure and function to the Bellen lab's CRIMIC lines, although, as the authors
show, there are neural expression pattern differences that, in some cases, may be
due to whether or not the endogenous 3'UTRs are included in the reporter mRNA.
Recapitulation of endogenous expression patterns by HG lines is validated in the
adult brain using scRNAseq data sets. There are, however, numerous major criticisms
(listed below) of both the tool validation and the mechanistic biology sections that
must be addressed.

We thank the reviewer for the insightful comments. The revision involves a
significant overhaul to focus on the validation of genetic tools, including the HG lines,
deGradHA, and OlyA^w, and, as a proof-of-principle example, the demonstration of
cell type-specific functions of SMases in the brain. To refine the scope and enhance
the coherence of this paper, we have removed the data on changes in the aging
brain and the mechanism of lipoprotein shuttling. We have also enhanced the review
of current literature by including relevant sphingolipid studies using the *Drosophila*
*melanogaster* in the Introduction, Results, and Discussion sections. We believe these
revisions significantly improve the quality of this paper and better demonstrate the
utility of the genetic toolkits to the fly community.

Below are the point-by-point responses to the reviewer's comments.

MAJOR CRITICISMS:

1) The authors should provide evidence as to which of the HG knock-ins maintain
gene function and which disrupt it e.g are the HG knock-ins homozygous viable or

lethal?

**Response 4-1**

We agree that it is important to verify if the HG knock-ins disrupt the functions of the
endogenous genes. As suggested by the reviewer, we now report the viability of all
HG lines in the revised manuscript.

*Line146-156: "We isogenized the HG lines to the w^{1118} background through three
generations of backcrossing and then examined whether the HG insertion caused
any lethality. Although the coding sequence and regulatory elements were not
edited, 5 out of 52 isogenized HG lines, including schlank, acsl, Elov17, spin, and
Sap-r, showed lethality at the adult stage (Table EV1). Additionally, when these
five lines were crossed with Cre stocks to remove the 3XP3RFP selection marker,
the genotype of interest could not be recovered. Besides the five lethal alleles, the
CG11426 HG line was infertile after Cre-excision. These observations suggest that
while most HG insertions did not cause any gross phenotypes, there are instances
where HG insertion can disrupt gene function, potentially from adding the 3XHA
tag to the c-terminus of the protein."*

2) The HG knock-in driver expression patterns and protein localizations are not
described in sufficient detail to be useful to the community. To have a better
appreciation of the cell-type specificity of expression of each enzyme it is important
to display elav and repo channels separately from mCherryNLS and keep the same
scale between images (extended Fig. 4). The repo staining seems variable between
the different lines. For instance, ifc and schlank seem to have a higher number of
glial cells compared to lace and kdsr (extended Fig. 4). Also, it should be clarified
whether the neuron/glia ratio is calculated only in the area displayed or in the whole
brain.

**Response 4-2-1**

We agree that presenting images of each channel with consistent co-stainings of
neuronal and glial markers would greatly improve data quality. This issue was also
brought up by Referee #1 (Response 1-10). In the revised manuscript, we used the
3D images of the whole-mount brain for quantification and showed that glia
accounts for 10% of all brain cells from 182 adult brains analyzed, which is consistent
with the literature (Raji and Potter, 2021). We now showed the representative image
of whole-mount brains for each channel and an enlarged view of a single slice from
the z-stack images with immunostainings to demonstrate the cell type-specific
expression patterns (New Fig 3, Appendix Fig. S5-21).

Kdsr-HA protein seems to have a much wider expression than its transcription
pattern (Fig. 3), why is this? Further testing with an independent method would be
required to support the authors surprising conclusion that Kdsr protein is
transported from neurons to glia. Merged channel overlay panels should be added to
Fig. 3 to aid interpretation. CG6962-HA also seems to have a broader expression
pattern than the CD8:GFP reporter, why is this?

**Response 4-2-2**

We thank the reviewer for highlighting the discrepancy between the GFP and HA
staining patterns in the original Figure 3, an issue also raised by Referee #1
(Response 1-12). We have repeated these experiments multiple times and

consistently observed the HA signal to be broader than the GFP signal. We
considered several potential explanations for this observation, including:
1. Protein stability: The HA-tagged protein has a longer half-life than GAL4 and
mCD8GFP, resulting in the detection of protein where the gene was no longer
actively transcribed.
2. Protein transport: The protein may be transported to non-expressing cells.
3. Detection sensitivity: The transcription activity detected by the GAL4/UAS system
may be below the detection limit, while the protein staining through antibody
hybridization and the biotin-streptavidin amplification method (rabbit-anti-
HA→anti-rabbit-biotin→streptavidin-Alexa fluorophore) revealed a signal.

While these explanations are plausible, further investigation is needed to elucidate
the underlying mechanism. We have added a paragraph in the Discussion section to
discuss plausible cell non-autonomous activities of SPL regulators and will explore
this intriguing observation in future studies.

*Line520-540: "Furthermore, we consistently observed a broader distribution of*
*anti-HA signals compared to fluorescent markers driven by HG, which raises the*
*intriguing possibility of intercellular transport of SPL-regulating enzymes within the*
*brain. Evidently, the cause of this phenomenon is likely gene-dependent, and the*
*discrepancy may also be attributed to other factors. For one, the differential*
*stability of 3XHA-tagged proteins and fluorescent markers might lead to a*
*temporal mismatch in signal detection. Or, low levels of transcription in certain*
*cells may result in detectable protein expression, as previous studies also observed*
*the discrepancy between the levels of cognate protein and mRNA molecules*
*(Buccitelli & Selbach, 2020). Altogether, while these findings could reveal novel*
*insights into the complex regulation of SPL pathways in the brain, further*
*investigation is needed to elucidate the underlying mechanisms."*

3) The usefulness of the HG lines for conditional protein degradation relies on how
efficient the protein knockdown with deGradHA is. Functional validation of this
appears to rely on a non-quantitative comparison between a few confocal panels,
from a region that is not specified i.e no low power "overview" images shown (Fig
5b-c and 7e). A reliable quantitative method is required for these new genetic tools,
which are being reported here for the first-time e.g similar to the western blot
readout for protein levels used in Ext data Fig 1a.

**Response 4-3**

We thank the reviewer for highlighting the need for quantitative validation of
protein knockdown efficiency using deGradHA. To address this, we performed
Western blot analysis of the levels of 3XHA-tagged proteins in *CG6962/dSMPD4* and
*aSMase* HG lines upon neuronal and glial deGradHA expression and provided
quantified results (New Fig. 5C-D; New Fig. EV4A-B).

*Line294-302: "Given that HG GAL4 activity showed that dSMPD4 is preferentially*
*expressed in neurons and that aSMase exhibited glia-enriched expression (Fig. 3C),*
*we targeted these SMases by neuronal or glial expression of deGradHA. Western*
*blot results showed that neuronal expression of deGradHA (driven by nSyb-LexA)*
*drastically reduced the levels of neuronal enriched dSMPD4-3XHA protein but not*

*the glia-enriched aSMase-3XHA protein in the brain (Fig. 5C; Fig. EV4A). In contrast,*
*glial expression of deGradHA (driven by repo-LexA) depleted aSMase-3XHA while*
*leaving dSMPD4-3XHA levels unchanged (Fig. 5D; Fig. EV4B). These results*
*demonstrated that deGradHA can control protein degradation in a cell type-*
*specific manner.”*

We have also included both overview and enlarged images in the revised manuscript
(New Fig. 5E; New Fig. EV4E-F) to provide a better context for the immunostaining
data.

4) The lipid selectivity/specificity of the new OlyA^w sensor for CerPE needs further
validation using a reliable quantitative method. Showing non-quantitative confocal
images of sensor expression in control, Cpes knockout and O/E flies side-by-side is
not sufficient (Fig. 4). Also, what is the weak and diffuse signal remaining in the Cpes
knockouts and what percentage does this apparently non-specific signal contribute
to the signal detected in control genotypes?

**Response 4-4**

We agree that quantitative analysis is necessary for the validation of the OlyA^w
probe. This issue was also brought up by the Referee #3 (Response 3-21). In the
revised manuscript, we now provide quantitative analysis of the OlyA^w pattern
changes in Cpes-knockout (New Fig. 6B-C), OlyA^w subcellular distributions (New
Appendix Fig. S49-50), and OlyA^w fluorescence intensity (New Fig. EV5A-D).
Regarding the background signals of OlyA^w, Referee #2 has also pointed out that
diffuse OlyA^w, which likely represents unbound OlyA^w, should be observed in cells in
which GAL4 is expressed independent of CerPE levels (Response 2-7-2). Indeed, we
observed a diffuse signal of OlyA^w in the L3 salivary gland of Cpes mutants compared
to the control (New Fig. 6B). However, since the OlyA^w pattern was significantly
different between control and Cpes^{KO}, what percentage of the background
contributes to signals detected in control genotypes remains elusive. Despite the
background signals, we confirmed that OlyA^w fluorescence correlates with the
cellular level of CerPE (New Fig. EV5A-D, New Fig. EV6B).

Line 328-346: *“We then examined the subcellular distribution of OlyA^w using*
*both the brain and L3 salivary glands, which contain larger cells more amenable*
*to subcellular localization. While the CerPE synthase (Cpes) located mainly in the*
*Golgi apparatus (Vacaru et al, 2013), OlyA^w did not colocalize with Golgi*
*markers GM130 (cis-Golgi) and Golgin245 (trans-Golgi) (Appendix Fig. S49A).*
*Instead, the OlyA^w fluorescence is mainly found in cortical ER in the L3 salivary*
*glands (Appendix Fig. S49B). In both adult brains and L3 salivary glands, OlyA^w*
*preferably colocalized with the ER membrane marker calnexin over other*
*organelles (Appendix Fig. S49-50). To determine whether OlyA^w fluorescence can*
*detect CerPE, we expressed OlyA^w in control and a CRISPR-knockout line of Cpes.*
*In Cpes-knockout salivary glands, the OlyA^w fluorescence became diffuse (Fig.*
*6B), and its colocalization with the ER membrane marker was significantly*
*reduced (Fig. 6C). The fluorescent intensity of OlyA^w on the ER membrane also*
*significantly decreased in Cpes-knockout salivary glands (Fig. 6D). Moreover,*
*knockdown of Cpes also reduced OlyA^w intensity in the salivary gland (Fig. EV5B),*
*while overexpressing Cpes significantly increased OlyA^w intensity (Fig. EV5C). In*
*addition, aSMase knockdown led to a drastic increase in the OlyA^w signal (Fig.*

*EV5D). Importantly, overexpression of Cpes also significantly increased OlyA^w*
*intensity in the adult brain (Fig. EV5E). These data demonstrate that OlyA^w*
*fluorescence intensity correlates with CerPE levels."*

5) The localisation of the tubulin-GAL4 driven OlyA[w] signal seems mostly in
neurons in the larval brain (Fig. 4b-c) can the authors explain this? Is the OlyA[w]
signal also visible in glia?

**Response 4-5**

*Indeed, the OlyA^w signals driven by tubulin-GAL4 appeared predominantly neuronal.*
*This is due to the high neuronal density in the brain, which is composed of*
*approximately 90% neurons and 10% glial cells. To verify if OlyA^w can be expressed in*
*glia, we examined its expression under the control of repo-GAL4 (pan glial driver)*
*and 54C-GAL4 (a pigment glial driver). OlyA^w driven by repo-GAL4 labeled glial nuclei*
*and membrane processes. (New Fig. EV5F), and OlyA^w driven by 54C-GAL4 labeled*
*the inter-ommatidial spaces (New Fig. 6E-G). These results confirmed that OlyA^w can*
*also be visible in glia.*

6) There are problems with the results section on "Glial lipoprotein mediates
neuronal CerPE levels". The evidence supporting the two-way shuttling model in
Figure 6 is very incomplete. The author knockdown several gene involved in lipid
transport, some of them, such as apolpp, apoltp and Mtp, have never been
described as being expressed in glia or neurons to our knowledge. It is critical to
show evidence that Mtp, apolpp and apoltp are indeed expressed in the cells that
are being targeted for knockdown. At present the effects of glial and neuronal
knockdowns have been assessed with only one readout in nSyb neurons, the CerPE
sensor, and this has yet to be properly validated (see point 4 above). Therefore, the
CerPE sensor should also be expressed in glia with both the neuronal and glial RNAi
manipulations and an independent readout for non-cell autonomous effects on lipid
metabolism would be required to support the two-way shuttling model.
Alternatively, this biological mechanism section could be deleted. This would be
acceptable as according to the title and abstract, the main point of the manuscript
appears to be to provide a new resource not to advance new biological mechanism
in detail.

**Response 4-6**

*We agree that more evidence is needed to support the mechanism of lipoprotein-*
*mediated CerPE shuttling, including lipoprotein expression and function in the brain*
*and CerPE detections in glial cells. Referee #2 had similar concerns regarding the*
*shuttling mechanism (Response 2-8). However, as suggested by Referee #4, we*
*decided to remove the section on lipoprotein-mediated CerPE shuttling after*
*discussing it with the editor.*

7) Related to the incomplete evidence for the Fig. 6 model and for the Fig.5 results, a
second independent genetic method is needed to validate all RNAi results. e.g a non-
overlapping RNAi line or a mutant. This is important as the effect sizes observed with
all of the RNAi lines (except neuronal knockdown of apoltp) are modest (less than
two fold).

**Response 4-7**

As mentioned in Response 4-6, we have removed the lipoprotein shuttling
mechanism from the revised manuscript. For the results shown in Figure 5 of the
original manuscript (corresponding to New Fig 7A-B), we repeated the experiments
using independent RNAi lines for *CG6962/dSMPD4* and *nSMase*, and the results
supported our conclusion that *CG6962/dSMPD4* is required in neurons while *aSMase*
is required in glia for neuronal CerPE homeostasis (New Fig. 7A-B). We further
showed that the knockdown of *aSMase* led to significantly increased OlyA^w intensity
in glia (New Fig. EV6A). Consistent with the OlyA^w results, our lipidomic data showed
that glial knockdown of *aSMase* increased the CerPE levels in the brain (New Fig.
EV6B). However, knockdowns of *nSMase* using independent RNAi lines in either
neurons or glia were not sufficient to change OlyA^w signals. We performed qPCR
analysis to examine the knockdown efficiency of independent *nSMase* RNAi lines
(VDRC 107062 and BDSC 36759) driven by *nSyb*-GAL4, and both *nSMase* RNAi lines
showed knockdown efficiency of less than 50% (Figure to Reviewers. 2). Together, in
the revised manuscript, we validated that *CG6962/dSMPD4* is required in neurons
and that glial *aSMase* exerts both cell-autonomous and non-autonomous functions.

*Figure for referee with unpublished data has been removed upon request by
the authors.*

8) Similarly, the authors refer to UAS-Luc-RNAi as the RNAi control, which in the
Materials Table is given the BDSC ID of #35788. This line is not an RNAi line but a
luciferase overexpression line. So, either the ID number is wrong, or the authors did
not use an RNAi control, which would mean that they cannot rule out non-specific
effects of activating the RNAi machinery. Also, what is the control genotype in Fig 5,
is it the same or different from Fig 6?

**Response 4-8**

We apologize for the wrong citation of the fly strain and the missing information on
the control genotype. The UAS-LUC-RNAi used in this study is BDSC 31603 (*y[1] v[1];*
*P{y[+t7.7] v[+t1.8]=TRiP.JF01355}attP2*). The genotype of the control and
experimental groups was spelled out in the Figure Captions of the revised
manuscript. The source of all strains are provided in the Methods.

9) In Fig.5h, the example image of *nSMase*-RNAi shows a higher OlyA^w signal than
*CG6962*-RNAi (especially in the neuropil region), but the quantification shows that
the increase is not as significant as *CG6962*-RNAi. Could the authors also clarify how
the quantification is made (see point below).

**Response 4-9**

We apologize for the mismatched representative images and the lack of
quantification methods for the SMase knockdown experiments. Since the neuronally
expressed OlyA^w probe predominantly labels cell bodies but not neuronal processes,
we measured the intensity in the cortical region of the Calyx from the posterior view
and excluded the neuropil from the quantification. The quantification details are
described in the Methods of the revised manuscript.

10) In addition to the previous point, lipidomic analysis (Fig. 7a-c) shows that
nSMaseKO leads to accumulation of CerPE, which seems inconsistent with the
OlyA[w] result (Fig. 5h). Could the authors explain this discrepancy?

**Response 4-10**

We thank the reviewer for the comment on the discrepancy between the OlyA^w
experiment using *nSMase* RNAi knockdowns and the lipidomic analysis using
*nSMase*-knockout flies. As mentioned in Response 4-7, we realized that *nSMase*
RNAi only reduced the RNA level up to 50%. The residual nSMase, despite RNAi
knockdown, might be sufficient to maintain CerPE catabolism in the adult brain.
Therefore, the discrepancy between the OlyA^w experiment using *nSMase* RNAi
knockdowns and the lipidomic analysis using *nSMase*-knockout flies could be
explained by the low knockdown efficiency of *nSMase* RNAi lines.

11) p62 seems to accumulate in the optic lobe of glial knock-out of aSMase (Fig. 5g).
However, it is important to confirm the finding of p62 accumulation independently,
using at least one of tools/methods used by the authors in this study e.g glial RNAi of
aSMase or the whole body CRISPR mutant. In addition, quantification of fluorescent
signals is lacking and should be performed for Fig. 5g.

**Response 4-11**

We thank the reviewer for the comment on validating the accumulation of p62 in
the glial knockout of aSMase. As an independent method, we dissected the adult
brains and performed p62 immunoblots, but the results did not show changes in p62
levels from the brain extracts of RNAi knockdown or conditional knockout flies of
*aSMase* (Figure to Reviewers. 3).

Figure for referee with unpublished data has been removed upon request by the authors.

However, we do not think results from immunoblotting and immunostaining
contradict each other. One possibility could be that the immunoblots of the entire
brain may not reflect the altered p62 levels that took place specifically in the optic
lobe.
In the revised manuscript, we validated the phenotype with more biological repeats
and provided a quantitative analysis of the p62 accumulation in the optic lobe (New
Fig. EV6C-D).
12) The method for confocal image fluorescent intensity quantifications (currently in
Fig. 5 and 6) needs to be added to the Methods section, specifying the region of
analysis (which part of cell bodies or neuropil), the software used, parameters being
quantified, and the normalisation methods used to account for sample-to-sample
intensity differences e.g due to mounting. In addition, are the fold changes shown
for the RNAi knockdowns in Fig 6a-b relative to "UAS-Luc-RNAi"? Please specify.
**Response 4-12**
We apologize for the missing details on the quantification method. As mentioned in
Response 4-9, we measured the intensity in the cortical region of the Calyx from the
posterior view. We use Image J for quantitative analysis and the Macros for brain

OlyA^W quantification is now provided as a supplemental document. The mean
intensity of OlyA^W for each sample was normalized to the UAS-LUC-RNAi control of
the same experiment, which accounts for the batch effects across independent
experiments. We did not normalize the sample-to-sample intensity difference within
the same experiment. The quantification details are now described in the Methods
section of the revised manuscript.

13) At present, the reactions catalysed by many *Drosophila* enzymes including those
of sphingolipid metabolism are not known. Instead, they are inferred from sequence
similarity/orthology to mammalian or yeast enzymes, where the actual enzymology
has been done. This type of inference can lead to mistakes as the exact substrates
and products of some enzyme classes are challenging to predict from only their
sequence. The authors should describe if/how their new tools will assist with filling
this important knowledge gap in the *Drosophila* metabolism field. In addition, it
would be very useful to add an additional column to Extended Table 1, indicating
whether the stated enzyme activity is known by direct assay or inferred by orthology.

**Response 4-13**

Indeed, the enzyme activities of most target proteins were not tested. We have now
added a column in Table EV1 to indicate whether the stated enzyme activity is
examined indicated by mutant phenotypes or inferred from similarities in sequence
and structure.

We have also added a paragraph in the Discussion section to spell out how these
reagents would advance SPL studies in *Drosophila*. Gene profiling using HG lines
serves as an important foundation, as it reveals the endogenous gene activity for
subsequent genetic manipulations by identifying tissue-specific expression and
temporal regulations. Moreover, functional analyses combining HG lines and RNAi
transgenes, CRISPR-knockout lines, or the deGradHA construct are also useful to
reveal the site of requirement and the link with specific phenotypes. The HG toolkit
allows systemic gene profiling and functional analysis of all candidate genes involved
in the SPL metabolism for a comprehensive understanding of the regulatory network
that controls enzyme functions and SPL landscapes *in vivo*.

*Line459-476: "The HG toolkit has several advantages over existing methods and*
*resources. We engineered GAL4 activity to be controlled by the complete cis-*
*regulatory element. Consequently, any subsequent investigation using the*
*Gal4/UAS system will occur under the endogenous control of the target gene.*
*Because of this design, the set of 3xHA-T2A-Gal4 knockin animals enables*
*spatiotemporal profiling of protein localization without the need to raise*
*antibodies. This imaging-based method allowed most SPL regulators to be directly*
*compared in terms of their expression patterns and protein distributions (Fig. 3-4).*
*Next, the integration of deGradHA with the LexA/LexAop system enhanced the*
*versatility for genetic manipulation (Fig. 5). By combining deGradHA with cell type-*
*specific promoters, we can selectively degrade the target protein in specific cell*
*types and observe its impact on distant tissues, allowing investigation of*
*nonautonomous functions of secreted proteins. Moreover, deGradHA directly*
*targets the protein of interest, overcoming common limitations of conditional gene*
*knockout/knockdown strategies, such as discrepancies between gene activity*
*versus protein level, and inefficient depletion of proteins with long half-lives. Hence,*

*the deGradHA complements existing methods of genetic manipulation, especially*
*when studying the spatiotemporal functions of proteins.”*

MINOR CRITICISMS:

1) Because of the stronger sphingolipid phenotype of nSMaseKO, it would be
interesting to characterize the possible brain phenotypes in the same way as
CG6962KO (Fig. 7f-h).

**Response 4-14**

We agree. In the revised manuscript, we performed the olfactory associative
learning assays (New Fig. 7J) and examined the feeding behaviors (New Appendix Fig.
S55C) in *nSMase^{KO}* flies. We found that associative learning ability is impaired in
*CG6962/dSMPD4^{KO}* flies at both 1 week and 3 weeks of age, whereas *nSMase^{KO}* flies
exhibit learning deficits at 3 weeks of age and not at 1 week (New Fig. 7J).

Additionally, we observed abnormal feeding behaviors of both *CG6962/dSMPD4* and
*nSMase* knockout flies at 1 week of age (New Appendix Fig. S55C).

2) Magnifications and scale bars would be better if they were kept the same
between images within the same figure (Figure 3, Figure 6, Extended Data Figure
1,4,5).

**Response 4-15**

We agree. In the revised manuscript, magnifications and scale bars were kept the
same among images from the same experiment except for the enlarged panels.

3) Introduction, lines 79-90 wrongly imply that lipidomics can only capture steady-
state-profiles not dynamics. However, it is well established that lipidomics, in
combination with stable isotope tracing, is a very powerful method for probing lipid
dynamics. The authors may want to mention this in the Introduction.

**Response 4-16**

We agree that lipidomics could detect the dynamic of lipids when combined with
isotope tracing. In the revised manuscript, we have toned down the statement in the
Introduction section.

*Line88-92: “While lipidomics by mass spectrometry is required to resolve the*
*molecular features necessary for identifying individual lipid species, standard mass*
*spectrometry profiling typically pools lipid species across entire organisms or*
*tissues. Imaging mass spectrometry techniques, though promising, remain limited*
*in the spatiotemporal resolution (Wang et al, 2022a; Miller et al, 2023).”*

4) There are multiple discrepancies in the text figure citations, some of which refer
to the wrong figures.

**Response 4-17**

We apologize for the wrong figure citations in the text. We have reorganized the
figures and checked repeatedly to make sure each panel is cited correctly.

5) A key is required to show blue corresponds to CG6962KO and grey to nSMaseKO
in Fig. 7a

6) In Fig.7b, the red line is not on "1".

7) In Fig.7d-h, black dots are not visible on black bars.

**Response 4-18**

We thank the reviewer for spotting the errors. We have modified the figures
accordingly in the revised manuscript (New Fig.7).

Dear Dr. Chan,

Thank you for transferring your manuscript to EMBO Reports, which was previously revised and re-reviewed at The EMBO Journal. I have now carefully read all documents and remaining referee concerns. As discussed before, we would like to offer publication at EMBO Reports, pending satisfactory minor revision.

I note that referees #1 and #3 appreciate the revision and recommend publication. However, referees #2 and #4 have some outstanding concerns. To address the overlapping concern of referees #2 and #4 regarding the OlyAw staining, please acknowledge the limitations in the text and tone down the relevant conclusions. Similarly, please address the concern of referee #4 regarding the RNAi lines by textual alterations. Please address these concerns and provide a complete point-by-point response and the manuscript text where all changes are marked.

Moreover, the editorial points below need to be addressed before I can accept the manuscript.

- We note that a previously published transcriptomic dataset is re-analyzed in the manuscript (Davie et al, 2018), which should be cited in the format of data citation. Please see <https://www.embopress.org/page/journal/14693178/authorguide#referencesformat> for details.
- Figure legends need to be included in the main manuscript text at the end, which are currently provided in a separate file.
- Please provide 3-5 keywords for your study. These will be visible in the html version of the paper and on PubMed and will help increase the discoverability of your work.
- We note that the doi's provided in the Data Availability section (10.17632/gg794sznsz.1; mCD8GFP, DOI: 10.17632/fty6n2yjxz.1) are currently not accessible.
- Along similar lines, please provide links that directly resolve to the deposited dataset in the Data Availability section. Moreover, this section is reserved for the primary datasets generated in the study. Therefore, please remove the following sentences: "The image data that support the findings of this study are available from the corresponding author, C.-C.C., upon reasonable request" and "The reagents, including HG lines, deGradHA, and OlyAw are deposited at the KYOTO Drosophila Stock Center." The second sentence should be included in the Reagent and Tools table with the relevant accession numbers.
- Please rename the Declaration of interests section as Disclosure and Competing Interests Statement.
- Please remove the Author Contributions section from the manuscript text.
- Funding information should be complete both in the manuscript text and the manuscript tracking system. We note that this information is currently missing from the manuscript tracking system. The Comments box should not be used as funding information can only be retrieved from the separate entries during publication.
- We note that Table EV1 is currently missing a legend, which should be included in the file itself as a separate tab. We note that Appendix Table S1 is also suited to be an EV table - i.e. Table EV2. If you prefer to keep it as an Appendix Table, it should be included in the Appendix file. Either way, it should also have a legend.
- The legends of the Appendix figure should be included in the Appendix file, under every relevant figure. Also, the Appendix file is currently missing a Table of Contents, where page number of every item is listed.
- Reagents & Tools table needs to be removed from the manuscript text and submitted as a separate word file.
- Please fill in and submit a source data checklist (please see https://www.embopress.org/pb-assets/embo-site/EMBOPress_Source_Data_Checklist.xlsx).
- Abbreviations need to be removed (provided in the legends file). Abbreviations should be defined in brackets after their first mention in the text, not in a list of abbreviations.
- Our production/data editors have asked you to clarify several points in the figure legends - Figure Legends (main + EV):
 - o Please note that the exact p values are not provided in the legends of figures 3C, 4C, 5C, 7A, B, C, D, E, G, I; EV4 C, EV5 D, E.
 - o Please indicate the statistical test used for data analysis in the legend of figure 2D.
 - o Please note that the box plots need to be defined in terms of minima, maxima, centre, bounds of box and whiskers, and percentile in the legend of figure 7G.
 - o Please note that information related to n is missing in the legend of figure 2D.
 - o Please note that the error bars are not defined in the legend of figure 2D.
 - o Please note that the 6H, I; EV3 arrowhead are not defined in the legend of figure. This needs to be rectified.
- The file containing 'synopsis' and 'bullet points' needs to be submitted separately.
- In addition, please provide an image for the synopsis. This image should provide a rapid overview of the question addressed in the study but still needs to be kept fairly modest since the image size cannot exceed 550 (width) x 300-600 (height) pixels. We note that a synopsis image is provided in the manuscript file, which should be provided as a separate file. Also, I note that it is on the busier side. During resubmission, please make sure that when re-sized as mentioned above, all items are visible and labels are legible.

Thank you again for giving us to consider your manuscript for EMBO Reports, I look forward to your minor revision.

Kind regards,

Deniz Senyilmaz Tiebe

--

Deniz Senyilmaz Tiebe, PhD
Senior Scientific Editor
EMBO Reports

Response to the Editor

Dear Dr. Chan,

Thank you for transferring your manuscript to EMBO Reports, which was previously revised and re-reviewed at The EMBO Journal. I have now carefully read all documents and remaining referee concerns. As discussed before, we would like to offer publication at EMBO Reports, pending satisfactory minor revision.

I note that referees #1 and #3 appreciate the revision and recommend publication. However, referees #2 and #4 have some outstanding concerns. To address the overlapping concern of referees #2 and #4 regarding the OlyAw staining, please acknowledge the limitations in the text and tone down the relevant conclusions. Similarly, please address the concern of referee #4 regarding the RNAi lines by textual alterations. Please address these concerns and provide a complete point-by-point response and the manuscript text where all changes are marked.

Response to Editor:

We appreciate the opportunity for the minor revision and the guidance provided by the editor. Please note that changes made in the revised manuscript file are marked in red font. Below are point-to-point responses to the editorial points and referee comments.

Moreover, the editorial points below need to be addressed before I can accept the manuscript.

- We note that a previously published transcriptomic dataset is re-analyzed in the manuscript (Davie et al, 2018), which should be cited in the format of data citation. Please see <https://www.embopress.org/page/journal/14693178/authorguide#referencesformat> for details.

Response to Editor 1:

We have edited the citation in the text where the published dataset was used. We have also edited the reference list in the format of data citation.

Line 194-203 (Results): *"We further compared the cell-type expression patterns of these genes with single-cell transcriptomics of the adult brain (Appendix Fig. S11-K; <https://scope.aertslab.org/> (Davie et al, 2018; Data ref: Davie et al., 2018)). Clustering revealed both glial and neural expression of *sply* and *lace* (broader than CRIMIC and HG lines), and strong glial enrichment of *mdy* (consistent with HG lines). We chose *wun2* for further validation against the patterns of SPL metabolic genes that were well resolved in subtype glia in the transcriptomic dataset (Davie et al, 2018; Data ref: Davie et al., 2018), because *wun2* was identified as a marker gene of an astrocyte-like glia cluster (Fig. 2F). "*

- Figure legends need to be included in the main manuscript text at the end, which are currently provided in a separate file.

Response to Editor 2:

We have added the Figure legends at the end of the main manuscript.

- Please provide 3-5 keywords for your study. These will be visible in the html version of the paper and on PubMed and will help increase the discoverability of your work.

Response to Editor 3:

We have added five keywords (Brain; Sphingolipids; systemic profiling; spatial heterogeneity; cell-type specificity) after the Abstract.

- We note that the doi's provided in the Data Availability section (10.17632/gg794sznsz.1; mCD8GFP, DOI: 10.17632/fty6n2yjxz.1) are currently not accessible.

Response to Editor 3:

We have published the data, making it publicly accessible.

- Along similar lines, please provide links that directly resolve to the deposited dataset in the Data Availability section. Moreover, this section is reserved for the primary datasets generated in the study. Therefore, please remove the following sentences: "The image data that support the findings of this study are available from the corresponding author, C.-C.C., upon reasonable request" and "The reagents, including HG lines, deGradHA, and OlyAw are deposited at the KYOTO Drosophila Stock Center." The second sentence should be included in the Reagent and Tools table with the relevant accession numbers.

Response to Editor 4:

We have removed the sentences regarding data deposition. We have included the list of flies in the Reagents and Tools table.

- Please rename the Declaration of interests section as Disclosure and Competing Interests Statement.

Response to Editor 5:

We have renamed the section as “*Disclosure and Competing Interests Statement*.”

- Please remove the Author Contributions section from the manuscript text.

Response to Editor 6:

We have removed the Author Contributions section.

- Funding information should be complete both in the manuscript text and the manuscript tracking system. We note that this information is currently missing from the manuscript tracking system. The Comments box should not be used as funding information can only be retrieved from the separate entries during publication.

Response to Editor 7:

We have updated the funding information in the manuscript and the tracking system.

- We note that Table EV1 is currently missing a legend, which should be included in the file itself as a separate tab. We note that Appendix Table S1 is also suited to be an EV table - i.e. Table EV2. If you prefer to keep it as an Appendix Table, it should be included in the Appendix file. Either way, it should also have a legend.

Response to Editor 8:

We have added the legends for Table EV1 and Appendix Table S1.

- The legends of the Appendix figure should be included in the Appendix file, under every relevant figure. Also, the Appendix file is currently missing a Table of Contents, where page number of every item is listed.

Response to Editor 9:

We have included the legends in the Appendix file and added a Table of Contents, where page number of every item is listed.

- Reagents & Tools table needs to be removed from the manuscript text and submitted as a separate word file.

Response to Editor 10:

We have submitted the Reagents and Tools table as a separate word file.

- Please fill in and submit a source data checklist (please see

https://www.embopress.org/pb-assets/embosite/EMBOpress_Source_Data_Checklist.xlsx).

Response to Editor 11:

We have uploaded a source data checklist to the manuscript tracking system.

- Abbreviations need to be removed (provided in the legends file).

Abbreviations should be defined in brackets after their first mention in the text, not in a list of abbreviations.

Response to Editor 12:

We have removed the Abbreviation list in the figure legends..

- Our production/data editors have asked you to clarify several points in the figure legends - Figure Legends (main + EV):
 - o Please note that the exact p values are not provided in the legends of figures 3C, 4C, 5C, 7A, B, C, D, E, G, I; EV4 C, EV5 D, E.

Response to Editor 12:

We have provided the exact p values in the figure legends.

- o Please indicate the statistical test used for data analysis in the legend of figure 2D.

Response to Editor 13:

We have indicated the statistical test used for data analysis in the legend of Figure 2D, defined the error bars, and provided the exact p value and sample size.

- o Please note that the box plots need to be defined in terms of minima, maxima, centre, bounds of box and whiskers, and percentile in the legend of figure 7G.

Response to Editor 14:

We have defined the box plots in the legends of Figure 7G.

- o Please note that information related to n is missing in the legend of figure 2D.

(Please see Response to Editor 13)

- o Please note that the error bars are not defined in the legend of figure 2D.

(Please see Response to Editor 13)

- o Please note that the 6H, I; EV3 arrowhead are not defined in the legend of figure. This needs to be rectified.

Response to Editor 15:

We have defined the arrowhead in the legends of Figures 6H, I, EV3.

- The file containing 'synopsis' and 'bullet points' needs to be submitted separately.

Response to Editor 16:

We have submitted the synopsis and bullet points separately.

- In addition, please provide an image for the synopsis. This image should provide a rapid overview of the question addressed in the study but still needs to be kept fairly modest since the image size cannot exceed 550 (width) x

300-600 (height) pixels. We note that a synopsis image is provided in the manuscript file, which should be provided as a separate file. Also, I note that it is on the busier side. During resubmission, please make sure that when resized as mentioned above, all items are visible and labels are legible.

Response to Editor 16:

We have simplified the synopsis image (550 x 600).

Response to the Referees

Referee #1:

We have carefully reviewed the revised version of the manuscript "Functional profiling and visualization of the sphingolipid metabolic network in vivo", along with the authors' response to reviewers. We appreciate the thorough effort the authors have put into addressing the previous concerns.

The authors have addressed our major concern of better resolution images, and they have made significant edits to the manuscript as suggested.

Additionally, the authors have conducted many of the requested experiments and the new data significantly strengthen the manuscript.

We find the authors' responses to the reviewer comments to be detailed and satisfactory. In our assessment, the revised figures and analyses are appropriate and contribute meaningfully to the overall clarity and robustness of the work. Overall, we consider the manuscript to be in a significantly improved state and believe it presents a solid body of work with clear scientific merit.

Response 1: We thank the referee for acknowledging the value of this manuscript and the support for its publication.

Referee #2:

Fei-Yang Tzou et al., 205 (revised manuscript)

In the revised manuscript, the authors have streamlined the resource section and withdrawn several claims made in earlier version of the manuscript. This revision primarily focuses on the resource aspect of the study and the authors have made endeavors to deposit the data and provide access to the tools through a public resource center. Barring one major concern requiring the authors to edit the results and discussion section of the manuscript, we have no other serious objection to the publication of the manuscript.

Response 2: We thank the referee for the careful review of our revised manuscript and the insightful feedback regarding the strengths and caveats of

our study. Below is our response to the remaining concern regarding the utility of OlyA^w biosensor.

Major concern

The surprising finding is the fact that the authors do not find robust plasma membrane staining of OlyA^w. If the protein was indeed mostly secreted as the authors intended, and it was binding to CPE, we would anticipate and expect strong and universal plasma membrane staining across cells and tissues. The findings and the data provided by the authors most likely indicate that the designed OlyA^w probe results in substantial fraction of the protein not folding properly during the synthesis. This is the reason the authors see a predominant ER staining {which shows a Pearson correlation coefficient of 0.4 in the control and close to 0.3 in the *cpes* mutant (should have been down to almost zero in the absence of endogenous CPE, if the staining was specific)}. Generally, secreted proteins rarely show an intracellular pattern of localization unless under very defined special circumstances. In the absence of existence of data to prove specific staining of ER due to binding to CPE, it is important that statements such as "OlyA^w fluorescence is mainly found in cortical ER in the L3 salivary glands" do not misguide the field to conclude that CPE is found abundantly in the ER in *Drosophila* tissues. We would not like to belabor this point further other than require the authors to unequivocally indicate such staining patterns could result from proteins undergoing misfolding during synthesis in the ER.

Response 2-1: We agree that the OlyA^w probe is expected to be mainly located on the plasma membrane or the Golgi lumen because of the secretory sequence and its binding to CerPE. However, the OlyA^w showed a cortical ER pattern and colocalized with the ER membrane marker in the adult brain and the L3 salivary glands. This ER localization was only slightly reduced in the CerPE-deficient/*Cpes*^{KO} backgrounds. The minor difference in OlyA^w between control and *Cpes*^{KO} suggested that the majority of the OlyA^w fluorescence colocalized with the ER marker is background signals that may be due to protein misfolding or inefficient ER export during protein secretion. Therefore, without further evidence of its binding to CerPE in the ER, it remains an open question whether the intracellular OlyA^w signals indicate CerPE localization. Given the limitation, we have clarified the limitation and toned down our conclusion in the revised manuscript.

Line 328-348 (Results): “We then examined the subcellular distribution of OlyA^w using both the brain and L3 salivary glands, which contain larger cells more amenable to subcellular localization. While the CerPE synthase (Cpes) located mainly in the Golgi apparatus (Vacaru et al, 2013), OlyA^w did not colocalize with Golgi markers GM130 (cis-Golgi) and Golgin245 (trans-Golgi) (Appendix Fig. S49A). Instead, the OlyA^w fluorescence is mainly found in cortical ER in the L3 salivary glands (Appendix Fig. S49B). In both adult brains and L3 salivary glands, OlyA^w preferably colocalized with the ER membrane marker calnexin over other organelles (Appendix Fig. S49-50). In Cpes-knockout salivary glands, the OlyA^w colocalization with calnexin was slightly reduced (Fig. 6B-C). These results indicated that while intracellular localizations of OlyA^w are partially dependent on Cpes, the background fluorescence of ER retention limits its utility for reporting the subcellular localization of CerPE lipids. Next, we validated whether the intensity of OlyA^w fluorescence correlates with cellular levels of CerPE. The fluorescent intensity of OlyA^w on the ER membrane significantly decreased in Cpes-knockout salivary glands (Fig. 6D). Moreover, knockdown of Cpes also reduced OlyA^w intensity in the salivary gland (Fig. EV5B), while overexpressing Cpes significantly increased OlyA^w intensity (Fig. EV5C). In addition, aSMase knockdown led to a drastic increase in the OlyA^w signal (Fig. EV5D). Importantly, overexpression of Cpes also significantly increased OlyA^w intensity in the adult brain (Fig. EV5E). These data demonstrate that OlyA^w fluorescence intensity correlates with CerPE levels.”

Line 380-383 (Results): “These findings support that OlyA^w can be used to indicate cellular CerPE levels in a cell types-specific manner. Moreover, despite diffuse and potentially non-specific signals within the cell, the punctate OlyA^w signals reveal the dynamics of CerPE-enriched membrane rafts.”

Line 594-605 (Discussion): “While the OlyA^w biosensor is useful to indicate cellular accumulations of CerPE and to reveal membrane raft dynamics, a key consideration when interpreting the OlyA^w signal is the background fluorescence in the cortical ER, which was also observed in Cpes^{KO} cells, where CerPE is mostly depleted. This suggested that a fraction of the biosensor may misfold and become trapped in the ER, resulting in a signal that is independent of CerPE's presence. Moreover, severe cellular

phenotypes of Cpes mutants (Kunduri et al, 2018, 2022) may impair protein homeostasis of OlyA^w itself, complicating the interpretation of how OlyA^w reflects CerPE reduction and spatial distribution. Given both the OlyA^w intrinsic localization in the ER and the confounding effects of the mutant background, its signal should not be used to infer the subcellular localization of CerPE without corroborating evidence.”

Referee #3:

I am impressed with the revision, and think the ms is sufficiently improved to be published.

Response 3: We thank the referee for the positive comment on the revision and the support for the publication.

Referee #4:

There is no doubt that the authors extensive revisions involving several additional experiments have improved the manuscript. Many of our initial criticisms have now been satisfactorily addressed, including deleting the overinterpreted section on "Glial lipoprotein mediates neuronal CerPE levels" and adding some text to the Discussion about possible reasons for the GFP versus HA discrepancy in staining. However, there remain three of our original points that do not appear to have been adequately addressed and where further experimental data are still needed:

Response 4: We thank the referee for the positive comments on the revision and suggestions for improving the scientific rigor of our manuscript. Below are point-to-point responses to the remaining concerns regarding the validation of the OlyA^w biosensor and RNAi lines.

RESPONSE 4-4:

The author stated that "OlyA^w did not colocalize with Golgi markers GM130 (cis-Golgi) and Golgin245 (trans-Golgi) (Appendix Fig. S49A)" but in the quantification in Fig. S49 and S50, Golgi markers gave a relatively high Pearson's coefficient, suggesting co-localisation. The authors need to explain

this clearly.

Response 4-1: As the referee noted, we conclude that OlyA^w did not colocalize with Golgi markers by visual inspection, but the quantification of Pearson's coefficient ($r = 0.262$) of OlyA^w and anti-GM130 stainings is the second highest among all the markers tested. One could speculate that the discrepancy is due to the colocalization of OlyA^w with non-specific stainings of anti-GM130 in the cortical ER structure, which was not shown in the representative figure (Appendix Fig. S49B). We have explained this explicitly in the Result section of the revised manuscript.

Line 335-338 (Results): *Despite a lack of obvious visual overlap with the Golgi markers, quantitative analysis showed a modest correlation ($r = 0.262$) between OlyA^w and anti-GM130 signals, which could be attributed to non-specific antibody staining within the cortical ER.*

The change in OlyA^w intensity with Cpes KO is minimal (Fig 6B, 6D) but much greater than with Cpes RNAi (Fig. EV5B). The authors need to explain this and provide validation that the Cpes KO line is effective.

Response 4-2: We thank the reviewer for this insightful question. The difference in OlyA^w intensity between the Cpes^{KO} and Cpes RNAi lines is complicated by two factors:

1. Potential background signals: As also noted by referee 2 (please see our Response 2-1), the knockout of Cpes did not cause a dramatic change in OlyA^w localization and intensity. The minor difference in OlyA^w between control and Cpes^{KO} suggested that the majority of the OlyA^w fluorescence colocalized with the ER marker is background signals that may be due to protein misfolding or inefficient ER export during protein secretion.
2. Severity of the Cpes^{KO} phenotype: A complete knockout of Cpes causes severe cellular stress and structural defects (Kunduri et al., 2018), which may impair the cell's general ability to synthesize, modify, and degrade proteins, including the OlyA^w biosensor itself. This is a significant confounding variable in the Cpes^{KO} that is likely less pronounced in the partial knockdown from RNAi.

Therefore, even though the CerPE level is most depleted in the Cpes^{KO}, structural defects and background fluorescence independent of CerPE

binding may cause the overall OlyA^w intensity to be higher than expected in the KO background compared to the RNAi knockdown.

We have clarified the issue of background fluorescence in the Results and Discussion sections of the revised manuscript.

Line 328-348 (Results): *“We then examined the subcellular distribution of OlyA^w using both the brain and L3 salivary glands, which contain larger cells more amenable to subcellular localization. While the CerPE synthase (Cpes) located mainly in the Golgi apparatus (Vacaru et al, 2013), OlyA^w did not colocalize with Golgi markers GM130 (cis-Golgi) and Golgin245 (trans-Golgi) (Appendix Fig. S49A). Instead, the OlyA^w fluorescence is mainly found in cortical ER in the L3 salivary glands (Appendix Fig. S49B). In both adult brains and L3 salivary glands, OlyA^w preferably colocalized with the ER membrane marker calnexin over other organelles (Appendix Fig. S49-50). In Cpes-knockout salivary glands, the OlyA^w colocalization with calnexin was slightly reduced (Fig. 6B-C). These results indicated that while intracellular localizations of OlyA^w are partially dependent on Cpes, the background fluorescence of ER retention limits its utility for reporting the subcellular localization of CerPE lipids. Next, we validated whether the intensity of OlyA^w fluorescence correlates with cellular levels of CerPE. The fluorescent intensity of OlyA^w on the ER membrane significantly decreased in Cpes-knockout salivary glands (Fig. 6D). Moreover, knockdown of Cpes also reduced OlyA^w intensity in the salivary gland (Fig. EV5B), while overexpressing Cpes significantly increased OlyA^w intensity (Fig. EV5C). In addition, aSMase knockdown led to a drastic increase in the OlyA^w signal (Fig. EV5D). Importantly, overexpression of Cpes also significantly increased OlyA^w intensity in the adult brain (Fig. EV5E). These data demonstrate that OlyA^w fluorescence intensity correlates with CerPE levels.”*

Line 380-383 (Results): *“These findings support that OlyA^w can be used to indicate cellular CerPE levels in a cell types-specific manner. Moreover, despite diffuse and potentially non-specific signals within the cell, the punctate OlyA^w signals reveal the dynamics of CerPE-enriched membrane rafts.”*

Line 594-605 (Discussion): *“While the OlyA^w biosensor is useful to indicate cellular accumulations of CerPE and to reveal membrane raft dynamics, a key consideration when interpreting the OlyA^w signal is the background fluorescence in the cortical ER, which was also observed in Cpes^{KO} cells, where CerPE is mostly depleted. This suggested that a fraction of the biosensor may misfold and become trapped in the ER, resulting in a signal that is independent of CerPE's presence. Moreover, severe cellular phenotypes of Cpes mutants (Kunduri et al, 2018, 2022) may impair protein homeostasis of OlyA^w itself, complicating the interpretation of how OlyA^w reflects CerPE reduction and spatial distribution. Given both the OlyA^w intrinsic localization in the ER and the confounding effects of the mutant background, its signal should not be used to infer the subcellular localization of CerPE without corroborating evidence.”*

For the validation of the CRISPR/Cas9-mediated knockout of *Cpes*, we observed the dorsal closure defect, the photo-sensitive epilepsy phenotype, and the semi-lethality in the pupal stage as the previously reported *Cpes* mutant (Kunduri et al., 2018). We also conducted genomic PCR and quantitative RT-PCR to validate *Cpes* depletion at the molecular level.

RESPONSE 4-7:

The authors have now used two independent RNAi lines for dSMPD4 and nSMase to show consistent changes in OlyA^w intensity with neuronal or glial drivers. However, the huge difference between the Bloomington and VDRC dSMPD4 RNAi lines (Fig. 7A) suggests that the VDRC line is not effective. So, with only 1 of 2 independent RNAi lines working, that still leaves open the possibility of an off-target effect. Therefore, the previous criticism still stands: "a second independent genetic method is needed to validate all RNAi results. e.g a non-overlapping RNAi line or a mutant.". Unfortunately the same is also true for aSMase.

Response 4-3: While we acknowledge the potential off-target effects of RNAi lines in the OlyA^w experiments, we want to highlight multiple lines of evidence that indicate the cell type-specific role of dSMPD4 and aSMase in regulating CerPE degradation. First, dSMPD4 was specifically expressed and distributed to neurons, while aSMase was expressed in glia (Figure 3, 4). Regarding dSMPD4, CRISPR/Cas9-mediated knockout of dSMPD4 caused the accumulation of specific CerPE species in the brain lipidomic analysis (Fig.

7D), supporting the role of dSMPD4 in catalyzing CerPE degradation. Furthermore, the neuron-specific function of dSMPD4 was demonstrated by the nuclear blebbing phenotype observed upon cell type-specific protein degradation (Appendix Fig. S55A-B). Despite potential off-target effects of the RNAi line, collective evidence suggests that dSMPD4 regulates CerPE degradation in neurons.

For aSMase, both RNAi lines (VDRC 12227 and BDSC 36760) caused significant increases in OlyA^w intensity in the L3 salivary glands (Fig. EV5D), indicating the role of aSMase in regulating CerPE degradation. Glial knockdown of aSMase using the VDRC 12227 RNAi line caused CerPE accumulations in both OlyA^w and brain lipidomic analysis (Fig. EV6A-B). However, the RNAi line (BDSC 36760) was very difficult to maintain, and we lost the line during revision. So, the experiment could not be repeated. As an independent genetic method to validate whether aSMase functions specifically in the glia, somatic knockout experiments using UAS-aSMase-sgRNA and UAS-Cas9 driven by cell type-specific GAL4s were performed (Fig. EV6C-E). Glia-specific knockout of aSMase led to Ref(2)P accumulation and locomotion defects (Fig. EV6C-E). These results indicate that aSMase functions in glia to regulate CerPE degradation.

RESPONSE 4-10:

The authors have clarified the discrepancies between OlyA^w and lipidomic experiments with the effectiveness of the RNAi checked by PCR. The PCR data provided for Reviewer #2 should be included in the supplementary data as it is important information. It would also be preferable but not essential to include OlyA^w results using the KO lines for both nSMase and dSMPD4 to see if they show the same OlyA^w or lipidomic trends as the RNAi lines.

Response 4-4: We have included the RT-qPCR data in the Appendix of the revised manuscript (Appendix Fig. S52).

Dr. Chih-Chiang Chan
National Taiwan University
Graduate Institute of Physiology
1 Sec1 Jen-Ai Rd
Rm1043
Taipei 100
Taiwan

Dear Dr. Chan,

I am pleased to inform you that your manuscript has been accepted for publication in EMBO reports. Your manuscript will be processed for publication by EMBO Press. It will be copy edited and you will receive page proofs prior to publication. Please note that you will be contacted by Springer Nature Author Services to complete licensing and payment information.

Yours sincerely,

Kurt Weir
Editor
EMBO Reports
